# Salvador Urban Network Transportation (SUNT): A Landmark Spatiotemporal Dataset for Public Transportation

## Abstract

Efficient public transportation management is essential for the development of large urban centers, providing several benefits such as comprehensive coverage of population mobility, improvement of the local economy with the offer of new jobs and the decrease of transport costs, better control of traffic congestion, and significant reduction of environmental impact limiting gas emissions and pollution. Realizing these benefits requires carefully pursuing two essential pathways: (i) deeply understanding the population and transit patterns and (ii) using intelligent approaches to model multiple relations and characteristics efficiently. This work addresses these challenges by providing a novel dataset that includes various public transportation components alongside machine learning models trained to understand and predict different real-world behaviors. Our dataset comprises daily information from about 710,000 passengers in Salvador, one of Brazil's largest cities, and local public transportation data with approximately 2,000 vehicles operating across nearly 400 lines, connecting almost 3,000 stops and stations. As benchmarks, we have fine-tuned diverse Graph Neural Networks to perform inference on vertices and edges, undertaking both regression and classification tasks. These models leverage temporal and spatial features concerning passengers and transportation data. We emphasize the greatest advantage of using our dataset lies in different possibilities of modeling a real-world urban mobility dataset, reproducing our results, overcoming selected models, and investigating several other open-problem situations listed in this manuscript as future work, which include the designing of new methods, optimization strategies, and environmental approaches. Our dataset, codes, and models are available at `https://github.com/suntdataset/sunt.git`.

## 1 Introduction

Efficient urban mobility requires a branch of strategies to manage traffic, delivering improvements, such as increasing safety, reducing travel time, decreasing costs, and supporting environmental issues. Each strategy has guided researchers to explore different proposals like vehicle-to-vehicle communication, route optimization, adoption of the Internet of Things for connecting transit devices, and effective scheduling of the public transportation system [Zhang et al., 2011, Rahmani et al., 2023].

In this work, we focus our investigation on the public transportation system due to its importance to the population. Moreover, any decision regarding this system directly impacts urban mobility, especially in developing countries, where it is often the only means of transport available to low-income populations. When poorly planned, it delivers low-quality services with delayed and overloaded vehicles, concentrates traffic in specific regions while leaving others unattended, and aggravates pollution with higher gas emission rates.

Intelligent Transportation Systems address these issues by incorporating data monitoring, heuristics to extract new information, and Artificial Intelligence (AI) to support decision-making tasks. In summary, such systems collect and analyze data from passengers and vehicles to uncover implicit patterns that can identify bottlenecks, sudden changes, and areas for improvement.

As discussed by Ceder [2016], achieving a successful public transportation system depends on three key elements. Firstly, it requires gathering and understanding adequate data. Next, the data must be utilized for intelligent planning and decision-making. Finally, the plans and decisions must be effectively implemented to ensure smooth operations and control.

To exemplify the adoption of such keys in real-world scenarios, researchers have collected data from passengers' fares and vehicle locations to create a heuristic that estimates boarding and alighting times and locations in London (England) [Gordon et al., 2013, Wang et al., 2011]. Similarly, researchers collected passenger and vehicle data from Harbin (China), which was later modeled using unsupervised machine learning methods to understand public transit riders' travel patterns better [An et al., 2017]. Researchers from Seoul (the Republic of Korea) also developed a methodology for estimating non-tagged alighting stop information gradually, by considering the characteristics of trip types and utilizing transportation card data [Lee et al., 2021]. In New York (USA), researchers analyzed data from the transit system, where riders swipe a fare card only when entering a station or boarding a bus. They used this information to estimate alighting stops based on the bus boarding locations [Barry et al., 2009]. The similar problem was addressed in Southeast Queensland (Australia) by using Deep Neural Network. This network was used to predict unknown alighting locations after being trained in a dataset with a combination of transactional and public transit network attributes [Assemi et al., 2020]. Although these aforementioned citations are more related to our work, further research on the problem of inferring boarding-alighting locals in public transportation systems is detailed in a review published by [Mohammed & Oke, 2023].

After an in-depth investigation on published manuscripts focused on public transportation, we noticed a limitation on the availability of totally public dataset containing a comprehensive quantitative, spatial, and temporal information about passengers, vehicles, lines, stops, and stations. In this paper, we overcome this issue by making available a massive dataset with all the data from Salvador (Brazil) collected for five months in 2024. Salvador is the capital of Bahia, a state located in the northeast region of Brazil. Salvador is situated in an area of approximately 694 km$^2$ with a population of around 3 million and a Gross Domestic Product (GDP) per capita near USD 4,220.18.

Our dataset, referred to as Salvador Urban Network Transportation (SUNT), is organized into two parts: (i) a raw set of data that the reader can process according to their needs, and (ii) a graph connecting all these data, respecting their geospatial and temporal restrictions. In addition, we also shared adjusted models as benchmarks, since classical time series analyses to Graph Neural Networks, to perform classification and regression tasks on vertices and edges. Benchmarking public transportation datasets forms the foundation for developing effective, efficient, and sustainable transportation systems that meet the evolving needs of society.

In summary, the main contributions of this work are as follows:

- A set of four raw datasets collected over five months with information about vehicles, passengers, and stops/stations;

- A preprocessed dataset as a complex network with several attributes derived from the bus velocities, time and distance between stops, boarding-alighting information, and so on;

- A variety of models using time series and Graph Neural Networks, along with their parameters, to ensure the reproducibility and improvements of our results;

- A list of future investigations that can inspire researchers to use our dataset and increase the current features and information.

## 2 RELATED WORK

Graph Neural Networks (GNNs) have been widely considered in Intelligent Transportation Systems (ITS) [Zhang et al., 2011] for addressing challenges such as more precisely representing complex relationships between vehicles, stations, and passengers. This natural capability to model such relationships and handle irregularly structured data has boosted their advancement in the field. Real-world graph datasets of public transportation are vital in advancing research, innovation, and real-time decision-making in route planning, scheduling, and resource allocation [Rahmani et al., 2023, Iliopoulou & Kepaptsoglou, 2019]. Constructing datasets is a significant challenge in developing

models for ITS due to the difficulty of collecting and integrating comprehensive data while also respecting privacy when gathering information from traffic sensors, users, and GPS.

In addition, public transportation datasets play an important role in modeling spatiotemporal dependencies, enabling adaptation to dynamical change in network structures, identifying significant agents affecting trajectories, and accurately representing long-term dependencies. Currently, only a few publicly available ITS datasets exist with some of these features as, for example, TaxiBJ [Zhang et al., 2017, Bai et al., 2019], BikeNYC [Zhang et al., 2017, Bai et al., 2019], Shanghai Metro dataset [Xie et al., 2023], Hangzhou Metro dataset [Liu et al., 2022, Ren et al., 2023, Xie et al., 2023], Beijing Metro dataset [Zhang et al., 2020, Xie et al., 2023], Chongqing Metro [Xie et al., 2023], Stockholm County [Klar & Rubensson, 2024], METR-LA [Jiang et al., 2023, Wu et al., 2019, Cini et al., 2023, Du et al., 2021], PEMS-BAY [Wu et al., 2019, Chen et al., 2020, Shao et al., 2022, Oreshkin et al., 2021, Cini et al., 2023, Du et al., 2021], and UVDS [Bui et al., 2021, Rahmani et al., 2023]. Therefore, most available datasets for graph-based transportation representation lack spatiotemporal features and integration of multifaceted data necessary for effective adaptation in dynamic urban environments [Hu et al., 2020, Poursafaei et al., 2022, Huang et al., 2024]. Moreover, recent studies on GNN advancement have underscored deficiencies in current benchmark datasets [Shchur et al., 2018], including limited graph availability, smaller scale in vertices and edges, and constrained class diversity. As a result, creating large, high-quality, and comprehensive graph datasets remains a significant challenge both for GNN research and for ITS applications [Li et al., 2024]. In addition to providing a comprehensive dataset, this work offers a significant advantage by including a set of pre-trained models as benchmarks. These benchmarks range from traditional time series models to varied GNN architectures, as discussed in the following sections.

## 3 SUNT DATASET CONSTRUCTION

The transportation used by the local population in Salvador comprises three systems: regular buses, subway, and BRT (Bus Rapid Transit). The regular bus system is the most extensive transportation in Salvador, serving most of the population. Currently, there are about 1,900 buses distributed on approximately 400 lines with around 3,000 stops and stations, supporting roughly 470,000 passengers daily. The subway system spans about 35 km across 2 lines with 20 stations. Approximately 210,000 passengers use this system daily. The BRT (Bus Rapid Transit) system was recently inaugurated, further enhancing urban mobility and serving about 30,000 passengers daily. Currently, about 40 buses are operating on 3 lines and 20 stations.

Our dataset, referred to as Salvador Urban Network Transportation (SUNT), contains information from all systems organized into raw and processed data. The processing steps considered in this work explore concepts based on trip chaining, as detailed next.

### 3.1 TRIP CHAINING

In this study, we utilized an Automated Data Collection System (ADCS) to gather data from multiple sources [Mohammed & Oke, 2023], resulting in two distinct raw datasets. The first dataset was obtained from the Automatic Vehicle Location (AVL) system, which monitors all regular and BRT buses, providing details about their geospatial positions over time. The second dataset, the Automatic Fare Collection (AFC) system, contains information from the ticketing systems, recording the time when users' contactless cards are used for payments. In addition to the exact time of card usage, it also includes details on the vehicles and their respective lines.

Additionally, we used static data based on the General Transit Feed Specification (GTFS) format, which defines a standard format for public transportation schedules associated with geographic information (http://gtfs.org/). Using this format, we provided geospatial details about stations and stops along with their sequential order, lines, and directions. Finally, we also provide a dataset containing Local Trip Information (LTI), which includes details about the expected and actual departure and arrival times for all vehicles on every line and in each direction. Due to the dynamic nature of data collected from the AVL system, missing data may occur, resulting in random loss of information about vehicle activities. This issue can be easily addressed by combining redundant vehicle information from GTFS and LTI.

After organizing these four datasets (AVL, AFC, GTFS, and LTI), the first challenge is to find out the boarding locations for all users. As illustrated in Figure 1(a), this information is computed by integrating AVL and AFC, and retrieving the exact latitude and longitude positions when the users' cards performed the payments. Using these positions, we can estimate the closest stop or station that indicates the boarding local. Next, we merge multiple boarding locations to classify the users' trips as initial, intermediate, and final. Such a classification is relevant to map all possible connections that compose a complete user's trip. Finally, all boarding positions with their respective time instants are used to organize trip chains describing the passengers' behavior.

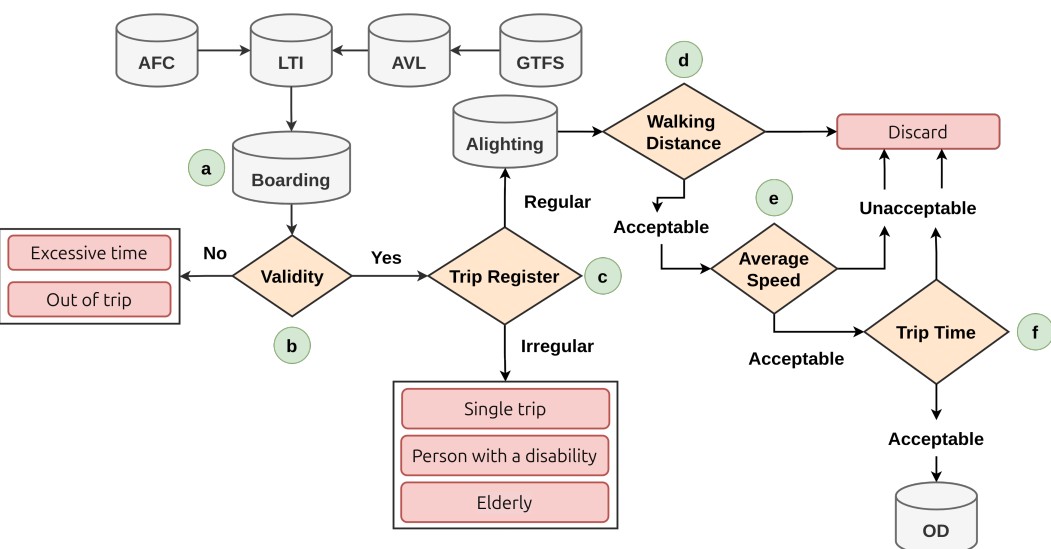

Figure 1: Steps used to create our origin-destination dataset. Red boxes represent boarding data with no alighting correspondence.

In the next phase, Figure 1(b), we assess the validity of the boarding registration by checking two specific conditions. Firstly, a user's boarding is discarded if the time difference between the AFC-recorded fare payment and the AVL-recorded bus arrival at the stop exceeds a certain threshold. This threshold has two possible values: (i) 20 minutes for bus stations; and (ii) 5 minutes for regular stops. This differentiation is necessary because buses typically remain longer at stations. Secondly, another discarding possibility happens when there is no direct connection between AVL and AFC records, which is considered in this figure as "out of trip".

In the subsequent phase, Figure 1(c), we analyzed user types to determine the feasibility of estimating their alighting points. In Salvador, there is no device to validate the passengers' alighting; therefore, the main challenge is to estimate it by analyzing the following boarding. Moreover, it is impossible to track older people because they are not individually identified. According to local policies, the fares for such passengers are recorded as general users without identification. Consequently, we are unable to estimate their alighting points. Another particular case that prevents us from identifying users' alighting points occurs when there is only a single trip registration on a given day. In such cases, we can only determine the boarding point, with no information available about the alighting point. Therefore, we cannot consider such situations in our analyses.

For all remaining cases, we can infer the alighting points by combining a set of boarding points per user. To better understand this inference, consider the three scenarios illustrated in Figure 2. In Scenario I, we observe a passenger boarding at 8:00 AM (B1) at Stop (b) and then boarding again at 6:00 PM (B2) at Stop (f). Therefore, it can be inferred that the passenger boarded at Stop (b), disembarked at Stop (f) on the first trip (b → f), and then made the return journey at the end of the day (f → b).

In the second scenario, we observe a user trip with a connection. In this situation, there are two boarding points for each trip. Initially, the user boarded at Stops (b), at 8:00 AM (B1), and (d), 8:20 AM (B2), being the first alighting registered at Stop (d). At the end of the day, the user boarded at

Stops (j), at 6:00 PM (B3), and (d), at 6:10 PM (B4), respectively. Therefore, we infer the first user's trip was b → j starting at 8:00 AM, and their return was j → b at 6:00 PM.

In our final scenario, we illustrate a situation when a user utilizes a connection between two different stops by walking a short distance between them. In this case, they register a first boarding at Stop (b), at 8:00 AM (B1), and the second one at Stop (x), 8:50 AM (B2). As one may notice, Stop (x) is in a different line. Hence, we look for its closest stop, respecting the maximum walking distance (Δ), Stop (f) in this case, to represent the first alighting. Considering they register another boarding at Stop (u), at 7:00 PM (B3), followed by boarding at Stop (f), at 7:30 PM (B4), we can map their full daily trip using the same rule previously considered. Therefore, we infer the first user's trip was b → fΔx → u starting at 8:00 AM, and their return was u → xΔf → b at 7:00 PM.

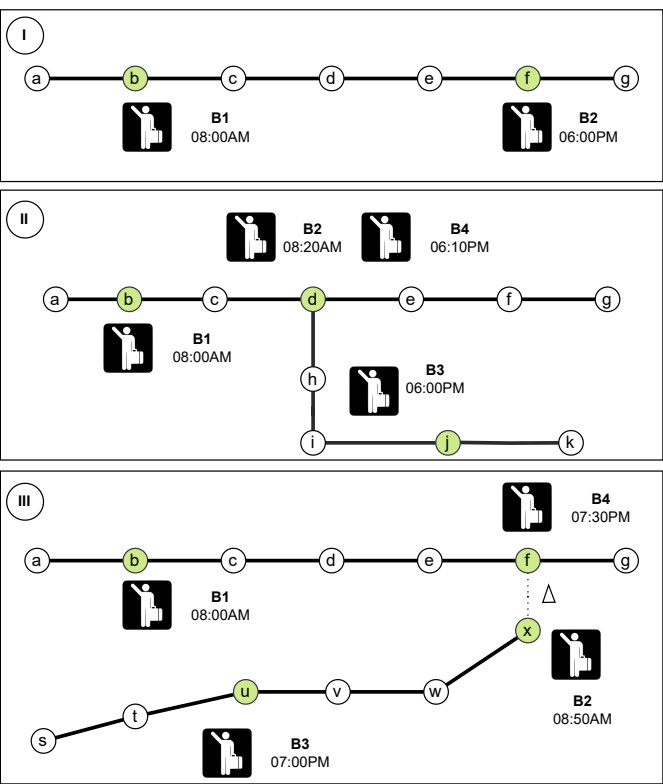

Figure 2: Scenarios illustrating three different boarding-alighting situations: (I) a single line, (II) lines with a connection, and (III) two lines connected by walking distance.

As shown in Figure 1(d), a walking distance is deemed acceptable if it is limited to 1.1 km. Concerning the average velocity, Figure 1(e), and the trip time, Figure 1(f), all registers with values greater than 80 km/h and 2 hours are unconsidered. These values were estimated by local specialists based on the passengers' usage patterns and the transportation infrastructure in Salvador. We emphasize that the reader can modify these values according to their needs once both raw and processed data are shared.

In Figure 1, all red boxes represent situations in which we cannot precisely use the passengers' occurrences in our analyses. Nevertheless, even in minority cases, it is essential to consider their general behavior to mitigate imprecision in further estimations, such as the load of passengers on the buses. In this case, we use the data distribution for each line to allocate these occurrences across different buses, as recommended by the literature. After this correction, we have the processed Origin-Destination (OD) dataset. The following section details all variables derived from the boarding-alighting information, resulting in our main contribution: the Salvador Urban Network Transportation (SUNT) dataset.

## 3.2 GRAPH MODELING

The organization of the OD dataset with passengers' boarding and alighting allowed us to create the SUNT dataset, embedding a set of quantitative, temporal, and geospatial variables as a complex network. Formally, we have used information on latitude, longitude, and time to create a spatial-temporal graph $G = \{G_1, G_2, ..., G_T\}$. For all $t = 1, ..., T$, $G_t = (V, E)$ stands for an attributed and directed graph at time $t$, where $V = \{v_1, v_2, ..., v_N\}$ is the set of $N$ vertices corresponding to the bus stops and stations, and $E$ is the set of edges corresponding to feasible routes. A directed edge $(v_i, v_j) \in E$ connects vertices $v_i, v_j \in V$ if, and only if, there is a feasible route for the bus traffic from the corresponding station $v_i$ to $v_j$ in the network. $G_t$ is a fixed graph structure since sets $V$ and $E$ do not change over time.

Figure 3(a) shows the map of Salvador with all vertices stored in our SUNT dataset, i.e., stops and stations used by regular and BRT buses, as well as subways. The geospatial information allows us to place them on the map, respecting their actual geographic position and the distances connecting them by the physical streets.

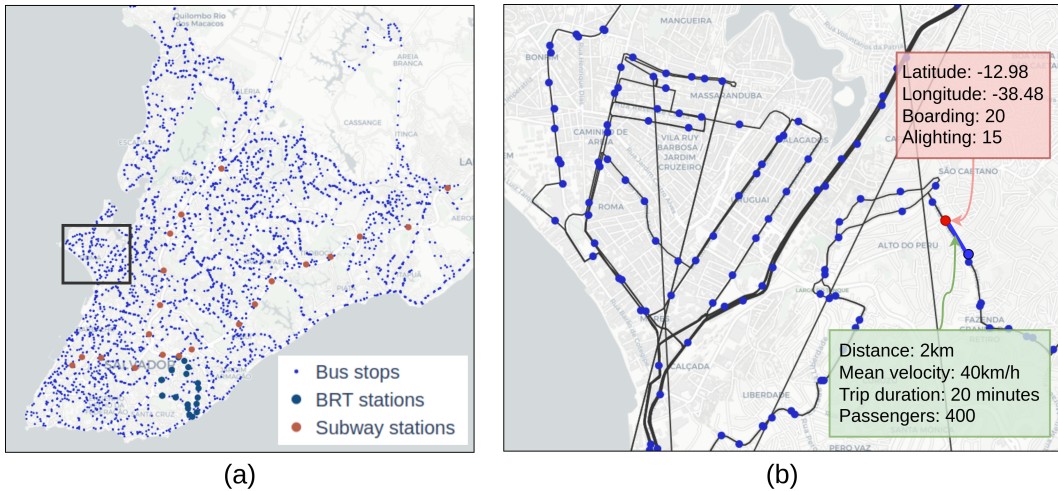

(a)                               (b)

Figure 3: (a) Salvador map with all stops and stations used by regular buses, BRT, and subway; (b) a sample of stops and stations (nodes) represented by blue dots and their respective lines (edges).

In our context, spatial data do not depend on time $t$, i.e., their information is time-invariant. Specifically, in every vertex $v_i \in V$, we store the following features: geographical position, number of boarding and alighting per vehicle, and passenger load. The features specifically concerning edges $(v_i, v_j) \in E$ include the distance between stops and stations, the trip duration, the mean velocity, and the Renovation Factor (RF). The RF is a well-known metric used in transportation research to assess the total demand in a line, i.e., it is computed on a set of edges that belong to the line [ITDP, 2016]. Formally, this metric is the ratio of the total demand of a line to the load on its critical link. Higher renovation factors occur when there are many short trips along the line. Corridors with very high renovation factor rates are more profitable because they handle the same number of paying customers with fewer vehicles [ITDP, 2016]. Besides the individual features, there is relevant information shared by both vertices and edges, such as the number of passengers per vehicle, lines and directions, vehicle characteristics, altitude, and trips.

The black bounding box in Figure 3(a) represents an essential region of the city, which gathers different lines and connections. Figure 3(b) zooms in this region with a portion of the full graph, illustrating some bus stops as vertices and lines connecting them as edges. The red explaining box contains some features related to that bus stop (vertex) such as its latitude and longitude position, and the amount of boarding and alighting passengers. In the green explaining box, we illustrate some features related to a line (edge), such as the distance connecting two stops, the mean velocity and trip duration among the buses in that section, and the total amount of traveling passengers.

All information shared by SUNT was collected from March 2024 to July 2024 and aggregated into 5-minute intervals. This interval allows the data to be represented as a temporal graph, in addition to

the spatial information. However, we emphasize that this interval can be adjusted according to the readers' requirements. It is possible to work with a static graph using a single interval or to summarize all days using, for example, a mean function. Additional details about all the data comprising SUNT are available in Apendix B.

## 4 EXPERIMENTAL SETUP

Besides sharing SUNT, we have also trained models as benchmarks to demonstrate that valuable insights can be derived from different learning tasks. In this manuscript, we focused on three particular learning tasks: node classification, edge classification, and node regression.

In our experiments, we created a temporal graph by selecting data from a work day (March 1st, 2024) whose observations on passenger loading were collected every 5 minutes. To train the classification models, we have selected the following features: passenger loading, mean velocity and distance between stops/stations, total boarding and alighting, and number of lines, vehicles, and trips. The labels were defined by computing the average loading in every node (stop/station). Next, we have selected a single graph from another day (March 8th, 2024 at 7 PM) to test the performances of our models. In this case, every node was encoded as "high" if the loading was greater than its average; otherwise, we considered it "low". The majority class is 60% for high loading.

Similarly, in the edge classification, we have considered the same data interval but labeled edges according to the following rule: if the mean velocity (bus speed) in a route is greater than its average, the label is "high"; otherwise, it was considered low. The majority class is 51% for low velocity. We highlight that these data intervals were designed to address local demands and showcase the capabilities of our dataset. Alternative configurations can easily be adjusted using our provided codes.

We have designed our experiments for node and edge classification using 10-fold cross-validation. We considered the same validation metrics in both tasks to assess the obtained models: Accuracy, F1-score, Matthews Correlation Coefficient (MCC), Precision, and Recall. Aiming to keep the manuscript concise, other relevant metrics, such as the ROC curve and AUC, are shown in Appendix D, along with their mathematical details and interpretations.

Regarding the node regression, we have organized a set of experiments using minimal features, thus making it possible to use traditional univariate time series analyses. Therefore, we predict the passenger load in nodes and only use the distances between stops/stations to weigh edges. Due to the temporal dependencies, we trained the models using a sliding window strategy by fitting the model with 3-hour observations and validating the result with the subsequent one hour. As previously mentioned, each observation summarizes 5-minute data. Further details on the process of creating the time series are shown in Appendix A. The prediction results were assessed by using traditional regression metrics (MSE, MAE, RMSE, MAPE, and $R^2$), whose details are also discussed in Appendix D.

For the node and edge classification, we have trained the following models: CHEB Defferrard et al. [2016], GAT [Veličković et al., 2017] GCN [Kipf & Welling, 2016], SAGE [Hamilton et al., 2017], $S^2$GC [Zhu & Koniusz, 2021], EGC [Tailor et al., 2021], A-DGN [Gravina et al., 2023], LEConv [Ranjan et al., 2020], SuperGAT [Kim & Oh, 2022], and PAN [Ma et al., 2020].

In relation to the node regression, besides such models, we also trained other focused on data sequence: SARIMA [Box et al., 2015], LSTM [Sak et al., 2014], GRU [Cho et al., 2014b], GConvLSTM [Seo et al., 2018], GConvGRU [Seo et al., 2018], TGCN [Zhao et al., 2019], DCRNN [Li et al., 2017], AT3GCN [Bai et al., 2021], CHRONOS [Ansari et al., 2024], and SOFTS [Han et al., 2024]. In Appendix C, we provide more details about all models considered in our experiments.

## 5 BENCHMARKING RESULTS

Table 1 summarizes all results obtained by the three learning tasks explored in this paper as benchmarks. About the node classification, Table 1(a), CHEB presented the best performance for all metrics. In Table 1(b), we noticed all models presented very similar behavior, with a tiny advantage for GAT. Although the performances are around 60%, we considered satisfactory results due to the

complexity of predicting numerical values to edges, and there is learning once the dataset is perfectly balanced. For both tasks, the obtained results summarize the average of all 10 validation folds.

Table 1: Results for the learning tasks: node classification, edge classification, and node regression.

| (a) Node Classification Results | | | | |
|---|---|---|---|---|
| **Model** | **Accuracy** | **F1** | **MCC** | **Precision** | **Recall** |
| **CHEB** | **0.99±0.01** | **0.98±0.03** | **0.98±0.03** | **1.0±0.00** | **0.97±0.05** |
| GAT | 0.91±0.02 | 0.81±0.05 | 0.75±0.06 | 0.81±0.04 | 0.82±0.09 |
| GCN | 0.93±0.04 | 0.82±0.11 | 0.79±0.11 | 0.96±0.03 | 0.73±0.16 |
| SAGE | 0.93±0.03 | 0.84±0.07 | 0.81±0.07 | 0.93±0.05 | 0.78±0.12 |
| $S^2GC$ | 0.97±0.01 | 0.96±0.01 | 0.93±0.02 | 0.96±0.01 | 0.96±0.02 |
| EGC | 0.98±0.01 | 0.98±0.01 | 0.97±0.02 | 0.98±0.03 | 0.98±0.01 |
| A-DGN | 0.93±0.05 | 0.90±0.07 | 0.85±0.01 | 0.95±0.06 | 0.86±0.09 |
| LEConv | 0.98±0.01 | 0.98±0.02 | 0.96±0.02 | 0.98±0.03 | 0.98±0.02 |
| SuperGAT | 0.92±0.02 | 0.89±0.03 | 0.82±0.04 | 0.92±0.03 | 0.87±0.04 |
| PAN | 0.99±0.01 | 0.99±0.01 | 0.98±0.01 | 0.99±0.01 | 0.98±0.01 |

| (b) Edge Classification Results | | | | |
|---|---|---|---|---|
| **Model** | **Accuracy** | **F1** | **MCC** | **Precision** | **Recall** |
| CHEB | 0.51±0.01 | **0.60±0.21** | 0.02±0.04 | 0.45±0.16 | **0.87±0.31** |
| **GAT** | **0.53±0.03** | 0.31±0.32 | **0.08±0.06** | **0.52±0.31** | 0.38±0.42 |
| GCN | 0.52±0.02 | 0.47±0.27 | 0.07±0.05 | 0.50±0.19 | 0.62±0.44 |
| SAGE | 0.50±0.00 | 0.20±0.32 | -0.0±0.02 | 0.25±0.35 | 0.30±0.48 |
| $S^2GC$ | 0.50±0.00 | 0.47±0.32 | 0.01±0.02 | 0.35±0.24 | 0.70±0.48 |
| EGC | 0.50±0.02 | 0.42±0.22 | -0.0±0.04 | 0.46±0.17 | 0.46±0.28 |
| A-DGN | 0.50±0.00 | 0.34±0.35 | 0.02±0.03 | 0.44±0.36 | 0.49±0.52 |
| LEConv | 0.50±0.00 | 0.54±0.28 | 0.02±0.03 | 0.56±0.16 | 0.80±0.42 |
| SuperGAT | 0.51±0.01 | 0.28±0.33 | 0.02±0.04 | 0.27±0.29 | 0.40±0.50 |
| PAN | 0.56±0.02 | 0.60±0.03 | 0.13±0.04 | 0.55±0.02 | 0.66±0.07 |

| (c) Node Regression Results | | | | |
|---|---|---|---|---|
| **Model** | **MSE** | **MAE** | **RMSE** | **MAPE** | **$R^2$** |
| SARIMA | 54154±55442 | 168±125 | 194±143 | 1.39±0.39 | -2.78±0.83 |
| LSTM | 32219±30608 | 115±85 | 145±105 | 0.70±0.22 | -0.85±0.94 |
| GRU | 30463±29276 | 109±83 | 140±103 | 0.71±0.23 | -0.69±65 |
| GCN | 2907±3346 | 32±25 | 44±30 | 0.33±0.05 | 0.73±0.16 |
| CHEB | 1223±1035 | 17±10 | 27±17 | 0.37±0.05 | 0.87±0.08 |
| SAGE | 1259±1027 | 18±10 | 31±17 | 0.32±0.05 | 0.85±0.08 |
| GAT | 11397±10423 | 72±49 | 88±59 | 0.51±0.05 | 0.20±0.16 |
| GConvLSTM | 30075±28953 | 109±82 | 139±102 | 0.82±0.18 | -0.68±0.91 |
| GConvGRU | 29778±28723 | 108±82 | 138±102 | 0.71±0.17 | -0.65±0.91 |
| TGCN | 31356±29931 | 112±84 | 143±103 | 0.70±0.20 | -0.79±0.9 |
| DCRNN | 34109±31976 | 121±86 | 151±105 | 0.86±0.10 | -1.09±0.84 |
| AT3GCN | 36175±33319 | 127±87 | 158±105 | 1.14±0.29 | -1.44±0.60 |
| $S^2GC$ | 1575±1623 | 23±15 | 33±20 | 0.29±0.02 | 0.81±0.13 |
| EGC | 9354±7673 | 61±34 | 84±46 | 0.29±0.02 | -0.01±0.43 |
| A-DGN | 1987±1731 | 23±13 | 38±22 | 0.34±0.05 | 0.81±0.05 |
| **LEConv** | **894±703** | **15±8** | **26±14** | **0.28±0.02** | **0.89±0.06** |
| SuperGAT | 15213±14515 | 83±58 | 101±70 | 0.56±0.08 | 0.04±0.31 |
| PAN | 3967±5745 | 36±33 | 48±40 | 0.40±0.03 | 0.72±0.15 |
| CHRONOS | 2520 ± 2551 | 27±18 | 43±28 | 0.53±0.21 | 0.77±0.04 |

Table 1(c) shows the node regression results. After training all models using a sliding window on observations collected by 7 days, randomly chosen, we have assessed the predicted values over the next 1-day ground truth as shown in Figure 4(a). In this table, the lower the values, the better the results, but the $R^2$ coefficient. Hence, the best general results were obtained with LEConv. These results contain the mean and standard deviation values for 5 selected nodes. In Figure 4(b), we illustrate the predicted values for the top-five regression models estimated for a single node. As noticed in the tabulated results, we highlight the performance achieved by LEConv, which follows the general ground truth behavior.

We emphasize the results presented in this section were performed to illustrate the possibility of obtaining important insights from our data. In Appendice A, we explore additional results, investigate

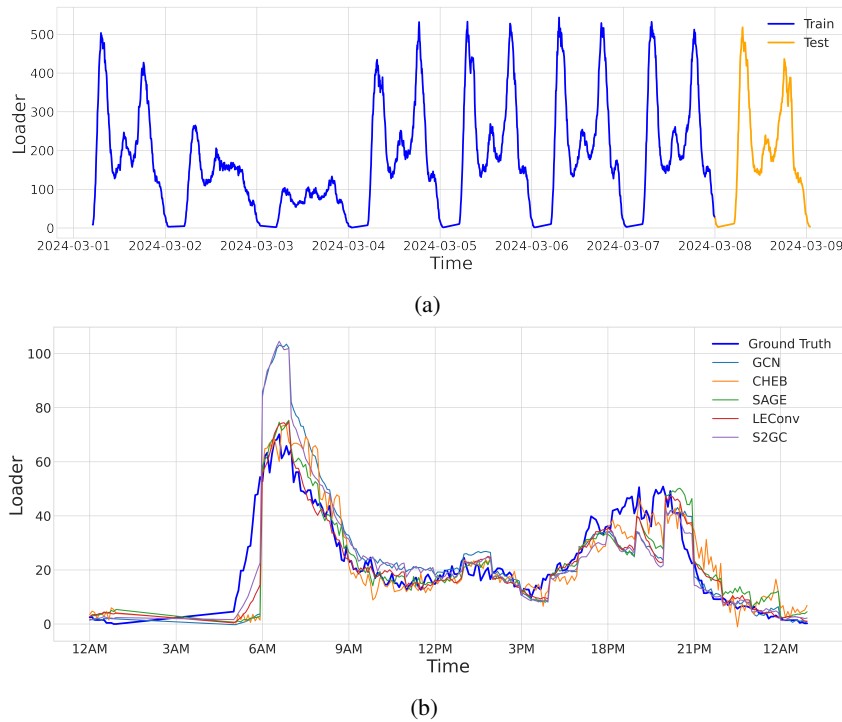

Figure 4: (a) Time series for a single stop with observations sampled in seven days for training and one day for test; (b) Comparison between the expected values (ground truth) and results predicted by different models.

other learning tasks (e.g., the adoption of concept drift methods to detect changes in data streams [Koh et al., 2024]), visualize outcomes, and discuss experiments integrating SUNT with other urban datasets.

## 6 CONCLUSION

This paper introduced SUNT, a novel dataset collected from public transportation in Salvador, Brazil. This dataset is notably relevant to the scientific community for supporting investigations in several domains, such as planning public transportation, designing computational approaches, and managing environmental impact. As previously mentioned, other researchers have published related datasets, ratifying the importance of this subject. However, our dataset stands out due to its massive information and complete availability. Unlike manuscripts that only share outcomes, we have fully shared collections of raw and graph-based details of vehicles, passengers, stations, time, and geographic properties.

We also analyzed SUNT using diverse time series and machine learning models to demonstrate the feasibility of the learning process, showing that it is possible to derive valuable insights from our data. To ensure the entire reproducibility of our results, along with all datasets, we have shared our pre-trained models with hyper-parameters, architecture details, sources, notebooks, and experiments.

**Potential Positive and Negative Impacts:** Analyses from SUNT pave new ways to provide positive social impacts, such as better planning the allocation of buses to lines, reasonably defining regular and express trips, thus reducing traffic jams and carbon emissions, and offering better trip experiences. About negative impacts, SUNT is not automatically updated, neither allowing the inclusion of new data nor detecting general changes. However, we have locally designed a protocol to maintain and evolve the datasets and benchmarks over time to attract new users and citations. Details about how we plan to keep and update SUNT are listed in Appendix E.

**Limitations:** Although we have sought the best-known models in the literature, their architectures and parametrization can be individually analyzed to improve the results. Secondly, we explored a limited variation of attributes available in our dataset. Learning from other attributes or broad combinations of them is also possible. Thirdly, we have analyzed three learning tasks based on node and edge classification and regression. Researchers can also use different attributes and their transformations as targets. Fourthly, other graph structures may provide important information, mainly varying the edges' weights. Finally, we have predefined some parameters related to the application, such as 5-minute intervals and 1.1km walking distance, due to the particularities of our local scenario. Such definitions may not attend other research. However, by operating our shared scripts used to create SUNT, readers can use the raw datasets to redefine them according to their needs.

**Future work:** By sharing SUNT and the pre-trained models, we expect to provide a robust dataset along with baselines for the community, supporting the advancement of several investigation possibilities like time-based models, graph algorithms, spatial approaches, deep neural networks, routing simulations, and search heuristics. To illustrate such possibilities, we have listed future work that is worth investigating from our perspective: (i) GNN approaches designed to pass messages using both temporal and spatial information; (ii) Multi-objective optimization approaches to find the shortest path based on edges weighted by distance and time considering traffic jam; (iii) Multimodal GNN combines different features (e.g., temporal, spatial, numerical, and categorical data) with varying encoding approaches as message passing; (iv) Queue theory to address the problem of attending passengers from a stop A to B; (v) Concept Drift methods designed to identify when passengers' pattern changes in real-world automatically; and (vi) GNN to guide search processes in meta-heuristics, e.g., in a multi-agent evolutionary algorithm, each agent handles a part of the search and, in each generation, GNN could help to select the most suitable agent at each step of the evolutionary process. In Appendices A and A.5, we present preliminary experiments that demonstrate the investigation discussed in (v) and the integration with other urban datasets for various applications, respectively.

## Author Contributions

This section was omitted due to the double-blind requirements.

## Acknowledgments

This section was omitted due to the double-blind requirements.

## Reproducibility Statement

To ensure the reproducibility and completeness of this paper, besides sharing all data and codes in a software repository, we have included five appendices. Appendix B presents the general information of the dataset, including details on both the raw and graph-based data. Appendix C provides a theoretical background about all the learning models used as benchmarks, including details about their architectures and parameters. Appendix D contains the complete evaluation process designed to train and assess the performance of these models. Appendix A presents an in-depth discussion of potential future work, while Appendix A.5 illustrates an example of integration with other urban datasets. Appendix E presents complete information about the dataset and benchmark access.

## Ethics Statement

In our work, only one attribute raised a potential ethical concern: the identification of users' cards in the AFC data. However, this identification does not correspond to the actual card numbers, but rather an internal code that cannot be used to retrieve any personal information from external access. Although such recovery is highly unlikely, we implemented a hash-based solution (collision-free) to convert all internal identifications, adding an additional layer of privacy. It is important to note that no other attribute links individual users to their public transportation usage.

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

# A SUPPLEMENTARY RESULTS

The proposed dataset opens several avenues for future research. Its size, robustness, and structured design make it a valuable resource for advancing the state of the art in multiple areas, as demonstrated by the diverse learning tasks discussed in this appendix.

## A.1 TIME SERIES ANALYSES

To illustrate analyses over time, we selected some stops/stations in which the transit of passengers is intense, with different possibilities for connections between lines and buses. Figure 5 shows a time series (in blue) whose observations represent the loading of passengers at a given station, collected every 5 minutes from March 1st, 2024 to March 9th, 2024. As one may notice, the time series is characterized by a significant frequency fluctuation as noise that may affect its modeling and prediction. To overcome this issue, we used simple moving average (SMA) approach, with a window size of 12 observations, to smooth the time series, as illustrated by the red line.

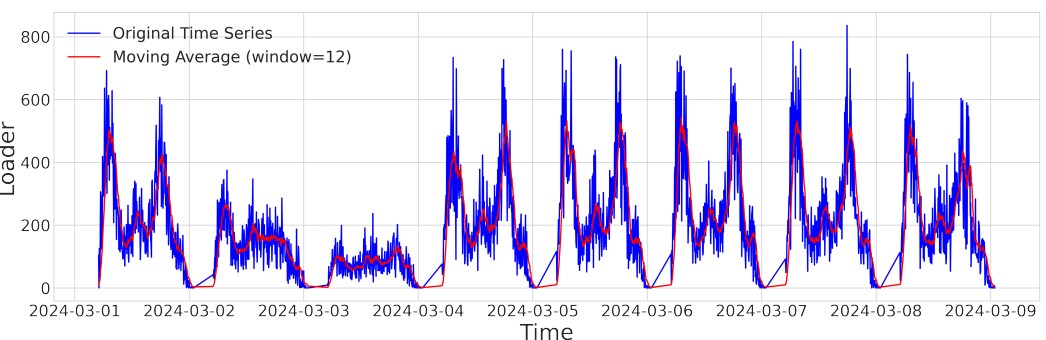

Figure 5: In blue, the time series containing loading information in a bus station, collected every 5 minutes between March 1st, 2024, and March 9th, 2024. In red, we show the time series transformed by SMA using a window size of 12 observations.

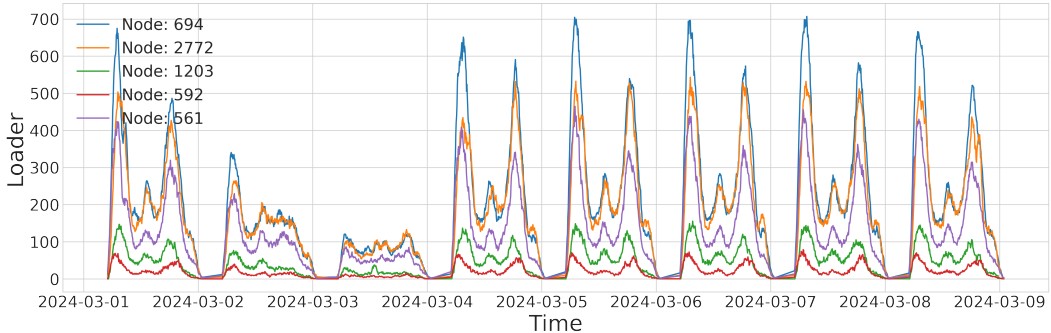

Figure 6: Five time series with intense transit of passengers to illustrate the node regression task.

Similarly, to illustrate the relationship between different stations and the importance of considering the graph structure, from all 2,871 possible nodes, we have selected the top 5 stations with the highest transit of passengers. As shown in Figure 6, all respective time series were smoothed before being modeled by the following models.

These time series were extensively analyzed in Section 5 (Benchmarking Results), where several models were applied to predict passenger loads across the five stations. The models used include a classical approach (SARIMA), DNN-based models (LSTM and GRU), transformer-based models (CHRONOS), and GNN-based models (remaining results). Details about these approach and models are dicussed in Appendix C.

Instead of showing the prediction results produced by CHRONOS alongside other models in Figure 4, we chose to individually showcase its performance in Figure 7. Analyzing this result reveals an impressive performance, particularly given that no specific training was conducted to model the time series observations from these selected stations. In summary, CHRONOS leverages advancements in language model architectures to achieve zero-shot performance in time series forecasting.

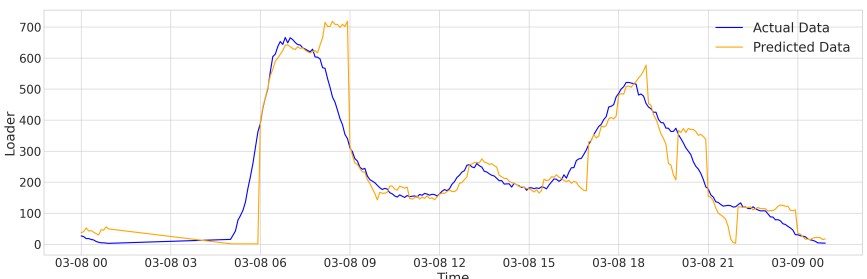

Figure 7: Observations predicted by CHRONOS.

Still focused on learning from the temporal relationship between multiple transportation modes (BRT, Subway, and Bus), we have selected a given station where all three vehicles intersect to serve the local population. Figure 8 illustrates the resulting time series of passenger loads over six consecutive days.

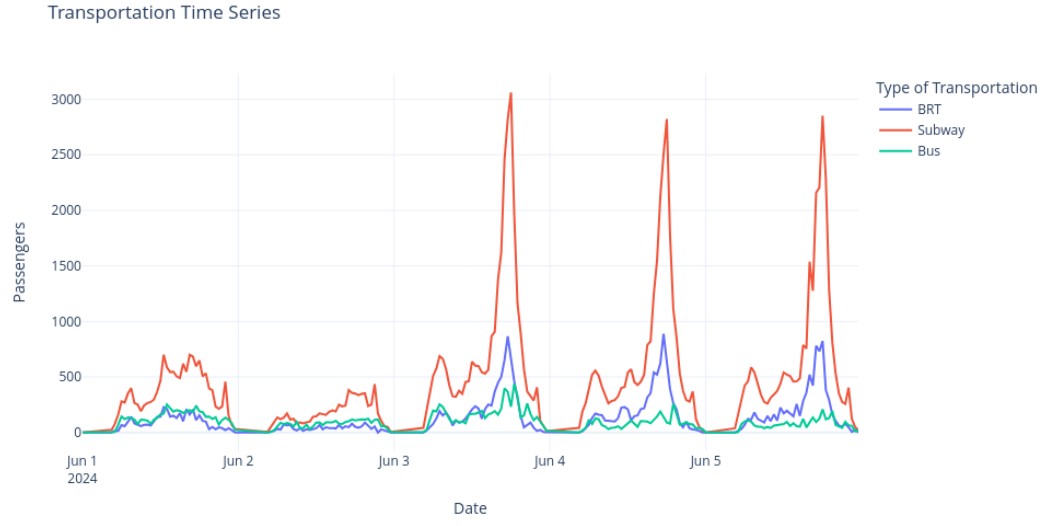

Figure 8: Time series representing multiple transportation modes.

In the subsequent experimental evaluation, we employed a recently published approach called SOFTS [Han et al., 2024]. This method introduces a novel module that uses a centralized strategy to

enhance efficiency while reducing dependence on the quality of individual time series. Specifically, it aggregates all time series into a global core representation, which is then distributed and fused with individual series representations to facilitate effective channel interactions.

In our experiments, we configured SOFTS with a prediction horizon of 12 and trained for 100 epochs. The learning rate was set to 0.0003. The model architectures consisted of an initial model with 256 units, a core representation with 64 units, and a final model with 256 units.

The main goal is to leverage the behaviors of each time series, assuming potential interactions between passengers switching between different transportation modes. Figure 9 illustrates how the predictions initially diverge from the expected number of passengers using conventional buses. However, the final predicted values gradually converge toward the expected ones. This result emphasizes the importance of incorporating multiple transportation modes to better understand population behavior and highlights the potential for further investigation of this approach in long-term predictions.

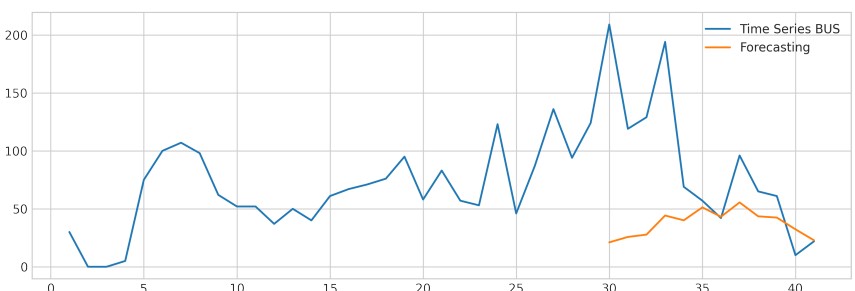

Figure 9: .

## A.2 Concept Drift

Passenger's pattern behavior changes can be predicted via Concept Drift (CD) [Gama et al., 2014, Bifet & Gavalda, 2007, Koh et al., 2024]. This technique models unforeseen changes in prediction variables or relationships among input features over time to ensure machine learning models' accuracy and robustness in prediction. These changes can arise from various factors such as human behavior, environment, or adaptations in the system being modeled. Concept Drift is particularly suitable for dynamic environments, such as transportation systems. For instance, social or extraordinary events can impact urban traffic and consequently influence passenger behavior during specific periods of the day.

In Figure 10, we illustrate using black-dashed lines the usage of a CD method to detect when the system changes its behavior (blue line) in relation to the prediction performances (red and yellow lines). The CD method considered in this experiment was the Page-Hinkley test [Page, 1954].

Another concept detection example is illustrated in Figure 11. In this case, rather than using CD to monitor the learning performance of classification algorithm, we have directly assessed the passenger load (referred to as "Amplitude") between March, 2024 and June, 2024. To illustrate the benefits of using such detectors, we selected a bus stop located near the Federal University of Bahia, the largest university in Salvador. In the figure below, the red vertical dashed lines indicate the moments when concept drifts were detected. The first drift corresponds to the start of the academic semester, marked by an increase in student use of public transportation. The second drift captures the onset of a strike, which disrupted classes and led to a decrease in passenger numbers.

The drifts detected in this experiment were automatically identified using the Fourier Transform (FT), as described in Algorithm 1. The first three lines perform a spectral analysis, extracting components and frequencies from the signal. In Lines 6–9, a window-based strategy is applied to compute amplitudes, enabling the identification of changes. An important step in this algorithm is Line 12, where differences between pairs of amplitudes are calculated. Finally, in the remaining lines, observations are filtered based on whether the changes exceed a specified threshold.

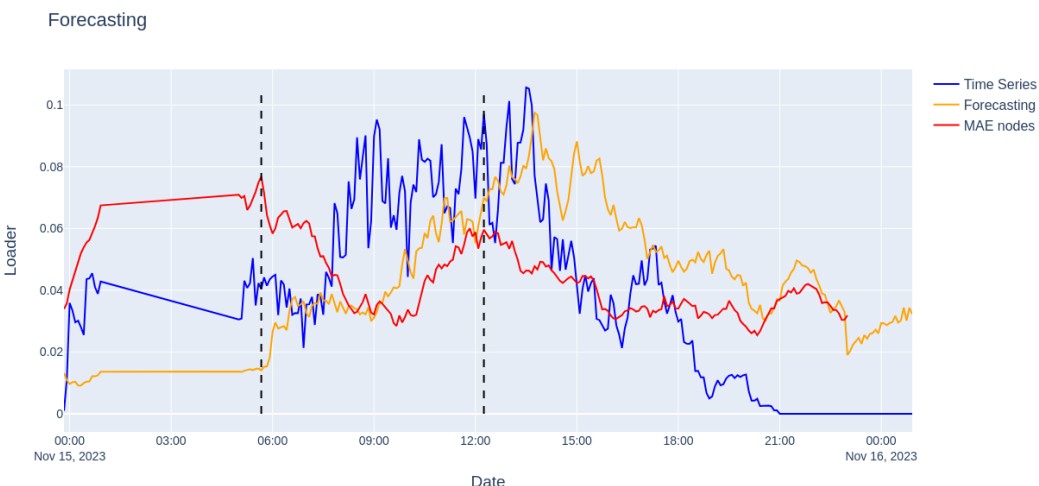

Figure 10: Example of using Concept Drift to detect when learning models is out of date.

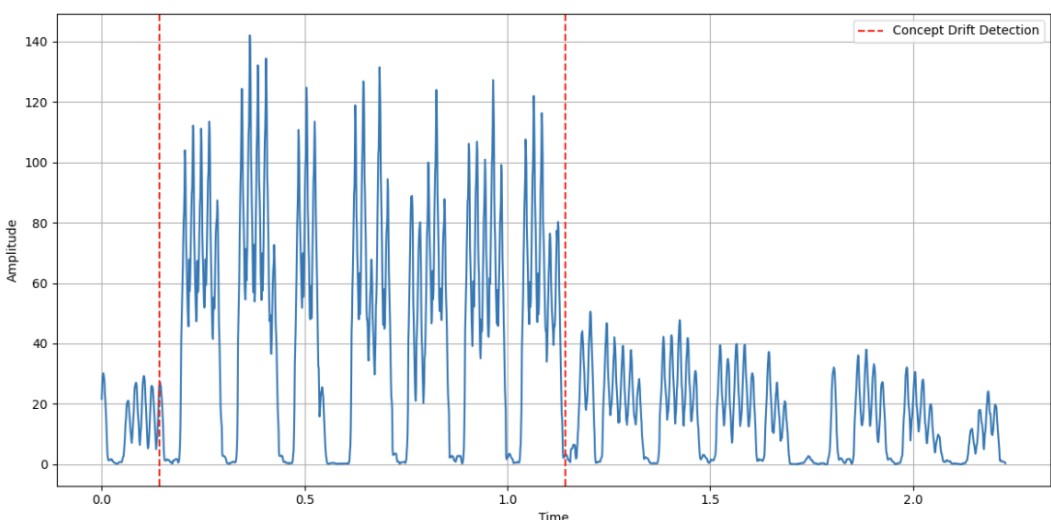

Figure 11: Example of using Concept Drift to detect when the passengers' behavior changes.

### A.3 OUTLIER DETECTION

Another important task in managing public transportation is the identification of outliers. There is a conceptual distinction between CD and outlier detection [Gama et al., 2014]. CD refers to an actual change in behavior, where a monitored system transitions from one state to another. In previous example, when the strike began, student behavior changed, leading to a decrease in the number of passengers using public transportation, signaling a shift in the system. In contrast, outliers do not signify a change in system behavior; they simply represent abnormal events that temporarily disturb the system.

To illustrate outliers in our scenario, consider Figure 12, which depicts the accumulated time buses took to travel between Stops A and B. It is evident that the expected time follows a consistent pattern. However, on April 4, 2024, heavy rain caused a tree to fall, blocking a major avenue connecting these stops. Using the Isolation Forest algorithm, we successfully identified this outlier (represented by the red dots). This example underscores the importance of outlier detection for automatically identifying abnormal events, thereby supporting policymakers in making more informed decisions.

---

**Algorithm 1:** Detecting drifts using Fourier Transform

---

**Input:** signal (time series), frequency (Hz), sampling_rate (Hz), threshold (%)
**Output:** indices_cd

1
2 $f \leftarrow$ FT(signal)
3 frequencies $\leftarrow$ FT_freq($|$signal$|, d = 1/$sampling_rate)
4 spectrum_magnitude $\leftarrow |$f$|$
5
6 period $\leftarrow 1/$frequency
7 ws $\leftarrow$ period $\times$ sampling_rate
8 amplitudes $\leftarrow [\max($signal$[i : i + $ws$]) - \min(signal[i : i + $ws$])$
9 $\qquad$ for $i$ in range$(0, |$signal$| - $ws$ + 1, $ws$)]$
10
11
12 differences $\leftarrow |$diff(amplitudes)$|$
13
14 max_amplitude $\leftarrow \max($amplitudes$[: -1], $amplitudes$[1 :])$
15 absolute_threshold $\leftarrow ($threshold$/100) \times $max_amplitude
16
17 indices_cd $\leftarrow$ where(differences $>$ absolute_threshold)$[0]$
18
19 **return** *indices_cd*

---

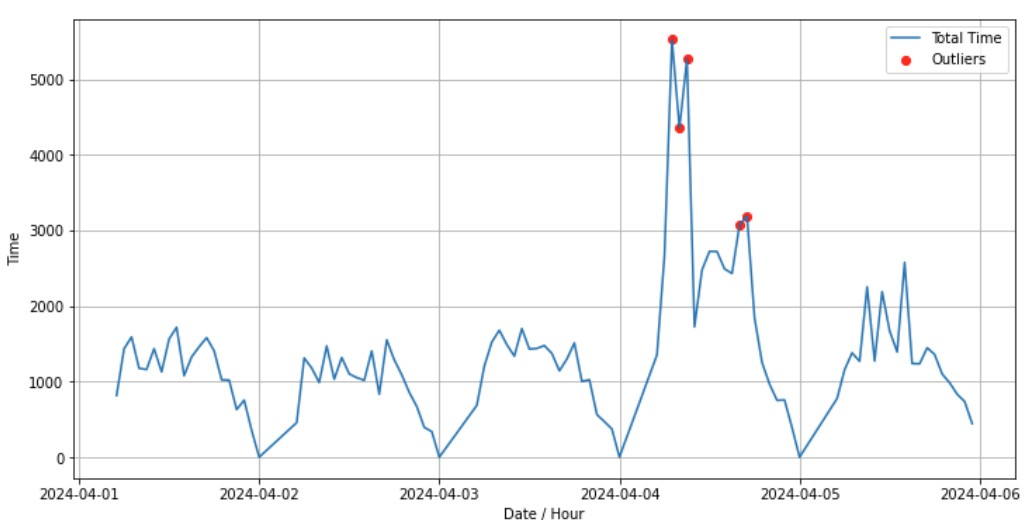

Figure 12: Using Isolation Forest to detect outliers in public transportation.

## A.4 PUBLIC TRANSPORTATION PLANNING

All experiments presented thus far have focused on evaluating learning methods. In this section, we shift our focus to applications commonly used in public transportation management that do not necessarily require training a model.

In our first evaluation, we analyzed the Origin-Destination (OD) matrix to determine the maximum passenger load during different time intervals. According to [Ceder, 2016], one of the fundamental objectives of transit service provision is to ensure sufficient capacity to accommodate the maximum number of passengers on board along the entire route within a given time period. Let us denote this time period (typically one hour) as $j$. Based on the peak-load factor concept, the required number of vehicles for period $j$ is given by:

$$\mathcal{M}_j = \frac{\bar{P}_{mj}}{\gamma_j c} \tag{1}$$

In this equation, $\bar{P}_{mj}$ is the average maximum number of passengers (max load) observed on-board in period $j$, $c$ enotes the vehicle's capacity (the total number of seats plus the maximum allowable standees), and $\gamma_j$ is the load factor for period $j$, where $0 \leq \gamma_j \leq 1.0$.

To illustrate the importance of calculating the max load $\mathcal{M}_j$, we have selected a specific line and analyzed the max-load stops in Figure 13 during four different time intervals: (i) rush time at 7 a.m., and 4 p.m., (A) and (B), respectively; and (ii) off-peak time at 10 a.m., and 3 p.m., (C) and (D), respectively. By analyzing this maps, one might be able to understand how the maximum load stops are varying within different time intervals. This information facilitates more effective bus planning and distribution, enhancing service delivery to better meet the population's needs.

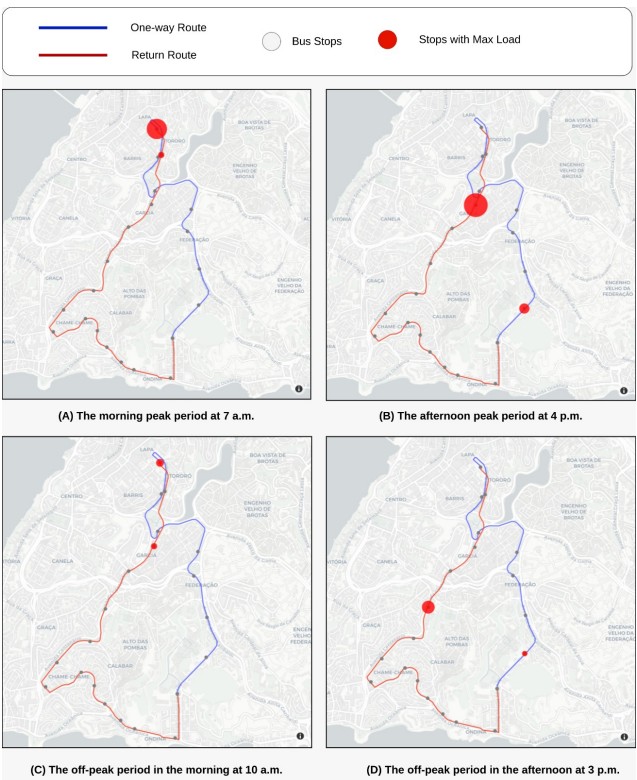

Figure 13: Max load calculated for a specific line during different relevant time intervals.

In our second example presented in Figure 14, we illustrate how the information about max load can be used in practice to plan timetables, specifying which buses must be set as Express and Normal. Typically, when passenger data at stops/stations and along routes is unavailable, it becomes difficult to optimize bus services effectively. For example, during a rush hour interval (e.g., 8 AM–9 AM), 9 buses might be scheduled to serve passengers traveling from stop A to stop B, passing through intermediate stops. Without an estimate of maximum passenger load, all buses would need to return from stop B to stop A via the same route, stopping at all intermediate stations. Such a strategy has some problems: it wastes time and fuel, besides delaying the arriving time at A. Considering A is a neighborhood and B downtown, the amount of passengers from B to A is considerably lower during this rush time. Knowing the optimal number of buses required for the return trip and their appropriate schedules can significantly mitigate these issues.

In Figure 14, we illustrate three strategies for determining the bus type: (i) randomly selecting buses; (ii) dividing the hour into intervals based on the expected number of normal buses and designating the next bus within each interval as normal; and (iii) assigning normal buses based on the nearest

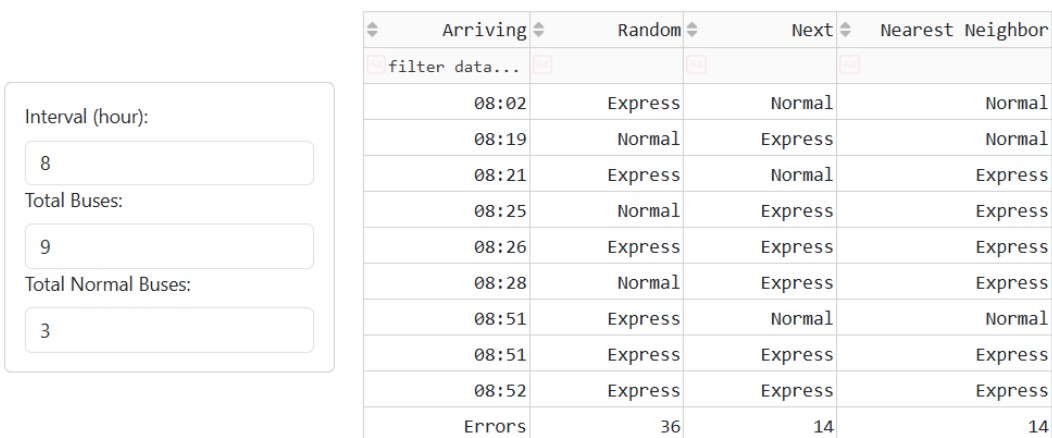

Figure 14: Planning timetables after calculating max load.

neighbor approach, aiming to minimize the interval between them. The error is calculated as the absolute difference between the actual intervals between buses and the expected intervals. With this information, policymakers can effectively reduce passengers' waiting times at stops and stations while optimizing the management of bus transit within the city.

A.5    INTEGRATION WITH OTHER URBAN DATASETS

To demonstrate the feasibility of integrating the proposed dataset with others in the urban scenario, we consider an existing dataset for public schools in Salvador made available by the Municipal Department of Education at `https://dados.salvador.ba.gov.br/search?tags= educacao` (in Portuguese). This dataset stores the public school's name, latitude, longitude, neighborhood, complete address, and administration data.

Integrating the school's and SUNT datasets involves computing each school's closest bus stop or station. Furthermore, passengers in the SUNT dataset can be categorized as students by their transportation card data. This integration leads to a wide range of new applications related to the proposed dataset. For example, one may be interested in investigating the student passenger loading at bus stops or stations near the schools on a given day and time, as shown in Figure 15. This example shows that the applicability of the proposed dataset transcends the traffic domain.

A.6    EXPLORING FUTURE WORK

Multi-objective optimization problems aim at simultaneously optimizing two or more conflicting objective functions. It means that improving one objective implies worsening another. Several characteristics add to the difficulty of multi-objective problems. Firstly, these problems entail finding a set of optimal solutions, i.e., those that yield the best compromise among objectives. Secondly, the distribution of the efficient points in the objective space may make the intensification and diversification balance difficult. Thirdly, the difficulty increases with the number of objectives.

Several graph-based multi-objective optimization problems have been investigated over the last decades, including spanning trees, shortest path, and maximum flow. The edges are assigned to two weights: the distance and the travel time between adjacent vertices, which may be anticorrelated values. The distance is geospatial information from the dataset, and the travel time is temporal information computed from the traffic jam. The multi-objective shortest path problem consists of finding the set of optimal paths from S to T (starting and target points) by simultaneously optimizing both the total distance and travel time.

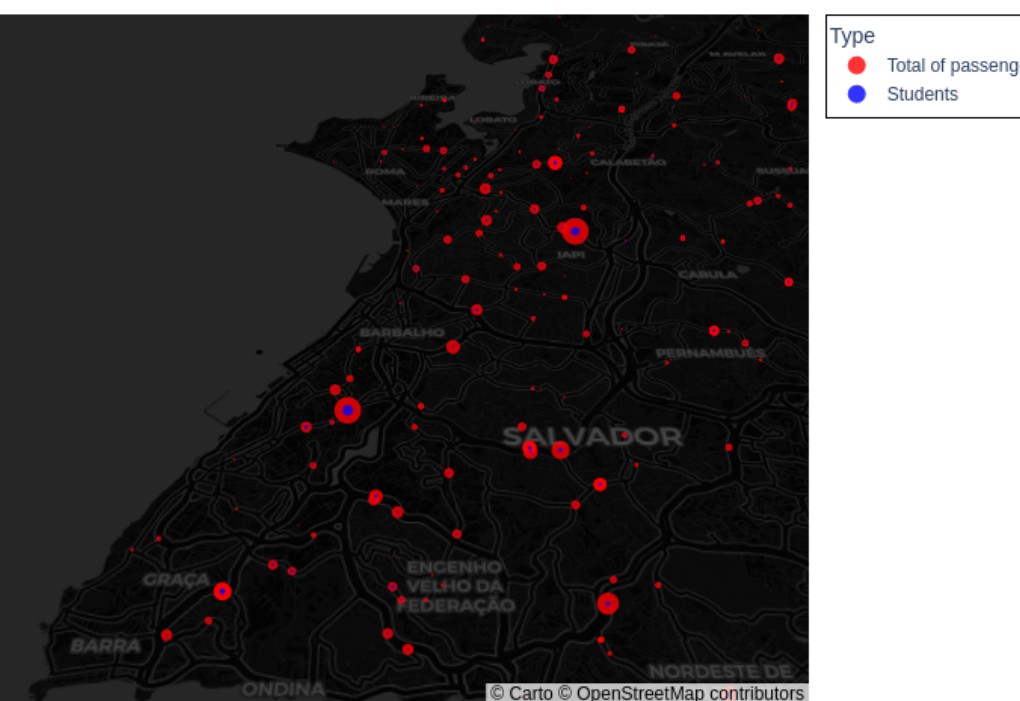

Figure 15: Integrating SUNT with data from public schools. The total number of passenger per stop/station is shown in red. Students are represented by blue dots.

Machine learning techniques have been successfully employed to enhance the search process of metaheuristic algorithms. Common examples in the literature include multi-agent approaches, which involve a set of intelligent and autonomous entities (agents) collaborating to solve a problem harmoniously. Each agent employs one or more appropriate methods to solve the problem, learns from its experiences, and cooperates by exchanging information with other agents. An example is particle swarm optimization algorithms, where each particle is modeled as an agent; another example is evolutionary algorithms based on multiple search operators, where each operator is an intelligent agent. These examples may maintain a GNN model to select in each algorithm iteration an agent to execute.

## B  DATASET GENERAL INFORMATION

Efficient urban mobility requires a branch of strategies to manage traffic, delivering improvements, such as increasing safety, reducing travel time, decreasing costs, and supporting environmental issues.

In this work, we focus our investigation on efficient urban mobility by modeling data from public transportation systems due to its importance to the population. Any decision regarding this system directly impacts urban mobility, especially in developing countries, where it is often the only means of transport available to low-income populations. When poorly planned, it delivers low-quality services with delayed and overloaded vehicles, concentrates traffic in specific regions while leaving others unattended, and aggravates pollution with higher gas emission rates.

After an in-depth investigation of published manuscripts focused on public transportation, we noticed a limitation in the availability of a totally public dataset containing comprehensive quantitative, spatial, and temporal information about passengers, vehicles, lines, stops, and stations.

Moreover, despite the increasing advancements in GNN methodologies, particularly for intelligent transportation systems (ITS), there remains a significant lack of datasets with detailed information about public transportation with their respective passengers. As highlighted in Table 2, many existing

graph datasets are completely represented, often missing essential spatiotemporal features, integration of multifaceted data, and sufficient scale to address the complexities of real-world challenges.

The missing information ("-") in this table reflects the fact that several datasets commonly used in research articles are partially described in the publications and are not freely shared in public repositories with the same level of detail as ours. For example, we have noticed that information about the number of nodes, edges, or specific temporal intervals is often unavailable. As a result, researchers face challenges in reproducing experiments or fully understanding the scope and limitations of the datasets referenced in these studies. On the other hand, we offer the Salvador Urban Network Transportation (SUNT) dataset, which stands out as an exception, offering 2,871 nodes, 4,526 edges, and a temporal granularity of less than one minute, with an in-depth dataset construction, which are pivotal for addressing key deficiencies identified in recent studies on learning benchmarks. SUNT offers a robust foundation for developing models that can learn complex spatiotemporal patterns and adapt to rapidly changing conditions in ITS scenarios. Additionally, being recently collected, it reflects an updated urban configuration, in contrast to the most recent previously available dataset, which dates back to 2019.

| Dataset | #Nodes | #Edges | Period | Shortest time interval |
|---|---|---|---|---|
| METR-LA | 207 | 2,369 | March 1, 2012 to June 30, 2012 | 5 minutes |
| PeMS-BAY | 325 | 1,515 | January 1, 2017 to May 31, 2017 | 5 minutes |
| TaxiBJ | – | – | July 1, 2013 to October 30, 2013
March 1, 2014 to June 30, 2014
March 1, 2015 to June 30, 2015
November 1, 2015 to April 10, 2016 | 30 minutes |
| BikeNYC | 50 | – | April 1, 2014 to September 30, 2014 | 1 hour |
| Shanghai Metro | 288 | 958 | July 1, 2016 to September 30, 2016 | 15 minutes |
| Hangzhou Metro | 80 | 248 | January 1, 2019 to January 31, 2019 | 15 minutes |
| Beijing Metro | 276 | – | February 29, 2016 to April 3, 2016 | – |
| Chongqing Metro | 170 | – | March 1, 2019 to March 31, 2019 | 15 minutes |
| Stockholm County | – | – | – | – |
| UVDS | 104 | – | Three months | 5 minutes |
| **SUNT** | **2,871** | **4,526** | **March 1, 2024 to October 30, 2024** | **< 1 minute** |

Table 2: Characteristics of SUNT Compared to Common Graph Datasets

In this paper, we make SUNT available, a massive dataset organized into two parts: (i) a raw set of data that the reader can process according to their needs, and (ii) a graph connecting all these data, respecting their geospatial and temporal restrictions. In addition, we also shared adjusted models as benchmarks since the classical time series analyses to Graph Neural Networks to perform classification and regression tasks on vertices and edges. Benchmarking public transportation datasets forms the foundation for developing effective, efficient, and sustainable transportation systems that meet the evolving needs of society.

All steps performed to create SUNT, illustrated in Figure 1, were presented in Section 3 (SUNT Dataset Construction) of our manuscript. In the following sections, we provide more details about all raw and graph-based datasets.

## B.1 AUTOMATIC VEHICLE LOCATION (AVL)

AVL (Automatic Vehicle Location) technology records real-time vehicles' geographical location. The Global Positioning System (GPS) is commonly used to this end [Riter & McCoy, 1977]. The collected data are parameters to estimate several other information, such as passengers boarding and alighting, public transportation network planning, monitoring and controlling traffic operations, and air quality improvements, among other benefits [Hickman, 2004].

In the SUNT dataset, AVL records can be divided into two datasets: AVL-lines and AVL-vehicles. AVL-lines comprise static information regarding bus lines, whose features are shown in Table 3. Column `route_short_name` identifies the bus line. Column `pt_sequence` presents the stop sequences of the bus line. Column `direction_id` gives the direction of the line, where 1 stands for one-way and 0 for return trip. Columns `longitude` and `latitude` give the geographical

coordinate of the bus stop identified in column `stop_id`. Column `route_long_name` names the bus stop. Column `service_code` identifies the trip along the line.

Table 3: AVL-lines features: information regarding bus lines.

| route_short_name | pt_sequence | direction_id | longitude | latitude | stop_id | route_long_name | service_code |
|---|---|---|---|---|---|---|---|
| 0116 | 1 | 1 | -38.51123 | -12.983389 | 43768720 | Avenida Vale Do Tororo | 53786 |
| 0116 | 2 | 1 | -38.511097 | -12.986428 | 45832898 | Avenida Vale do Tororo, 291 | 53786 |
| 0116 | 3 | 1 | -38.511448 | -12.990091 | 44782328 | Praça Dr. João Mangabeira | 53786 |
| 0116 | 4 | 1 | -38.504387 | -12.990533 | 44784448 | Av. Vaco da Gama, S/N - | 5378 |
| 0116 | 5 | 1 | -38.501972 | -12.992005 | 44784449 | Av. Vasco da Gama, 271 - | 53786 |
| 0116 | 6 | 1 | -38.499004 | -12.993324 | 45833116 | Av. Vasco da Gama, S/N - | 53786 |

Table 4 presents AVL-vehicles features, which comprise information concerning vehicles' routes and bus schedules. Column `gps_datetime` gives the date and time of the bus arriving at the stop identified in column `stop_id`. If column `gps_datetime` records two values, then the lowest is the bus arrival time at the stop, and the greatest is the bus departure time. The stop sequence of the bus line must be consistent with respect to the data in `gps_datetime`, i.e., for each step, the arrival time is less than the departure time, and this latter is less than the arrival time for the next stop. The remaining columns of Table 4 are similar to those described in Table 3.

Table 4: AVL-vehicle features: information concerning vehicles' routes and bus schedules.

| vehicle | route_short_name | direction_id | gps_datetime | longitude | latitude | stop_id | service code |
|---|---|---|---|---|---|---|---|
| 20001 | 0310 | 0 | 2024-03-01 05:53:20 | -38.512428 | -12.978642 | 45834426 | 45546 |
| 20001 | 0310 | 0 | 2024-03-01 05:53:53 | -38.509964 | -12.975935 | 45834425 | 45546 |
| 20001 | 0310 | 0 | 2024-03-01 05:53:57 | -38.509964 | -12.975935 | 45834425 | 45546 |
| 20001 | 0310 | 0 | 2024-03-01 05:54:02 | -38.508957 | -12.975689 | 44782954 | 45546 |
| 20001 | 0310 | 0 | 2024-03-01 05:54:47 | -38.508957 | -12.975689 | 44782954 | 45546 |
| 20001 | 0310 | 0 | 2024-03-01 05:55:58 | -38.507446 | -12.97867 | 44428471 | 45546 |

For each day, the SUNT dataset comprises 61 files of AVL-lines and AVL-vehicle from three bus companies. The size comprises, on daily average, 2.5 million entries in AVL-vehicle and 200 thousand entries in AVL-lines. Each company dataset contains their respective data concerning bus lines and vehicles.

## B.2 AUTOMATIC FARE COLLECTION (AFC)

Automated Fare Collection (AFC) is a fare payment system through smart cards or smartphones [Ampelas, 2001]. This process can occur either at the boarding or alighting, depending on the type of AFC system the vehicle operator implements. In addition, AFC also collects personal information and, in some scenarios, boarding and/or alighting locations [Mohammed & Oke, 2023].

Salvador's public transportation system collects data from buses, BRT, and subway passengers via AFC system. Concerning the buses, the collection occurs at two moments: when the passenger validates the ticket at the vehicle's turnstile or a mobile turnstile. AFC collects subway data from a turnstile installed at the station entrance. AFC system in BRT combines the collection methods used in the buses and subway.

Despite AFC operation in all transport systems, some limitations exist in collecting crucial data for planning and monitoring public transport. In the case of buses, AFC does not record location information and boarding and alighting times at bus stops; it only records the vehicle, line, and time of card registration at the turnstile.

A subsample of AFC, shown in Table 5, illustrates the used attributes: `cod_card` is the number of passenger's card, randomly generated to avoid recovering any user identification, `afc_datetime` represents the time when the passenger registers the payment, `integration` indicates the possibility of a connection between vehicles, `route_short_name` is the route identification, `direction_id` shows the bus direction (I – one way or V – return) considering its initial and final stops, `value` is the trip cost, and `vehicle` is the code used to identify the vehicle. The AFC dataset contains the following total of passenger trips: March – $36,851,307$, April – $38,238,530$, May – $39,424,894$, June – $33,680,595$, and July – $33,549,584$.

Table 5: AFC subsample.

| cod_card | afc_datetime | integration | route_short_name | direction_id | value | vehicle |
|---|---|---|---|---|---|---|
| 02310034266847 | 2024-03-01 06:22:03 | False | 1386 | I | 0.0 | 20390 |
| 02310034266847 | 2024-03-01 06:22:10 | False | 1386 | I | 0.0 | 20390 |
| 02310033002113 | 2024-03-01 06:22:57 | False | 1386 | I | 0.0 | 20390 |
| 02310032345960 | 2024-03-01 08:12:25 | False | 1386 | I | 0.0 | 20390 |
| 02320033736512 | 2024-03-01 06:04:08 | False | 1386 | I | 0.0 | 20390 |
| 03620033306428 | 2024-03-01 06:10:17 | False | 1386 | I | 5.2 | 20390 |

## B.3 GENERAL TRANSIT FEED SPECIFICATION (GTFS)

General Transit Feed Specification (GTFS) is an open standard for distributing relevant information about transit systems to users.

In the SUNT dataset, the GTFS provides 5 files (GTFS Agency, GTFS Routes, GTFS Trips, GTFS Stops Times, and GTFS Stops) that describe the entire network and services of public transportation related to bus companies.

GTFS Agency, present in Table 6, contains information about the bus companies, which are associated with GTFS Routes (Table 7) by the attribute agency_id. GTFS Routes contains information about bus lines and is associated with GTFS Trips (Table 8) by the attribute route_id. GTFS Trips shows all the trips and the paths followed by the bus and is directly associated with GTFS Stops Times (Table 9), which maps the chronological order of bus stops where each trip paused. Finally, GTFS Stops (Table 10) contains information about each bus stop and is associated with the GTFS Stops Times by the attribute stop_id.

Table 6: GTFS Agency: information about the bus companies.

| agency_id | agency_name | agency_url | agency_timezone | agency_lang | agency_phone |
|---|---|---|---|---|---|
| 1 | company_I | www. | America/Sao_Paulo | pt | |
| 2 | company_II | www. | America/Sao_Paulo | pt | |

Table 7: GTFS Routes: information about bus lines.

| route_id | agency_id | route_short_name | route_long_name | route_type |
|---|---|---|---|---|
| 4089 | 1 | 1230 | Sussuarana x Barra R1. | 3 |
| 4450 | 1 | 1321 | São Marcos x Barroquinha | 3 |
| 4518 | 1 | 1103 | Alto do Cruzeiro/Pernambués x Shop.Bela Vista/Term Ac.Norte | 3 |
| 4523 | 1 | 1405 | Estação Pirajá x Cajazeiras 8 | 3 |
| 4524 | 1 | 1137 | Pernambués x Barra | 3 |

Table 8: GTFS Trips: information about the trips and the paths followed by the bus.

| route_id | service_id | trip_id | direction_id | block_id | shape_id |
|---|---|---|---|---|---|
| 4089 | 26082_D_1046761 | 1046761_D_1_0 | 0 | 4089_001M | 26082_I |
| 4089 | 26082_D_1046761 | 1046761_D_1_1 | 1 | 4089_001M | 26082_V |
| 4089 | 26082_D_1046761 | 1046761_D_2_0 | 0 | 4089_002M | 26082_I |
| 4089 | 26082_D_1046761 | 1046761_D_2_1 | 1 | 4089_002M | 26082_V |
| 4089 | 26082_D_1046761 | 1046761_D_3_0 | 0 | 4089_002T | 26082_I |

The GTFS used is static and undergoes changes only when necessary, such as alterations to trips, routes, directions, or bus stops. Accordingly, the routes file has 412 registers, trips file has 51,615 registers, stops file has 2,975 registers, and stop times file has 1,679,961 registers.

Table 9: GTFS Stops Times: the chronological order of bus stops where each trip paused.

| trip_id | arrival_time | departure_time | stop_id | stop_sequence | pickup_type | drop_off_type |
|---------|--------------|----------------|---------|---------------|-------------|---------------|
| 1046761_D_1_0 | 08:30:00 | 08:30:00 | 43968810 | 1 | 0 | 0 |
| 1046761_D_1_0 | 08:31:41 | 08:31:41 | 47566106 | 2 | 0 | 0 |
| 1046761_D_1_0 | 08:33:49 | 08:33:49 | 44782337 | 3 | 0 | 0 |
| 1046761_D_1_0 | 08:34:55 | 08:34:55 | 44784470 | 4 | 0 | 0 |
| 1046761_D_1_0 | 08:35:44 | 08:35:44 | 44784471 | 5 | 0 | 0 |

Table 10: GTFS Stops: information about each bus stop.

| stop_id | stop_name | latitude | longitude | location_type | parent_station |
|---------|-----------|----------|-----------|---------------|----------------|
| 43968810_S | R. São Cristóvão 2 | -12.931565284729 | -38.444393157959 | 1 | |
| 43968810 | R. São Cristóvão 2 | -12.931565284729 | -38.444393157959 | 0 | 43968810_S |
| 47566106_S | Av. Ulysses Guimarães 4067 | -12.93385887146 | -38.4467735290527 | 1 | |
| 47566106 | Av. Ulysses Guimarães 4067 | -12.93385887146 | -38.4467735290527 | 0 | 47566106_S |
| 44782337 | Av. Ulysses Guimarães 4314-4322 | -12.9351501464844 | -38.4405784606934 | 0 | |

## B.4 LOCAL TRIP INFORMATION (LTI)

The bus company provides the Local Trip Information (LTI) dataset and maps the start of each trip made by a vehicle on a specific route. Table 11 presents the attributes of the trip mapping dataset, with the attributes start_trip and end_trip being the most important as they describe the start and end of each trip, and the attribute activity classifies the type of trip as either normal, leaving the garage, or returning to the garage. The LTI dataset complements the AVL dataset as it maps trip records, since the AVL does not contain this information. In relation to the total of instances per month, LTI comprises the following amounts: March – 771, 492, April – 773, 267, May – 771, 492, June – 717, 285, and July – 681, 320.

March and April, LTI contains a total of and instances, respectively.

Table 11: LTI: the start of each trip made by a vehicle on a specific route.

| route_short_name | service_code | direction_id | vehicle | start_trip | end_trip | activity |
|------------------|--------------|--------------|---------|------------|----------|----------|
| T014 | 74335 | I | 20401 | 01/03/2024 17:03:49 | 01/03/2024 17:10:45 | Leaving the garage |
| T014 | 74335 | I | 20516 | 01/03/2024 05:37:16 | 01/03/2024 05:40:36 | Leaving the garage |
| T014 | 74335 | I | 20516 | 01/03/2024 17:11:40 | 01/03/2024 17:20:58 | Normal |
| T014 | 74335 | I | 20086 | 01/03/2024 05:39:27 | 01/03/2024 05:46:38 | Leaving the garage |
| T014 | 74335 | I | 20401 | 01/03/2024 12:37:47 | 01/03/2024 12:42:04 | Returning to the garage |

## B.5 BOARDING DATA

The boarding data, illustrated in Table 13, contains a set of raw details created after integrating the AFC, LTI, AVL, and GTFS data, as shown in Figure 1. As one may notice, most attributes presented in this table are inherited from this integration. A very important attributed available in this dataset is the bus type. Although we are only using regular buses in our experiments, we emphasize that readers can work on different types of public transportation such as BRT and subway.

To avoid extending the paper length and affecting the readability of our manuscript, we have only included this table to illustrate the dataset and provide an idea of the various investigative possibilities. However, the data dictionary, containing detailed information about each feature, is available in the GitHub repository, where we share all data and sources.

## B.6 ALIGHTING DATA

The alighting data, illustrated in Table 13, is one of the most important dataset produced by our research. As previously mentioned, in the local scenario, users do not need to register when they leave the vehicles. After a series of inferences, we estimate users' alighting containing a set of relevant features to understand their behavior better. As mentioned in previous dataset, to avoid extending the paper length and affecting the readability of our manuscript, we have only included this table to

Table 12: Description of Columns in the Dataset Boarding

| Column | Sample Value | Dtype |
|---|---|---|
| tripuserid | 02300033357538_20240301184830 | object |
| type_bus | bus | object |
| user_type | driver | object |
| set | company_i | object |
| registers | 2 | int64 |
| trip_id | 20097_0310_7 | object |
| start_trip | 2024-03-01 17:56:43 | datetime64[ns] |
| end_trip | 2024-03-01 20:08:27 | datetime64[ns] |
| tolerance | NaT | datetime64[ns] |
| integration | False | bool |
| cod_card | 2300033357538 | object |
| stop_time | 2024-03-01 19:36:35 | datetime64[ns] |
| register_time | 2024-03-01 18:48:30 | datetime64[ns] |
| service_code | 45546 | object |
| route_short_name | 0310 | object |
| vehicle_afc | 20097 | object |
| vehicle | 20097 | object |
| stop_id | 44782849 | object |
| order | 1 | float64 |
| direction_id | I | object |
| trip_em | 7.0 | float64 |
| dif_boarding | 48.083 | float64 |
| trip | Inside | object |
| classification | irregular | object |
| motive | excessive time | object |
| trip | firt_trip | object |
| set_nb | company_i | object |
| stop_time_nb | 2024-03-01 20:04:39 | datetime64[ns] |
| route_short_name_nb | 1067 | object |
| vehicle_nb | 20446 | object |
| stop_id_nb | 44164980 | object |
| diff_nb | 0.53 | float64 |
| motive_pe | regular | object |
| target_boarding | irregular | object |

illustrate the dataset and provide an idea of the various investigative possibilities. However, the data dictionary, containing detailed information about each feature, is available in the GitHub repository, where we share all data and sources.

### B.7 ORIGIN-DESTINATION (OD) DATASET

The last dataset presented in Figure 1 is the Origin-Destination (OD) Dataset. In summary, we combining the previous dataset, we estimated important information as users' full trips. Consequently, we can estimate the bus loading in nodes (stops/stations) and edges (streets/avenues). The features available in our OD dataset is shown in Table 14. Similarly to previous data, the data dictionary is presented in the GitHub repository.

### B.8 GRAPH-BASED DATASET

Our final contribution to data sharing is the organization of SUNT as a temporal graph. We have integrated data collected over five months (from March to July 2024) into temporal graphs, with observations varying every 5 minutes. Although we have used this configuration, we stress that any temporal interval can be easily modified using our preprocessing codes.

Table 13: Dataset summary Alighting

| Column | Sample Value | Dtype |
|---|---|---|
| tripuserid | 02300033520791_20240301104958 | object |
| stop_time_ali | 2024-03-01 10:55:27 | datetime64[ns] |
| stop_id_ali | 44165441 | object |
| order_ali | 6.0 | float64 |
| walk_target | excessive | object |
| trip_ali | 8.0 | float64 |
| walk_dis | 1.299 | float64 |
| walk_time | 15.588 | float64 |
| walk_speed | 5.5 | float64 |
| diff_de_pe | 68.4 | float64 |
| wait_time | 52.812 | float64 |
| trip_dis | 1.884 | float64 |
| trip_time | 5 | float64 |
| vel_media | 22 | float64 |
| bridge | False | bool |
| bridge_type | no bridge | object |
| bridge_id | None | object |
| chain | bus-bus | object |
| target_ws | regular | object |
| target_avs | regular | object |
| target_tt | regular | object |
| target_td | regular | object |
| target_alighting | regular | object |

Table 14: Summary of Dataset Columns OD

| Column | Sample Value | Dtype |
|---|---|---|
| route_short_name | 1521 | object |
| register_code | 55037 | int64 |
| direction_id | I | object |
| pt_sequence | 1 | int64 |
| stop_id | 46021891 | int64 |
| vehicle | 30661 | int64 |
| trip_number | 1 | int64 |
| trip_id | 30661_1521_1266 | object |
| start_trip | 2024-03-01 06:59:11 | datetime64[ns] |
| end_trip | 2024-03-01 07:15:22 | datetime64[ns] |
| stop_time | 2024-03-01 06:59:11 | datetime64[ns] |
| n-boardings | 42.0 | float64 |
| n-alighting | 0 | float64 |
| lag_loading | 0 | int64 |
| balance | 0 | int64 |
| loading | 42 | int64 |

In this dataset, we share edge information as illustrated in Table 16, in which `src` is the origin stop/station, whereas `dst` is the destination. The distance between them is shown in `distance`. Their geospatial locations are stored into `src_lat`, `dst_lat`, `src_lon`, and `dst_lon`. The average speed and time during the collected interval is shown in `average_speed`, and `trip_time`, respectively. The passenger loading in that edge (steet or avenue) is available in `loading`.

In Table 16, we share information about nodes, i.e., details related to stops and stations. Some relevant information containing average values considering vehicles are: `loading` – passenger loading that crossed a given node; `n-boarding` and `n-alighting` – amount of boarding and alighting; `n-routes`, `n-trips`, and `n-vehicles` contain the number of routes, trips, and vehicles; and

Table 15: Sample of edge features.

| src | dst | distance | src_lat | dst_lat | src_lon | dst_lon | average_speed | trip_time | loading |
|-----|-----|----------|---------|---------|---------|---------|---------------|-----------|---------|
| 100009577 | 345936831 | 0.254 | -12.902 | -12.902 | -38.42 | -38.417 | 25.6 | 4 | 78 |
| 100722777 | 100722778 | 0.362 | -12.899 | -12.897 | -38.408 | -38.408 | 11.3 | 8 | 20 |
| 100722777 | 44782645 | 1.062 | -12.899 | -12.899 | -38.408 | -38.413 | 40.2 | 5 | 45 |
| 100722777 | 45833440 | 0.417 | -12.899 | -12.897 | -38.408 | -38.409 | 50.5 | 10 | 90 |
| 100722777 | 66771046 | 0.934 | -12.899 | -12.897 | -38.408 | -38.413 | 26.2 | 6 | 30 |

`average_speed` is the average of speed for each vehicle during their last trip up to the destination node.

Table 16: Sample of node features.

| node | loading | n-alighting | n-routes | n-boarding | n-trips | n-vehicles | average_speed |
|------|---------|-------------|----------|------------|---------|------------|---------------|
| 100009577 | 2.77 | 0.0 | 1.08 | 0.23 | 1.1 | 1.1 | 6.31 |
| 100722777 | 28.54 | 4.43 | 1.54 | 4.49 | 1.56 | 1.56 | 22.86 |
| 100722778 | 36.72 | 1.39 | 1.83 | 0.1 | 2.04 | 2.04 | 16.06 |
| 101214305 | 12.53 | 3.97 | 1.0 | 1.66 | 1.0 | 1.0 | 19.95 |
| 101269104 | 125.57 | 3.55 | 4.57 | 9.48 | 5.28 | 5.28 | 38.25 |

In Table 17, we organized the node and edge information by considering the time interval. In our dataset, this interval was set as 5 minutes; however, using our codes, readers can modify it according to their needs.

Table 17: Sample Node Features with time.

| interval | node | loading | n-boarding | n-alighting | n-vehicles | n-routes | n-trips | average_speed |
|----------|------|---------|------------|-------------|------------|----------|---------|---------------|
| 2024-03-01 05:00:00 | 100009577 | 0.0 | 0.0 | 0.0 | 0.0 | 0.0 | 0.0 | 0.0 |
| 2024-03-01 05:05:00 | 100009577 | 14.0 | 1.0 | 0.0 | 2.0 | 2.0 | 2.0 | 0.0 |
| 2024-03-01 05:10:00 | 100009577 | 0.0 | 0.0 | 0.0 | 0.0 | 0.0 | 0.0 | 0.0 |
| 2024-03-01 05:15:00 | 100009577 | 0.0 | 0.0 | 0.0 | 0.0 | 0.0 | 0.0 | 0.0 |
| 2024-03-01 05:20:00 | 100009577 | 2.0 | 2.0 | 0.0 | 1.0 | 1.0 | 1.0 | 0.0 |
| 2024-03-01 05:25:00 | 100009577 | 18.0 | 0.0 | 0.0 | 1.0 | 1.0 | 1.0 | 0.0 |
| 2024-03-01 05:30:00 | 100009577 | 0.0 | 0.0 | 0.0 | 0.0 | 0.0 | 0.0 | 0.0 |
| 2024-03-01 05:35:00 | 100009577 | 0.0 | 0.0 | 0.0 | 0.0 | 0.0 | 0.0 | 0.0 |
| 2024-03-01 05:40:00 | 100009577 | 1.0 | 1.0 | 0.0 | 1.0 | 1.0 | 1.0 | 0.0 |
| 2024-03-01 05:45:00 | 100009577 | 0.0 | 0.0 | 0.0 | 0.0 | 0.0 | 0.0 | 0.0 |

### B.9 DATA STATISTICS

Aiming to understand our dataset better, we have used some basic statistics to describe the most relevant variables. While these statistics are not included in this manuscript for the sake of brevity, they are available in a well-organized Python notebook, which can be accessed at `https://github.com/suntdataset/sunt/blob/main/Statistics.ipynb`.

## C LEARNING MODELS

This section summarizes the benchmarks used in our experimental setup, providing details about the packages, parameters, and architectures considered to model the SUNT dataset.

### C.1 TIME SERIES ANALYSES

The modeling process using univariate time series was performed considering two well-known methods: Seasonal Auto-Regressive Integrated Moving Average (SARIMA), Gated Recurrent Units (GRU), and Long Short-Term Memory (LSTM). Although there are several methods designed to reach this objective, we have selected those as benchmarks for representing classical statistic-based and ML-based methods. In addition, recognizing the growing importance of foundation models in time series analysis, specifically those that can perform tasks without explicit training (zero-shot learning),

we also considered a language model adapted for time series forecasting, named CHRONOS. Finally, we used multivariate time series forecasting to make predictions on our dataset. This method is important for public transportation because it considers multiple factors at each moment, like different types of transportation (subway, BRT, and bus), to make more accurate predictions. For this, we used the Series-cOre Fused Time Series forecaster (SOFTS).

### C.1.1 SEASONAL AUTO-REGRESSIVE INTEGRATED MOVING AVERAGE (SARIMA)

SARIMA comprises the combination of Auto-Regressive (AR) and Moving Average (MA) methods [Shumway & Stoffer, 2011]. To understand these methods better, let $\mathbf{X}_t = \{x_0, x_1, x_2, \ldots, x_t\}$ be a time series composed of observations collected over time (t). We can also express time series as a sum of non-observable components $\mathbf{X}_t = T_t + S_t + \varepsilon_t$, such that $T_t$ represents the trend, $S_t$ the seasonality, and $\varepsilon_t$ the noise produced by a random process. Typically, $T_t$ and $S_t$ are seen as deterministic, i.e., a given observation strictly depends on past ones. On the other hand, $\varepsilon_t$ represents the stochasticity, being determined by probability density functions.

By considering the deterministic and stochastic components, one can model time series in terms of $q$ past random observations as Moving Average process, MA($q$), $x_t = x_{t-1} + \theta_1 \cdot \varepsilon_{t-1} + \theta_2 \cdot \varepsilon_{t-2} + \cdots + \theta_q \cdot \varepsilon_{t-q}$, such that $\{\theta_q\}$ are constants and $\{\varepsilon_t\}$ are values produced by a purely random process with mean $E(\mathbf{X}_t) = 0$ and variance $var(\mathbf{X}_t) = \sigma^2$ [Box et al., 2015, Shumway & Stoffer, 2011]. Time series can also be modeled by $p$ past observations as an Autoregressive process, AR($p$), $x_t = \phi_1 \cdot x_{t-1} + \phi_2 \cdot x_{t-2} + \cdots + \phi_p \cdot x_{t-p} + \varepsilon_t$, such that $\{\phi_p\}$ are constants and $\varepsilon_t$ is value produced by a purely random process with mean $E(\mathbf{X}_t) = 0$ and variance $var(\mathbf{X}_t) = \sigma^2$ [Box et al., 2015, Shumway & Stoffer, 2011].

These processes can also be combined to model time series according to $q$ past noise values and $p$ past observations using an Autoregressive and Moving Average (ARMA) process, as shown in Equation 2. Formally, this process is represented by ARMA($p, q$), in which $p$ is the autoregressive part AR($p$), and $q$ is the moving average one MA($q$) [Box et al., 2015, Shumway & Stoffer, 2011].

$$x_t = c + \phi_1 x_{t-1} + \phi_2 x_{t-2} + \ldots + \phi_p x_{t-p} + \varepsilon_t + \theta_1 \varepsilon_{t-1} + \theta_2 \varepsilon_{t-2} + \ldots + \theta_q \varepsilon_{t-q} \quad (2)$$

The ARMA process was designed to model stationary time series once their mean and variance do not vary over time [Box et al., 2015, Shumway & Stoffer, 2011]. In situations where the stationary restriction cannot be ensured, one may use an autoregressive integrated moving average process, ARIMA($p, d, q$), defined in Equation 3. In summary, the non-stationary source is removed by replacing $x_t$ with $w_t$, such that $w_t = \nabla^d x_t$, thus providing a model for originally non-stationary time series.

$$x_t = c + \phi_1 w_{t-1} + \phi_2 w_{t-2} + \ldots + \phi_{p+d} w_{t-p-d} + \varepsilon_t + \theta_1 \varepsilon_{t-1} + \theta_2 \varepsilon_{t-2} + \ldots + \theta_q \varepsilon_{t-q} \quad (3)$$

The seasonal ARIMA model, referred to as SARIMA, was developed to use differencing at a lag equal to the number of seasons (s) [Shumway & Stoffer, 2011], aiming to remove seasonal effects as shown in Equation 4.

$$\Phi_P(B^s)\phi(B) \bigtriangledown_s^D \bigtriangledown^d x_t = \delta + \Theta_Q(B^s)\theta(B)\varepsilon_t \quad (4)$$

In time series analyses, $B$ is just a backshift operator, i.e., $(1 + B + B^2)x_t$ is equivalent to $(x_t + x_{t-1} + x_{t-2})$. As aforementioned, $\phi(B)$ and $\theta(B)$ are ordinary AR and MA processes, respectively. On the other hand, $\Phi_P(B^s)$ is the seasonal autoregressive operator, and $\Theta_Q(B^s)$ is the seasonal moving average operator with $P$ and $Q$ orders, respectively. Finally, the ordinary and seasonal difference components are represented by $\bigtriangledown^d = (1 - B)^d$ and $\bigtriangledown_s^D = (1 - B^s)^D$. SARIMA is denoted by ARIMA($p, d, q$) $\times$ ($P, D, Q$)$_s$.

To exemplify the usage of ARIMA, let ARIMA($0, 1, 1$) $\times$ ($0, 1, 1$)$_{12}$ be a model used to analyze a temporal data with seasonal fluctuations occurring in every 12 months, which is shown in Equation 5 and expanded in Equation 6.

$$(1 - B^{12})(1 - B)x_t = (1 + \Theta B^{12})(1 + \theta B)\varepsilon_t \tag{5}$$

$$(1 - B - B^{12} + B^{13})x_t = (1 + \theta B + \Theta B^{12}) + \Theta\theta B^{13})\varepsilon_t \tag{6}$$

Previous formulation can also be represented by Equation 7, similarly to the AR, MA, ARMA, and ARIMA definitions.

$$x_t = x_{t-1} + x_{t-12} - x_{t-13} + \varepsilon_t + \theta\varepsilon_{t-1} + \Theta\varepsilon_{t-12} + \Theta\theta\varepsilon_{t-13} \tag{7}$$

In our experiments, we have use the SARIMAX function implemented in the *statsmodels* package (version 0.13.5). To find the optimal parameters, we have performed a grid search in the following interval: $p = [0, 2]$, $d = [0, 2]$, $q = [0, 5]$, $P = [0, 5]$, $D = [0, 2]$, $Q = [0, 2]$, $s = [0, 2]$, and trend = ['n', 'c', 't', 'ct'] such that 'n' means there is no trend, 'c' indicates a constant trend (i.e. a degree zero component of the trend polynomial), 't' indicates a linear trend over time, and 'ct' is both constant and linear.

### C.1.2 GATED RECURRENT UNITS (GRU)

The adoption of traditional Deep Neural Network (DNN) approaches to model temporal data has required long products of matrices, which can lead to vanishing or exploding gradients [Zhang et al., 2023]. Several approaches were designed to deal with this issue, including Gated Recurrent Units (GRU) [Cho et al., 2014a]. GRU can be seen as a Recurrent Neural Network that implements two important gates: reset and update. The reset gate is responsible for controlling how previous network states will be remembered, i.e., it helps to capture short-term dependencies over time [Zhang et al., 2023]. The update gate controls the influence of old states on the new ones, thus supporting the model of long-term dependencies over time [Zhang et al., 2023].

The reset gate ($\mathbf{R}_t$) in a given time instant $t$ is expressed as Equation 8, such that $\mathbf{X}_t$ is a sample of the time series (minibatch), $\mathbf{H}_{t-1}$ is the output produced by the hidden state in previous time instant, and $\mathbf{W}_{xr}$, $\mathbf{W}_{hr}$, and $\mathbf{b}_r$ represent, as in usual ANN (Artificial Neural Network), learnable weights and bias.

$$\mathbf{R}_t = \sigma(\mathbf{X}_t\mathbf{W}_{xr} + \mathbf{H}_{t-1}\mathbf{W}_{hr} + \mathbf{b}_r) \tag{8}$$

Likewise, Equation 9 mathematically describes the update gate. In both equations, $\sigma(\cdot)$ is a sigmoid function to bound values into the interval $(0, 1)$.

$$\mathbf{Z}_t = \sigma(\mathbf{X}_t\mathbf{W}_{xz} + \mathbf{H}_{t-1}\mathbf{W}_{hz} + \mathbf{b}_z) \tag{9}$$

After defining the reset and update gates, it is necessary to create a candidate hidden state integrating the reset gate with the state updating mechanism [Zhang et al., 2023], as shown in Equation 10. As previously mentioned, $\mathbf{W}_{xh}$ and $\mathbf{W}_{xh}$ are, respectively, learnable weights trained on the input data ($\mathbf{X}_t$) and the elementwise product operation between the output produced by the reset gate ($\mathbf{R}_t$) and the hidden state ($\mathbf{H}_{t-1}$) in previous time instant. Moreover, $\mathbf{b}_h$ is the bias used to train the candidate hidden state. It is worth mentioning that the sigmoid activation function $\sigma(\cdot)$ used in this step was replaced by a hyperbolic tangent function $\tanh(\cdot)$.

$$\tilde{\mathbf{H}}_t = \tanh(\mathbf{X}_t\mathbf{W}_{xh} + (\mathbf{R}_t \odot \mathbf{H}_{t-1})\mathbf{W}_{hh} + \mathbf{b}_h) \tag{10}$$

In the last step, we have to incorporate the update gate ($\mathbf{Z}_t$). In summary, $\mathbf{Z}_t$ weighs the influences of the hidden state in previous time instant ($\mathbf{H}_{t-1}$) and the new candidate hidden state ($\tilde{\mathbf{H}}_t$) to determine the new hidden state in the current time instant $t$ [Zhang et al., 2023].

$$\mathbf{H}_t = \mathbf{Z}_t \odot \mathbf{H}_{t-1} + (1 - \mathbf{Z}_t) \odot \tilde{\mathbf{H}}_t \tag{11}$$

Table 18: GRU architecture and parameters.

| Layer | Input Shape | Output Shape | #Param |
|---|---|---|---|
| GRU | [2871, 36] | [2871, 12] | 65,292 |
| (gru) GRU | [2871, 36] | [2871, 128], [1, 128] | 63,744 |
| (linear) Linear | [2871, 128] | [2871, 12] | 1,548 |

In our experiments, a GRU model was implemented using the library Pytorch (version 1.12.1), using the architecture shown in Table 18.

Finally, during the training phase, we have defined the learning rate and the number of epochs as 0.001 and 500, optimized with the Adam algorithm.

### C.1.3 Long Short-Term Memory (LSTM)

LSTM is also a DNN designed to deal with the problem of balancing long- and short-term information over time. The practical difference between GRU and LSTM is the implementation of three gates: Input, Forget, and Output gates [Hochreiter & Schmidhuber, 1997]. Similarly to reset and update gates, the LSTM gates are implemented in Equations 12, 13, and 14, respectively, such that $\mathbf{W}$. and $\mathbf{b}$. are weight and bias parameters [Zhang et al., 2023].

$$\mathbf{I}_t = \sigma(\mathbf{X}_t \mathbf{W}_{xi} + \mathbf{H}_{t-1} \mathbf{W}_{hi} + \mathbf{b}_i) \tag{12}$$

$$\mathbf{F}_t = \sigma(\mathbf{X}_t \mathbf{W}_{xf} + \mathbf{H}_{t-1} \mathbf{W}_{hf} + \mathbf{b}_f) \tag{13}$$

$$\mathbf{O}_t = \sigma(\mathbf{X}_t \mathbf{W}_{xo} + \mathbf{H}_{t-1} \mathbf{W}_{ho} + \mathbf{b}_o) \tag{14}$$

LSTM also implements a candidate memory cell, which is very similar to the previous gates, but using a hyperbolic tangent function instead a sigmoid one as shown in Equation 15 [Zhang et al., 2023].

$$\tilde{\mathbf{C}}_t = \tanh(\mathbf{X}_t \mathbf{W}_{xc} + \mathbf{H}_{t-1} \mathbf{W}_{hc} + \mathbf{b}_c) \tag{15}$$

Finally, the output ($\mathbf{C}_t$) produced by the memory cell uses the input ($\mathbf{I}_t$) and forget $\mathbf{F}_t$ gates to determine the influence of new ($\tilde{\mathbf{C}}_t$) and past ($\mathbf{C}_{t-1}$) information [Zhang et al., 2023].

$$\mathbf{C}_t = \mathbf{I}_t \odot \tilde{\mathbf{C}}_t + \mathbf{F}_t \odot \mathbf{C}_{t-1} \tag{16}$$

In our experiments, LSTM was implemented using the library Pytorch (version 1.12.1), using the architecture shown in Table 19.

Table 19: LSTM architecture and parameters.

| Layer | Input Shape | Output Shape | #Param |
|---|---|---|---|
| LSTM | [2871, 36] | [2871, 12] | 86,540 |
| (lstm) LSTM | [2871, 36] | [2871, 128] | 84,992 |
| (linear) Linear | [2871, 128] | [2871, 12] | 1,548 |

Finally, during the training phase, we have defined the learning rate and the number of epochs as 0.001 and 500, optimized with the Adam algorithm.

### C.1.4 A Language Modeling Framework for Time Series (CHRONOS)

CHRONOS is a framework that considers the advancements in language model architectures and training methodologies to address the challenges of time series forecasting [Ansari et al., 2024]. Although both domains share a sequential nature, they diverge in terms of data representation.

Natural language is composed of discrete tokens from a predefined vocabulary, while time series are characterized by continuous-valued observations.

Consider a time series $\mathbf{X}_t = \{x_1, \ldots, x_{t+H}\}$, where the first $t$ time steps constitute the historical context, and the remaining $H$ represent the forecast horizon. Language models operate on tokens from a finite vocabulary, so using them for time series data requires mapping the observations $\mathbf{X}_t \in \mathbb{R}$ to a finite set of tokens. To this end, CHRONOS first scale and then quantize observations into a fixed number of bins.

As part of the scaling process of CHRONOS, individual entries of the time series are normalized by the mean of the absolute values in the historical context, according to Equation 17.

$$\tilde{x}_i = \frac{x_i}{s}, \tag{17}$$

where $s = \frac{1}{t} \sum_t^{i=1} |x_i|$.

Following, the real values of the scaled time series $\tilde{\mathbf{X}}_t$ are converted into discrete tokens, by employing quantization [Rabanser et al., 2020]. Such approach ignore time and frequency information, treating the time series simply as a sequence and, because of that, no modifications are required to the language model architecture, except adjusting the time series vocabulary size $|\mathcal{V}_{\texttt{ts}}|$, which is composed by sequences of tokens, which depends on the quantization and may be different from the vocabulary size of the original language model.

Such an approach disregards explicit time and frequency information, treating the time series as just a sequence. Consequently, no modifications to the language model architecture are necessary, aside from adapting the vocabulary size of the time series, $|\mathcal{V}_{\texttt{ts}}|$. This vocabulary consists of sequences of tokens derived from the quantization process, which may differ from the original language model's vocabulary size. Adapting the vocabulary size involves truncating or extending the input and output embedding layers of the language model to accommodate the specific token sequences derived from the time series quantization process.

The forecasts produced by CHRONOS are based on probabilistic models, which allow them to generate multiple possible future scenarios by sampling repeatedly from their predicted range. These forecasts are initially represented as token IDs, which must be converted into real numbers and adjusted to produce the final predictions. Therefore, it is necessary a dequantization function to translate the token IDs into real values. Finally, these values are adjusted by reversing the scaling process. For mean scaling, this involves multiplying the values by the scaling factor $s$ [Ansari et al., 2024].

### C.1.5 SERIES-CORE FUSED TIME SERIES FORECASTER (SOFTS)

SOFTS [Han et al., 2024] is a model designed for forecasting multivariate time series, which are time series data that include multiple variables, or channels, at each time step. It works by combining all series into a global representation and then redistributing this to improve channel interactions efficiently while reducing reliance on the quality of individual channels.

Given historical values $\mathbf{X}_t \in \mathbb{R}^{C \times L}$ where $L$ represents the length of the lookback window, and $C$ is the number of channels. The goal of SOFTS is to predict the future values $\mathbf{Y} \in \mathbb{R}^{C \times H}$, where $H > 0$ is the forecast horizon.

SOFTS consists of the following main components. First, it normalizes the time series by centering them to zero mean and scaling them to unit variance. After forecasting, it reverses the normalization on the predicted series. Next, the model performs embedding on the lookback window, capturing the essential features of the data using a linear projection to embed the series of each channel to $S_0 = \mathbb{R}^{C \times d}$, where $d$ is the hidden dimension.

The embedded series then undergoes interaction through multiple layers of the STar Aggregate-Redistribute (STAR) module $S_i = \texttt{STAR}(S_{i-1})$, $i = 1, 2, \cdots, N$. This module uses a star-shaped structure that allows information to flow between different channels, improving how they interact. The STAR module differs from other common methods like attention [Vaswani et al., 2017], GNN [Kipf & Welling, 2016], and Mixer [Tolstikhin et al., 2021], which use a distributed structure that depends

on the quality of each channel. In contrast, the STAR module uses a centralized structure. It first aggregates information from all channels into a global core representation and then sends this core information to each channel. This approach reduces the complexity of interactions and decreases the reliance on the quality of individual channels.

After passing through $N$ layers of STAR, a linear predictor generates the forecast. Considering the output series representation as $S_N$, the prediction is computed as $\mathbf{Y} = \texttt{Linear}(S_N)$.

## C.2 GRAPH NEURAL NETWORKS

The organization of the Origin-Destination (OD) dataset with passengers' boarding and alighting allowed us to create the SUNT dataset, embedding a set of quantitative, temporal, and geospatial variables as complex network. Formally, we have used information on latitude, longitude, and time to create a spatial-temporal graph $G = \{G_1, G_2, ..., G_T\}$. For all $t = 1, ..., T$, $G_t = (V, E)$ stands for an attributed and directed graph at time $t$, where $V = \{v_1, v_2, ..., v_N\}$ is the set of $N$ vertices corresponding to the bus stops and stations, and $E$ is the set of edges corresponding to feasible routes. A directed edge $(v_i, v_j) \in E$, $i = 1...N$, $j = 1...N$, connects vertices $v_i, v_j \in V$ if, and only if, there is a feasible route for the bus traffic from the corresponding station $v_i$ to $v_j$ in the network. $G_t$ is a fixed graph structure since sets $V$ and $E$ do not change over time.

In our context, spatial data do not depend on time $t$, i.e., their information is time-invariant. Specifically, in every vertex $v_i \in V$, we store the following features: geographical position, number of boarding and alighting per vehicle, and passenger load. The features specifically concerning edges $(v_i, v_j) \in E$ include the distance between stops and stations, the trip duration, the mean velocity, and the Renovation Factor (RF). The RF is a well-known metric used in transportation research to assess the total demand in a line, i.e., it is computed on a set of edges that belong to the line [ITDP, 2016]. Formally, this metric is the ratio of the total demand of a line to the load on its critical link. Higher renovation factors occur when there are many short trips along the line. Corridors with very high renovation factor rates are more profitable because they handle the same number of paying customers with fewer vehicles [ITDP, 2016]. Besides the individual features, there is relevant information shared by both vertices and edges, such as the number of passengers per vehicle, lines and directions, vehicle characteristics, altitude, and trips.

To understand Graph Neural Networks (GNN) in our research, we take an input graph $G$, along with a set of node features $\mathbf{X} \in \mathbb{R}^{d \times |V|}$, and use this information to generate node embeddings $\mathbf{z}_v, \forall v \in V$.

The execution of GNNs depends on encoder-decoder functions to represent the graph as node embeddings, which is processed by using Neural Message Passing (NMP). In each message-passing iteration performed during the training phase, new knowledge from node embeddings is updated according to information aggregated from their neighborhoods [Hamilton, 2020, Wu et al., 2021].

During each message-passing iteration in a GNN, a hidden embedding $\mathbf{h}_v^k$ corresponding to each node $v \in V$ is updated according to information aggregated from $v$'s graph neighborhood $\mathcal{N}(v)$ [Hamilton, 2020]. This message-passing update can be expressed by Equations (18) and (19).

$$\mathbf{h}_v^k = UPDATE^k\left(\mathbf{h}_v^{k-1}, \mathbf{m}_{\mathcal{N}(v)}^k\right) \tag{18}$$

$$\mathbf{m}_{\mathcal{N}(v)}^k = AGGREGATE^k\left(\left\{\mathbf{h}_u^k, \forall u \in \mathcal{N}(v)\right\}\right) \tag{19}$$

From these equations, consider UPDATE and AGGREGATE as Neural Networks. At each iteration $k$ of the GNN, the AGGREGATE function takes as input the set of embeddings of the nodes in $v$'s graph neighborhood $\mathcal{N}(v)$ and generates a message $\mathbf{m}_{\mathcal{N}(v)}$ based on this aggregated neighborhood information. The different iterations of message passing are also sometimes known as the different layers of the GNN [Hamilton, 2020]. The function UPDATE then combines the message $\mathbf{m}_{\mathcal{N}(v)}^k$ with the previous embedding $\mathbf{h}_v^{k-1}$ of node $v$ to generate the updated embedding $\mathbf{h}_v^k$. The initial embeddings at $k = 0$ are set to the input features for all the nodes, i.e., $\mathbf{h}_v^{(0)} = \mathbf{x}_v, \forall v \in V$. After running $K$ iterations of the GNN message passing, we can use the output of the final layer to define the embeddings for each node, according to Equation (20).

$$\mathbf{z}_v = \mathbf{h}_v^{(K)}, \forall v \in V \tag{20}$$

The authors in [Hamilton, 2020] highlight that since the AGGREGATE function takes a *set* as input, GNNs defined in this way are permutation equivariant by design.

After going through a series of iterations of GNN message passing, the embeddings for each node will also contain information about all the features in their neighborhood. For example, after the first iteration ($k = 1$), each node embedding will incorporate information from the nodes that are directly connected to it, which can be reached by a path of length 1 in the graph; after the second iteration ($k = 2$), each node embedding will incorporate information from not just its immediate neighbors, but also the nodes that can be reached by a path of length 2 in the graph. This process continues, and after $k$ iterations, each node embedding will comprise information about its $k$-hop neighborhood.

The way GNNs collect feature information locally is similar to how convolutional kernels in Convolutional Neural Networks (CNNs) gather information from specific regions in an image. However, while CNNs gather information based on spatial locations, GNNs gather information based on local graph neighborhoods [Hamilton, 2020, Zhang et al., 2018, Kipf & Welling, 2016, Wu et al., 2020].

There are several types of GNN architectures that have been proposed in the literature, some of the most common were used in this work: GCN(Graph Convolutional Network) [Kipf & Welling, 2016], SAGE (SAmple and aggreGatE) [Hamilton et al., 2017], GAT (Graph Attention Networks) [Veličković et al., 2017], and CHEB (Chebyshev spectral graph convolutional operator) [Defferrard et al., 2016]. They have been widely used in various tasks such as node classification, link prediction, and graph classification.

### C.2.1 GRAPH CONVOLUTIONAL NETWORK (GCN)

GCN uses a graph convolution operation to learn representations of nodes and edges in a graph. One of the key features of GCNs is weight-sharing, which means that the same weight matrix is used for every node in the graph, by employing the symmetric-normalized aggregation as well as the self-loop update approach. This allow GCNs to learn representations of large graphs efficiently.

The symmetric-normalized aggregation employed by Kipf & Welling [2016] in the GCNs is defined according to Equation ( 21).

$$\mathbf{m}_{\mathcal{N}(v)} = \sum_{u \in \mathcal{N}(v)} \frac{\mathbf{h}_u}{\sqrt{|\mathcal{N}(v)||\mathcal{N}(u)|}} \tag{21}$$

As a simplification of the neural message passing approach, now the aggregation is taken over the node's neighbors as well as the node itself, omitting the explicit update step. Therefore, the GCN message passing function is defined according to Equation (22).

$$\mathbf{h}_v^k = \sigma \left( \mathbf{W}^k \sum_{u \in \mathcal{N}(v) \cup \{v\}} \frac{\mathbf{h}_u}{\sqrt{|\mathcal{N}(v)||\mathcal{N}(u)|}} \right), \tag{22}$$

where the weight matrix $\mathbf{W}^k$ is a trainable parameter matrix and $\sigma$ denotes an elementwise non-linearity, as for example ReLU [Kipf & Welling, 2016].

In our experiments, GCN was implemented using the library Pytorch (version 1.12.1), using the architecture shown in Table 27.

Table 20: GCN architecture and parameters.

| Layer | Input Shape | Output Shape | #Param |
|-------|-------------|--------------|--------|
| GCN | [2871, 36], [2, 4158], [4158] | [2871, 12] | 6,284 |
| (conv1) GCNConv | [2871, 36], [2, 4158], [4158] | [2871, 128] | 4,736 |
| (linear) Linear | [2871, 128] | [2871, 12] | 1,548 |

### C.2.2 LOCAL EXTREMA CONVOLUTION (LECONV)

LEConv [Ranjan et al., 2020] is a GCN that captures local extremum information by utilizing the difference operator to determine the importance of nodes relative to their neighbors, as defined in Equation (23).

$$\mathbf{x}'_v = \mathbf{x}_v \mathbf{W}_1 + \sum_{u \in \mathcal{N}(v)} e_{(u,v)} \cdot (\mathbf{W}_2 \mathbf{x}_v - \mathbf{W}_3 \mathbf{x}_u), \tag{23}$$

where $e_{(u,v)}$ denotes the edge weight from source node $u$ to target node $v$; and $\mathbf{W}_1, \mathbf{W}_2, \mathbf{W}_3$ are learnable parameters.

In our experiments, LEConv was implemented using the library Pytorch (version 1.12.1), using the architecture shown in Table 21.

Table 21: LEConv architecture and parameters.

| Layer | Input Shape | Output Shape | #Param |
|---|---|---|---|
| LEConv | [2871, 36], [2, 4158], [4158] | [2871, 12] | 15,628 |
| (conv1) LEConv | [2871, 36], [2, 4158], [4158] | [2871, 128] | 14,080 |
| (linear) Linear | [2871, 128] | [2871, 12] | 1,548 |

### C.2.3 GRAPH ATTENTION NETWORKS (GAT)

On the other hand, GATs are a type of GNN that uses attention mechanisms to learn representations of nodes in a graph. Attention mechanisms allow GATs to focus on the most important neighbors for each node, improving the network's performance.

According to [Veličković et al., 2017], GAT uses attention weights to define a weighted sum of the neighbors as defined in Equation (24).

$$\mathbf{x}'_v = \sum_{u \in \mathcal{N}(v) \cup \{v\}} \alpha_{(v,u)} \mathbf{W} \mathbf{x}_u \tag{24}$$

where $\alpha_{(v,u)}$ denotes the attention on neighbor $u \in \mathcal{N}(v)$ when we are aggregating information at node $v$. It is possible to add multiple attention *heads*, which is closely related to the transformer architecture defined by Vaswani et al. [2017]. In this approach, one computes distinct attention weights using independently parameterized attention layers. In our experiments, GAT was implemented using the library Pytorch (version 1.12.1), using the architecture shown in Table 22.

Table 22: GAT architecture and parameters.

| Layer | Input Shape | Output Shape | #Param |
|---|---|---|---|
| GAT | [2871, 36], [2, 4158], [4158] | [2871, 12] | 6,540 |
| (conv1) GATConv | [2871, 36], [2, 4158], [4158] | [2871, 128] | 4,992 |
| (linear) Linear | [2871, 128] | [2871, 12] | 1,548 |

### C.2.4 SELF-SUPERVISED GRAPH ATTENTION NETWORK (SUPERGAT)

SuperGAT[Kim & Oh, 2022] was proposed for guiding attention heads with the presence or absence of an edge between a node pair. The authors exploit the link prediction task to self-supervise attention with the label 1 if an edge exists for a pair of nodes $u$ and $v$, and label 0 otherwise. Therefore, they analyze what graph attention learns and how it relates to the presence of edges. In this analysis, SuperGAT focus on two commonly used attention mechanisms, GAT's original single-layer neural network (GO) and dot-product (DP), been extended in two variants: SuperGAT scaled dot-product (SD), and SuperGAT mixed GO and DP (MX), as defined in Equation (25).

$$\mathbf{x}'_v = \alpha_{(v,v)} \mathbf{W} \mathbf{x}_v + \sum_{u \in \mathcal{N}(v)} \alpha_{(v,u)}^{\text{SD or MX}} \mathbf{W} \mathbf{x}_u, \tag{25}$$

where $\alpha_{(v,u)}^{SD}$ divides the dot-product of nodes by a square root of dimension as Transformer [Vaswani et al., 2017], preventing some large values to dominate the entire attention after softmax. In another direction, $\alpha_{(v,u)}^{MX}$ multiplies GO and DP attention with sigmoid, motivated by the gating mechanism of GRUs [Cho et al., 2014a]. Since DP attention with the sigmoid function represents the probability of an edge, it can softly exclude unlikely neighbors while implicitly prioritizing the remaining nodes [Kim & Oh, 2022].

In our experiments, SuperGAT was implemented using the library Pytorch (version 1.12.1), using the architecture shown in Table 23 with its MX variant.

Table 23: SuperGAT architecture and parameters.

| Layer | Input Shape | Output Shape | #Param |
|---|---|---|---|
| SuperGAT | [2871, 36], [2, 4158] | [2871, 12] | 6,540 |
| (conv1) SuperGATConv | [2871, 36], [2, 4158] | [2871, 128] | 4,992 |
| (linear) Linear | [2871, 128] | [2871, 12] | 1,548 |

### C.2.5 EFFICIENT GRAPH CONVOLUTION (EGC)

EGC [Tailor et al., 2021] provides both spatial and spectral interpretations for an isotropic GNN architecture [Tailor et al., 2021]. As an isotropic model, it requires memory proportional to the number of vertices in the graph ($\mathcal{O}(V)$); in contrast, anisotropic models require memory proportional to the number of edges ($\mathcal{O}(E)$). For a layer with in-dimension of $F$ and out-dimension of $F'$, the authors use $B$ basis weights $\mathbf{W}_b \in \mathbb{R}^{F' \times F}$, computing the output for each node $v \in \mathcal{V}$ by calculating combination weighting coefficients $w^{(v)} \in \mathbb{R}^B$, and weighting the results of each aggregation using the different basis weights $\mathbf{W}_b$. Therefore, the layer output for node $v$ is the weighted combination of aggregation outputs as defined in Equation (26).

$$\mathbf{x}'_v = \sum_{b=1}^{B} w_b^{(v)} \sum_{u \in \mathcal{N}(v)} \alpha_{(v,u)} \mathbf{W}_b \mathbf{x}_u, \tag{26}$$

where $\alpha_{(v,u)}$ is some function of nodes $v$ and $u$, and $\mathcal{N}(v)$ denotes the neighbours of $v$. A popular method pioneered by GAT, as previously mentioned, to boost representational power is to represent $\alpha$ using a learned function of the two nodes' representations such as attention heads.

In our experiments, EGC was implemented using the library Pytorch (version 1.12.1), using the architecture shown in Table 24.

Table 24: EGC architecture and parameters.

| Layer | Input Shape | Output Shape | #Param |
|---|---|---|---|
| EGC | [2871, 36], [2, 4158] | [2871, 12] | 5,164 |
| (conv1) EGConv | [2871, 36], [2, 4158] | [2871, 128] | 3,616 |
| (linear) Linear | [2871, 128] | [2871, 12] | 1,548 |

### C.2.6 SAMPLE AND AGGREGATE (SAGE)

SAGE is a GNN architecture that, instead of training individual embeddings for each node, learns a set of aggregation functions that are applied to the features of a node's local neighborhood to create an embedding for that node [Hamilton et al., 2017]. Each aggregator function aggregates information from a different number of hops, or search depth, away from a given node.

The embedding generation, or forward propagation algorithm, of SAGE assumes that the parameters of $K$ aggregator functions (denoted $AGGREGATE^{(k)}$, $\forall k \in 1, ..., K$), which aggregate information from node neighbors, were learned. Moreover, a set of weight matrices $\mathbf{W}^k$, which is used to propagate information between different search depths (layers of the model) was also learned [Hamilton et al., 2017].

Given a graph $G$ and features for all its nodes $\mathbf{x}_v$, $\forall v \in V$, $k$ denotes the current step in the depth of the search and $\mathbf{h}^k$ denotes a node's representation at this step. First, each node $v \in \mathcal{V}$ aggregates the representations of its immediate neighboring nodes using Equation (27), a slight modification of Equation (19).

$$\mathbf{h}_{\mathcal{N}(v)}^k = AGGREGATE^{(k)} \left( \left\{ \mathbf{h}_u^{k-1}, \forall u \in \mathcal{N}(v) \right\} \right) \tag{27}$$

Several types of aggregation functions can be used in SAGE, such as SUM, MEAN, MAX, and others [Hamilton et al., 2017].

After aggregating the neighboring feature vectors, SAGE then concatenates the node's current representation, $\mathbf{h}_v^{k-1}$, with the aggregated neighborhood vector, and this concatenated vector is fed through a fully connected layer with nonlinear activation function $\sigma$. Such concatenation is performed according to Equation (39).

$$\mathbf{h}_v^k = \sigma \left( \mathbf{W}^k \cdot CONCAT \left( \mathbf{h}_v^{k-1}, \mathbf{h}_{\mathcal{N}(v)}^k \right) \right) \tag{28}$$

After running $K$ iterations of the SAGE, we can use the output of the final layer to define the embeddings for each node, also according to Equation(20) [Hamilton et al., 2017]. In our experiments, SAGE was implemented using the library Pytorch (version 1.12.1), using the architecture shown in Table 25.

Table 25: SAGE architecture and parameters.

| Layer | Input Shape | Output Shape | #Param |
|---|---|---|---|
| SAGE | [2871, 36], [2, 4158] | [2871, 12] | 10,892 |
| (conv1) SAGEConv | [2871, 36], [2, 4158] | [2871, 128] | 9,344 |
| (linear_out) Linear | [2871, 128] | [2871, 12] | 1,548 |

### C.2.7 CHEBYSHEV SPECTRAL GRAPH CONVOLUTIONAL OPERATOR (CHEB)

In another direction, CHEB [Defferrard et al., 2016] implements an efficient generalization of the CNNs to arbitrary graphs, considering that basic convolutional filters on graphs can be represented as polynomials of the Laplacian $\mathbf{L}$ of $G$. An input vector $\mathbf{X} \in \mathbb{R}^N$ is a signal defined on a graph $G$ with $N$ nodes. Then a graph convolution of input signals $\mathbf{X}$ with filters $g$ on $G$ is defined by $*_{\mathcal{G}}\mathbf{X} = p_M(\mathbf{L})x$. Therefore, $p_M(\mathbf{L})$ is defined as the spectral filter of the eigenvalues of the Laplacian, and $M$ is the polynomial degree of the eigenvalues. According to Hamilton [2020], if we use a degree $M$ polynomial, then we ensure that the filtered signal at each node depends on information in its M-hop neighborhood. CHEB defined $p_M(\mathbf{L})$ using Chebyshev polynomials [Mason & Handscomb, 2002]. In our experiments, CHEB was implemented using the library Pytorch (version 1.12.1), using the architecture shown in Table 26.

Table 26: CHEB architecture and parameters.

| Layer | Input Shape | Output Shape | #Param |
|---|---|---|---|
| CHEB | [2871, 36], [2, 4158], [4158] | [2871, 12] | 15,500 |
| (conv1) CHEBConv | [2871, 36], [2, 4158], [4158] | [2871, 128] | 13,952 |
| (linear) Linear | [2871, 128] | [2871, 12] | 1,548 |

### C.2.8 SIMPLE SPECTRAL GRAPH CONVOLUTION ($\text{S}^2\text{GC}$)

According to Zhu & Koniusz [2021], without specially designed architectures, the performance of GCNs degrades quickly with increased depth. As the aggregated neighborhood size and neural network depth are two completely orthogonal aspects of graph representation, several methods focus on summarizing the neighborhood by aggregating M-hop neighborhoods of nodes while using shallow neural networks. However, these methods still encounter oversmoothing, and suffer from high computation and storage costs. Because of that, Zhu & Koniusz [2021] proposed a modified

Markov Diffusion Kernel to derive a variant of GCN called Simple Spectral Graph Convolution (S$^2$GC)) network with the softmax classifier after the linear layer:

$$\hat{Y} = \text{softmax}\left(\frac{1}{\text{M}} \sum_{m=1}^{\text{M}} \left((1-\alpha)\left(\hat{\mathbf{D}}^{-1/2}\hat{\mathbf{A}}\hat{\mathbf{D}}^{-1/2}\right)^m \mathbf{X} + \alpha\mathbf{X}\right) \mathbf{W}\right), \tag{29}$$

where $\hat{\mathbf{A}} = \mathbf{A} + \mathbf{I}$ denotes the adjacency matrix with inserted self-loops and $\hat{\mathbf{D}}$ be the diagonal degree matrix. In our experiments, S$^2$GC was implemented using the library Pytorch (version 1.12.1), where is named as SSGConv, using the architecture shown in Table 27.

Table 27: S$^2$GC architecture and parameters.

| Layer | Input Shape | Output Shape | #Param |
|---|---|---|---|
| S$^2$GC | [2871, 36], [2, 4158], [4158] | [2871, 12] | 6,284 |
| (conv1) S$^2$GConv | [2871, 36], [2, 4158], [4158] | [2871, 128] | 4,736 |
| (linear) Linear | [2871, 128] | [2871, 12] | 1,548 |

### C.2.9 PATH INTEGRAL BASED GRAPH NEURAL NETWORKS (PAN)

PAN [Ma et al., 2020] introduces a convolution operation that considers all paths between the message sender and receiver, with learnable weights based on path length, corresponding to the maximal entropy random walk. It extends the graph Laplacian into a new transition matrix, called the Maximal Entropy Transition (MET) matrix, derived from a path integral formalism [Feynman et al., 2010].

Similar to the basis formed by the graph Laplacian, the convolutional layer of PAN is based on the spectrum of $MET$ matrix, as defined in Equation (30)

$$X^{(h+1)} = MET^h X^h W^h, \tag{30}$$

where $h$ refers to the layer number. Applying $MET$ to the input $X$ essentially performs a weighted average over the neighbors of a given node. This raises whether the normalization consistent with the path integral formulation is the most effective approach in a data-driven context.

In our experiments, PAN was implemented using the library Pytorch (version 1.12.1), where is named as PANConv, using the architecture shown in Table 28.

Table 28: PAN architecture and parameters.

| Layer | Input Shape | Output Shape | #Param |
|---|---|---|---|
| PAN | [2871, 36], [2, 4158] | [2871, 12] | 6,286 |
| (conv1) PANConv | [2871, 36], [2, 4158] | [2871, 128], [2871, 2871] | 4,738 |
| (linear) Linear | [2871, 128] | [2871, 12] | 1,548 |

### C.3 ANTI-SYMMETRIC DEEP GRAPH NETWORKS (A-DGN)

A-DGN [Gravina et al., 2023] is a framework for effective long-term propagation of information in Deep Graph Networks (DGNs) architectures designed through the lens of ordinary differential equations (ODEs).

DGNs are a class of learning models designed to map and compress complex relational information from graphs into feature vectors that capture both the graph's topology and label information. DGNs are composed of multiple layers, which updates node representations by aggregating the nodes and their neighbors, following a message-passing approach. However, in some cases, relying solely on local node interactions is insufficient to learn meaningful embeddings. In such scenarios, DGNs often need to capture information from distant node interactions within the graph, which can be achieved by stacking multiple layers.

A-DGN preserves long-range information between nodes while establishing conditions that prevent gradient vanishing or explosion. Therefore, in A-DGN, a node is updated according to Equation (31).

$$\mathbf{x}'_v = \mathbf{x}_v + \epsilon \cdot \sigma((\mathbf{W} - \mathbf{W}^T - \gamma\mathbf{I})\mathbf{x}_v + \Phi(X, \mathcal{N}_{(v)}) + \mathbf{b}), \tag{31}$$

where $(\mathbf{W} - \mathbf{W}^T)$ is the anti-symmetric weight matrix; $\Phi(X, \mathcal{N}_{(v)})$ denotes a MessagePassing layer, which can be any function that aggregates nodes (and edges) information; $\mathbf{I}$ is the identity matrix; $\gamma$ is a hyper-parameter that regulates the strength of the diffusion; $\mathbf{X}$ is the node feature matrix of the whole graph; $\sigma$ is the discretization step; and $\mathbf{b}$ is a bias vector that contain the trainable parameters of the system.

In our experiments, A-DGN was implemented using the library Pytorch (version 1.12.1), where is named as AntiSymmetricConv, using the architecture shown in Table 29.

Table 29: A-DGN architecture and parameters.

| Layer | Input Shape | Output Shape | #Param |
|-------|-------------|--------------|--------|
| A-DGN | [2871, 36], [2, 4158] | [2871, 12] | 3,072 |
| (conv1) A-DGNConv | [2871, 36], [2, 4158] | [2871, 36] | 2,628 |
| (act) Tanh | [2871, 36] | [2871, 36] | – |
| (phi) GCNConv | [2871, 36], [2, 4158] | [2871, 36] | 1,296 |
| (linear) Linear | [2871, 36] | [2871, 12] | 444 |

## C.4 TEMPORAL GRAPH NEURAL NETWORKS

### C.4.1 GRAPH CONVOLUTIONAL RECURRENT NETWORKS (GCRN)

GCRN [Seo et al., 2018] is one extension of CHEB [Defferrard et al., 2016] for modeling and predicting time-varying graph-based data by merging CNN for graph-structured data and Recurrent Neural Networks (RNNs) to identify simultaneously meaningful spatial structures and dynamic patterns.

A special class of RNN is the LSTM, described in Section . To generalize the LSTM model to graphs, Seo et al. [2018] replaced the convolution defined by Equations 12, 13, 14, and 15, to a graph convolution $*_{\mathcal{G}}$, as defined in Equations 32, 33, 34, and 35.

$$\mathbf{I}_t = \sigma(\mathbf{W}_{xi} *_{\mathcal{G}} \mathbf{X}_t + \mathbf{W}_{hi} *_{\mathcal{G}} \mathbf{H}_{t-1} + \mathbf{b}_i) \tag{32}$$

$$\mathbf{F}_t = \sigma(\mathbf{W}_{xf} *_{\mathcal{G}} \mathbf{X}_t + \mathbf{W}_{hf} *_{\mathcal{G}} \mathbf{H}_{t-1} + \mathbf{b}_f) \tag{33}$$

$$\mathbf{O}_t = \sigma(\mathbf{W}_{xo} *_{\mathcal{G}} \mathbf{X}_t + \mathbf{W}_{ho} *_{\mathcal{G}} \mathbf{H}_{t-1} + \mathbf{b}_o) \tag{34}$$

$$\tilde{\mathbf{C}}_t = \tanh(\mathbf{W}_{xc} *_{\mathcal{G}} \mathbf{X}_t + \mathbf{W}_{hc} *_{\mathcal{G}} \mathbf{H}_{t-1} + \mathbf{b}_c) \tag{35}$$

In our experiments, we named GConvLSTM the GCRN architecture that stacks a graph CNN for feature extraction and an LSTM as described previously.

In addition, as mentioned by the authors Seo et al. [2018], GCRN is not limited to LSTMs and is straightforward to apply to any kind of recursive networks. In our experiments, we named GConvGRU the GCRN architecture that stacks a graph CNN for feature extraction and a GRU, by replacing formulation of GRU defined by Equations 8, 9, and 10 to Equations 36, 37, and 38.

$$\mathbf{R}_t = \sigma(\mathbf{W}_{xr} *_{\mathcal{G}} \mathbf{X}_t + \mathbf{W}_{hr} *_{\mathcal{G}} \mathbf{H}_{t-1} + \mathbf{b}_r) \tag{36}$$

$$\mathbf{Z}_t = \sigma(\mathbf{W}_{xz} *_{\mathcal{G}} \mathbf{X}_t + \mathbf{W}_{hz} *_{\mathcal{G}} \mathbf{H}_{t-1} + \mathbf{b}_z) \tag{37}$$

$$\tilde{\mathbf{H}}_t = \tanh(\mathbf{W}_{xh} *_{\mathcal{G}} \mathbf{X}_t + \mathbf{W}_{hh} *_{\mathcal{G}} (\mathbf{R}_t \odot \mathbf{H}_{t-1}) + \mathbf{b}_h) \tag{38}$$

In our experiments, GConvLSTM and GConvGRU were implemented using the library Pytorch (version 1.12.1), using the architectures shown in Tables 30 and 31, respectively.

Table 30: GConvLSTM architecture and parameters.

| Layer | Input Shape | Output Shape | #Param |
|---|---|---|---|
| GConvLSTM | [2871, 36], [2, 4158], [4158] | [2871, 12], [2871, 128], [2871, 128] | 87,436 |
| (recurrent) GConvLSTM | [2871, 36], [2, 4158], [4158] | [2871, 128], [2871, 128] | 85,888 |
| (conv$_{xi}$) CHEBConv | [2871, 36], [2, 4158], [4158] | [2871, 128] | 4,736 |
| (conv$_{hi}$) CHEBConv | [2871, 128], [2, 4158], [4158] | [2871, 128] | 16,512 |
| (conv$_{xf}$) CHEBConv | [2871, 36], [2, 4158], [4158] | [2871, 128] | 4,736 |
| (conv$_{hf}$) CHEBConv | [2871, 128], [2, 4158], [4158] | [2871, 128] | 16,512 |
| (conv$_{xc}$) CHEBConv | [2871, 36], [2, 4158], [4158] | [2871, 128] | 4,736 |
| (conv$_{hc}$) CHEBConv | [2871, 128], [2, 4158], [4158] | [2871, 128] | 16,512 |
| (conv$_{xo}$) CHEBConv | [2871, 36], [2, 4158], [4158] | [2871, 128] | 4,736 |
| (conv$_{ho}$) CHEBConv | [2871, 128], [2, 4158], [4158] | [2871, 128] | 16,512 |
| (linear)Linear | [2871, 128] | [2871, 12] | 1,548 |

Table 31: GConvGRU architecture and parameters.

| Layer | Input Shape | Output Shape | #Param |
|---|---|---|---|
| GConvGRU | [2871, 36], [2, 4158], [4158] | [2871, 12] | 65,292 |
| (recurrent) CHEBConv | [2871, 36], [2, 4158], [4158] | [2871, 128] | 63,744 |
| (conv$_{xz}$) CHEBConv | [2871, 36], [2, 4158], [4158] | [2871, 128] | 4,736 |
| (conv$_{hz}$) CHEBConv | [2871, 128], [2, 4158], [4158] | [2871, 128] | 16,512 |
| (conv$_{xr}$) CHEBConv | [2871, 36], [2, 4158], [4158] | [2871, 128] | 4,736 |
| (conv$_{hr}$) CHEBConv | [2871, 128], [2, 4158], [4158] | [2871, 128] | 16,512 |
| (conv$_{xh}$) CHEBConv | [2871, 36], [2, 4158], [4158] | [2871, 128] | 4,736 |
| (conv$_{hh}$) CHEBConv | [2871, 128], [2, 4158], [4158] | [2871, 128] | 16,512 |
| (linear)Linear | [2871, 128] | [2871, 12] | 1,548 |

### C.4.2 TEMPORAL GRAPH CONVOLUTIONAL NETWORK (TGCN)

TGCN [Zhao et al., 2019] is a neural network-based traffic forecasting method that combines the Graph Convolutional Network (GCN) and the Gated Recurrent Unit (GRU). The GCN is used to capture the topological structure of the urban road network for obtaining the spatial dependence, and the GRU is used to capture the dynamic variation of traffic information on the roads for obtaining the temporal dependence and eventually for realizing traffic prediction tasks.

In such a context, traffic forecasting can be viewed as learning a mapping function $f$ based on the road network, represented by a graph $G$ and feature matrix $\mathbf{X} \in \mathbb{R}^{N \times P}$, where $P$ is the number of node attribute features, that is, the length of the historical time series. Therefore, $\mathbf{X}_t$ denotes the traffic speed in all sections at time $t$. Traffic speeds of future $T$ moments are calculated as follows:

$$[\mathbf{X}_{t+1}, ..., \mathbf{X}_{t+T}] = f(G; \mathbf{X}_{t-n}, ..., \mathbf{X}_{t-1}, \mathbf{X}_t)), \tag{39}$$

where $n$ is the length of a given historical time series, which represents the time steps of the historical traffic data, and $T$ is the time series length that needs to be forecasted. Consider the historical traffic data as input into the TGCN model to obtain $n$ hidden states, $\mathbf{H}$, that covered spatiotemporal characteristics: $\{\mathbf{H}_{t-n}, \cdots, \mathbf{H}_{t-1}, \mathbf{H}_t\}$. The TGCN formulas are represented in Equations 40, 41, and 42, where $f(A, \mathbf{X}_t)$ represents 2-layer GCN, $\mathbf{X}$ represents the feature matrix, and $A$ the adjacency matrix.

$$\mathbf{R}_t = \sigma\left(\mathbf{W}_r * [f(A, \mathbf{X}_t), \mathbf{H}_{t-1}] + \mathbf{b}_r\right) \tag{40}$$

$$\mathbf{Z}_t = \sigma\left(\mathbf{W}_z * \left[f\left(A, \mathbf{X}_t\right), \mathbf{H}_{t-1}\right] + \mathbf{b}_z\right) \tag{41}$$

$$\tilde{\mathbf{H}}_t = \tanh(\mathbf{W}_h * \left[f\left(A, \mathbf{X}_t\right), \left(\mathbf{R}_t \odot \mathbf{H}_{t-1}\right)\right] + \mathbf{b}_h \tag{42}$$

Therefore, the TGCN model can deal with complex spatial dependence and temporal dynamics. In our experiments, TGCN was implemented using the library Pytorch (version 1.12.1), using the architecture shown in Table 32.

Table 32: Model architecture and parameters for TGCN.

| Layer | Input Shape | Output Shape | #Param |
|---|---|---|---|
| TGCN | [2871, 36], [2, 4158], [4158] | [2871, 12], [2871, 128] | 114,444 |
| (recurrent) TGCN | [2871, 36], [2, 4158], [4158] | [2871, 128] | 112,896 |
| (conv$_z$) GCNConv | [2871, 36], [2, 4158], [4158] | [2871, 128] | 4,736 |
| (linear$_z$) Linear | [2871, 256] | [2871, 128] | 32,896 |
| (conv$_r$) GCNConv | [2871, 36], [2, 4158], [4158] | [2871, 128] | 4,736 |
| (linear$_r$) Linear | [2871, 256] | [2871, 128] | 32,896 |
| (conv$_h$) GCNConv | [2871, 36], [2, 4158], [4158] | [2871, 128] | 4,736 |
| (linear$_h$) Linear | [2871, 256] | [2871, 128] | 32,896 |
| (linear) Linear | [2871, 128] | [2871, 12] | 1,548 |

### C.4.3 ATTENTION TEMPORAL GRAPH CONVOLUTIONAL NETWORK (A3TGCN)

A3TGCN [Bai et al., 2021] is an improvement of TGCN, in which the attention mechanism was introduced to re-weight the influence of historical traffic states and thus to capture the global variation trends of the traffic state.

Considering the previous formulation of TGCN, the hidden states were fed into the attention model to determine the context vector that covers the global traffic variation information. Thus, the weight, $a$, of each hidden state, $\mathbf{H}$, was calculated by Softmax using a multilayer perception: $\{a_{t-n}, \cdots, a_{t-1}, a_t\}$. The context vector that covers global traffic variation information was calculated using the weighted sum. Finally, forecasting results were outputted using the fully connected layer.

Therefore, A3TGCN uses GCN to capture the topological features of the road network, ensuring spatial dependence, GRU to capture the dynamic variation in node attributes, revealing the local temporal tendencies of traffic conditions, and an attention model to discern the global trends in traffic conditions. In our experiments, A3TGCN was implemented using the library Pytorch (version 1.12.1), using the architecture shown in Table 33.

Table 33: A3TGCN architecture and parameters.

| Layer | Input Shape | Output Shape | #Param |
|---|---|---|---|
| A3TGCN | [2871, 36], [2, 4158], [4158] | [2871, 12] | 101,005 |
| (recurrent) A3TGCN | [2871, 1, 36], [2, 4158], [4158] | [2871, 128] | 99,457 |
| (linear) Linear | [2871, 128] | [2871, 12] | 1,548 |

### C.4.4 DIFFUSION CONVOLUTIONAL RECURRENT NEURAL NETWORK (DCRNN)

DCRNN [Li et al., 2017] integrates diffusion convolution, the *Sequence to Sequence* architecture and the scheduled sampling technique, also to predict the future traffic speed given previously observed traffic flow.

DCRNN models the spatial dependency by relating traffic flow to a diffusion process, which explicitly captures the stochastic nature of traffic dynamics. This diffusion process is characterized by a random walk on a graph $G$. After many steps, a Markov process converges to a stationary distribution [Li et al., 2017].

As prior approaches, DCRNN similarly employs GRU to characterize temporal dependencies. It achieves this by substituting matrix multiplications within GRU with diffusion convolution $\star_{\mathcal{G}}$,

resulting in the development of the proposed Diffusion Convolutional Gated Recurrent Unit (DCGRU). Consequently, Equations 40, 41, and 42 are replaced by the subsequent Equations 43, 44, and 45.

$$\mathbf{R}_t = \sigma \left( \mathbf{W}_r \star_{\mathcal{G}} [\mathbf{X}_t, \mathbf{H}_{t-1}] + \mathbf{b}_r \right) \tag{43}$$

$$\mathbf{Z}_t = \sigma \left( \mathbf{W}_z \star_{\mathcal{G}} [\mathbf{X}_t, \mathbf{H}_{t-1}] + \mathbf{b}_z \right) \tag{44}$$

$$\tilde{\mathbf{H}}_t = \tanh \left( \mathbf{W}_c \star_{\mathcal{G}} [\mathbf{X}_t, (\mathbf{R}_t \odot \mathbf{H}_{t-1})] + \mathbf{b}_h \right) \tag{45}$$

Following that, DCRNN utilizes the *Sequence to Sequence architecture*, wherein historical time series data is inputted into an encoder. The final states of the encoder are then employed to initialize the decoder. Subsequently, the decoder generates predictions, relying on either preceding ground truth data or the model's output. Finally, to mitigate the discrepancy between the input distributions of training and testing, which can cause degraded performance, DCRNN integrates scheduled sampling, feeding the model with either the ground truth observation with probability $\epsilon_i$ or the prediction by the model with probability 1-$\epsilon_i$ at the $i$th iteration. During the training process, $\epsilon_i$ gradually decreases to 0 to allow the model to learn the testing distribution. In our experiments, DCRNN was implemented using the library Pytorch (version 1.12.1), using the architecture shown in Table 34.

Table 34: Model architecture and parameters for DCRNN

| Layer | Input Shape | Output Shape | #Param |
|---|---|---|---|
| DCRNN | [2871, 36], [2, 4158], [4158] | [2871, 12] | 127,884 |
| (recurrent) DCRNN | [2871, 36], [2, 4158], [4158] | [2871, 128] | 126,336 |
| (conv$_{xz}$) DCRNNConv | [2871, 164], [2, 4158], [4158] | [2871, 128] | 42,112 |
| (conv$_{xr}$) DCRNNConv | [2871, 164], [2, 4158], [4158] | [2871, 128] | 42,112 |
| (conv$_{xh}$) DCRNNConv | [2871, 164], [2, 4158], [4158] | [2871, 128] | 42,112 |
| (linear) Linear | [2871, 128] | [2871, 12] | 1,548 |

# D    EVALUATION PROCESS

This section summarizes the evaluation process designed to train and assess the performance of the learning models. After exploring classification and regression tasks, specific metrics were considered to analyze the obtained results, as discussed in the following sections.

## D.1    CLASSIFICATION TASKS

This section summarizes the evaluation process designed to assess the performance of the learning models. After exploring classification and regression tasks, specific metrics were considered to analyze the obtained results, as discussed in the following sections.

The validation metrics were computed using confusion (contingency) matrices, with the number of true positives (TP), true negatives (TN), false positives (FP), and false negatives (FN) aggregated from the outputs produced by a 10-fold cross-validation strategy.

The first adopted metric was accuracy (Equation 46), which is responsible for computing the total number of correct classifications, represented by the sum of TP and TN, divided by the number $n$ of examples used in validation (attempts), thus producing an overall performance result to compare distinct approaches.

$$\text{Acc} = \frac{(TP + TN)}{n} \tag{46}$$

Another metric considered in our experiments was the F1-score, which provides a harmonic mean of precision and recall. Precision computes the rate of correct classifications for the positive label over the total number of attempts classified as positive (both correct and incorrect). Recall computes the

number of true positive classifications divided by the total number of elements expected to be under the positive label.

$$\texttt{F1-score} = \frac{2 \times (\texttt{Recall} \times \texttt{Precision})}{(\texttt{Recall} + \texttt{Precision})} \tag{47}$$

In such equations, $n$ represents the total number of classified instances, Recall corresponds to the true positive rate (Equation 48), and Precision takes into account the number of instances correctly (true positive) or incorrectly (false positive) classified as positive (Equation 49).

$$\texttt{Recall} = \frac{TP}{(TP + FN)} \tag{48}$$

$$\texttt{Precision} = \frac{TP}{(TP + FP)} \tag{49}$$

We also used the Matthews correlation coefficient (MCC) [Matthews, 1975], which is similar to the well-known Pearson's correlation coefficient, with the main difference being the usage of the contingency matrix, as shown in Equation 50. The interpretation of this coefficient is based on the interval $[-1, +1]$, where $+1$ represents a perfect match between expected and predicted labels. When it is equal to 0, learning models are confirmed to provide random predictions. When values approach $-1$, a total disagreement is verified between expected and predicted labels.

$$\texttt{MCC} = \frac{(TP \times TN) - (FP \times FN)}{\sqrt{(TP + FP)(TP + FN)(TN + FP)(TN + FN)}} \tag{50}$$

Besides the results presented in the manuscript, we have also assessed the results by visually analyzing the chart produced by the ROC (Receiver Operating Characteristic) curve and computing the area under the curve (AUC). The ROC curve plots the true positive rate (TPR), Equation 48, against the false positive rate (FPR), Equation 51.

$$\texttt{FPR} = \frac{FP}{FP + TN} \tag{51}$$

### D.2 REGRESSION TASKS

Regarding node regression, we predict the passenger load at bus stops and stations over time. Due to the temporal dependency, we have trained and validated our analyses with a sliding window strategy instead of using cross-validation.

The first metric considered in our experiments was the mean squared error (MSE), shown in Equation, computing the differences between expected $(y_i)$ and predicted $(\hat{y}_i)$ values. We also used two related metrics: Root-Mean-Squared Error (RMSE) and Mean Absolute Error (MAE), both depicted in Equations 53 and 54.

$$\texttt{MSE} = \frac{1}{n} \sum^{n} (y_i - \hat{y}_i)^2 \tag{52}$$

$$\texttt{RMSE} = \sqrt{\frac{1}{n} \sum^{n} (y_i - \hat{y}_i)^2} \tag{53}$$

$$\texttt{MAE} = \frac{1}{n} \sum^{n} |y_i - \hat{y}_i| \tag{54}$$

Aiming to perform different interpretations, we have analyzed our results using the mean absolute percentage error (MAPE) and the coefficient of determination ($R^2$). MAPE, presented in Equation 3,

expresses the prediction errors in terms of percentages of actual values. $R^2$, presente in Equation 56, is a metric used to assess the proportion of variance in the expected values explained by the predicted ones, i.e., how closely the data matches the regression model. In this equation, $\bar{y}$ is the mean of the expected values.

$$\text{MAPE} = \frac{1}{n} \sum^{n} \left| \frac{y_i - \hat{y}_i}{y_i} \right| \tag{55}$$

$$R^2 = 1 - \frac{\sum^{n} (y_i - \hat{y}_i)^2}{\sum^{n} (y_i - \bar{y})^2} \tag{56}$$

### D.3 Loss Functions

The loss functions used to train ANN-based models depends on the analyzed task. For binary classification tasks, we have considered Binary Cross-Entropy (BCE) as defined in Equation 57, assuming the possible labels are $\{0, 1\}$. In this equation, $X^T$ is a set of instances selected to test the model, $y_i \in X^T$ is the class for the i-th tested instance, and $\hat{y}_i$ is its respective prediction.

$$\mathcal{L}_{\text{BCE}} = -\frac{1}{|X^T|} \sum_{i=1}^{|X^T|} y_i \log(\hat{y}_i) + (1 - y_i) \log(1 - \hat{y}_i) \tag{57}$$

In case of classification tasks with more classes, we have used Cross-Entropy (CE) defined in Equation 58, such that $C$ represents all possible classes.

$$\mathcal{L}_{\text{CE}} = -\frac{1}{|X^T|} \sum_{i=1}^{|X^T|} \sum_{c=1}^{|C|} y_{i,c} \log(\hat{y}_{i,c}) \tag{58}$$

Finally, GNNs trained to perform regression tasks can also use a special loss function as defined in Equation 59. In this equation, $\lambda$ is a hyper-parameter, and $L_{reg}$ is a normalization term to avoid overfitting issues.

$$loss = \|y_t - \hat{y}_t\| + \lambda L_{reg} \tag{59}$$

## E  Dataset and Benchmark Access

To support the maintenance of our contributions, we have shared all data and code in a GitHub repository (https://github.com/suntdataset/sunt.git). The data includes all raw and graph-based datasets. The code consists of scripts used to preprocess and transform the raw datasets, as well as models used as benchmarks. We emphasize that the code contains notebooks with commands organized into a structure that supports better comprehension of our work and the reproducibility of our results.

### E.1  Data Sheet

#### E.1.1  Motivation

1. **For what purpose was the dataset created?** *The SUNT dataset primarily aims to contribute to research in intelligent urban transportation fields, such as identifying transit patterns and developing machine learning solutions. Additionally, the dataset can benefit researchers from other domains interested in designing new theoretical and practical approaches, utilizing SUNT for validation purposes.*

2. **Who created the dataset and on behalf of which entity?** *The dataset was created by researchers from a public university (Omitted due to the double-anonymous requirements) in Brazil in collaboration with the companies Integra (Consortium of bus companies in Salvador) and NeoDados (Intelligent Solutions).*

3. **Who funded the creation of the dataset?** *This work was supported by Integra (Consortium of bus companies in Salvador), NeoDados (Intelligent Solutions), and CNPq (National Council for Scientific and Technological Development), Brazil, under grants (Omitted due to the double-anonymous requirements), and 68/2022 - Master's and Doctorate Program for Innovation - MAI/DAI.*

### E.1.2 DISTRIBUTION

1. **Will the dataset be distributed to third parties outside of the entity (e.g., company, institution, organization) on behalf of which the dataset was created?** *Yes, the dataset is entirely open to the public.*

2. **How will the dataset be distributed (e.g., tarball on website, API, GitHub)?** The dataset, codes, and models are already available at `https://github.com/suntdataset/sunt.git`.

3. **Have any third parties imposed IP-based or other restrictions on the data associated with the instances?** *No.*

4. **Do any export controls or other regulatory restrictions apply to the dataset or to individual instances?** *No.*

### E.1.3 MAINTENANCE

1. **Who will be supporting/hosting/maintaining the dataset?** *The GitHub will host the dataset. The authors will support and maintain the dataset as new data from the local public transport is collected over time.*

2. **How can the owner/curator/manager of the dataset be contacted (e.g., email address)?** *The owner/curator/manager(s) of the dataset can be contacted through the following emails (Omitted due to the double-anonymous requirements).*

3. **Is there an erratum?** *No. If any errors are discovered in the future, we will publish errata on the main page of our GitHub repository.*

4. **Will the dataset be updated (e.g., to correct labeling errors, add new instances, delete instances)?** *Yes, we will update the dataset as necessary to maintain its correctness and up-to-date information. Announcements will be made accordingly. We will publish the dataset versions on the main web page of the GitHub repository.*

5. **If the dataset relates to people, are there applicable limits on the retention of the data associated with the instances (e.g., were the individuals in question told that their data would be retained for a fixed period of time and then deleted)?** *The dataset does not contain personal information.*

6. **Will the older version of the dataset continue to be supported/hosted/maintained?** *Yes, all dataset versions will be continuously maintained, hosted, and improved with new information.*

7. **If others want to extend/augment/build on/contribute to the dataset, is there some mechanisms for them to do so?** *Once the dataset comprises information on the public transportation in Salvador, managed by local companies, all improvements will be administrated by the original researchers.*

### E.1.4 COMPOSITION

1. **What do the instances that comprise the dataset represent (e.g., documents, photos, people, countries)?** *Depending on the dataset, the instances may represent information about: vehicles, trips, temporal data, geospatial locations, users' boarding and alighting, and graph structures.*

2. **How many instances are there in total (of each type, if appropriate)?** *SUNT comprises 61 files of AVL-lines and AVL-vehicle from three bus companies. The size comprises, on daily average, 2.5 million entries in AVL-vehicle and 200 thousand entries in AVL-lines. Considering March, the AFC dataset contains a total of 36,851,307 passenger trips, while April has 38,238,530. In relation to March and April, LTI contains a total of 45,249,646 and*

*43,789,980 instances, respectively. GTFS is composed of: the routes file with 412 registers, trips file with 51,615 registers, stops file with 2,975 registers, and stop times file with 1,679,961 registers. In relation to the remaining dataset, created after the processing steps, the number instances may vary depending on how the temporal information is configured.*

3. **Does the dataset contain all possible instances or is it a sample of instances from a larger set?** *The dataset contains all instances considering the collected interval between March and April 2024.*

4. **Is there a label or target associated with each instance?** *The most important advantage of SUNT is the possibility of working on different learning tasks. In our benchmarks, we have created model focused on regression and classification (graph nodes and edges). However, depending on the users' objectives, labels can be created or adjusted to meed different learning requirements.*

5. **Is any information missing from individual instances?** *No.*

6. **Are there recommended data splits (e.g., training, development/validation, testing)?** *No.*

7. **Are there any errors, sources of noise, or redundancies in the dataset?** *No.*

8. **Is the dataset self-contained, or does it link to or otherwise rely on external resources (e.g., websites, tweets, other datasets)?** *The dataset is self-contained.*

9. **Does the dataset contain data that might be considered confidential?** *No.*

10. **Does the dataset contain data that, if viewed directly, might be offensive, insulting, threatening, or might otherwise cause anxiety?** *No.*

### E.1.5 COLLECTION PROCESS

1. **How was the data associated with each instance acquired?** *The integration process was performed using temporal and geospatial information of passengers, vehicles, trips, and stops/stations as discussed in Section 3 of our manuscript.*

2. **What mechanisms or procedures were used to collect the data (e.g., hardware apparatus or sensor, manual human curation, software program, software API)?** *We mostly used the ADCS (Automatic Data Collection System), which comprises AVL and AFC technologies, to collect the data.*

3. **Who was involved in the data collection process (e.g., students, crowdworkers, contractors), and how were they compensated (e.g., how much were crowdworkers paid)?** *Integra (Consortium of bus companies in Salvador) and NeoDados (Intelligent Solutions) have collected the data.*

4. **Does the dataset relate to people?** *No.*

5. **Did you collect the data from the individuals in question directly or obtain it via third parties or other sources (e.g., websites)?** *No.*

### E.1.6 USES

1. **Has the dataset been used for any tasks already?** *The dataset has been used only for the learning tasks reported in Section 4.*

2. **What (other) tasks could the dataset be used for?** *Further use of the dataset includes graph-based optimization problems, multi-objective optimization, and learning tasks using Concept Drift, among other possibilities. Please refer to Section A for detailed future works.*

3. **Is there anything about the composition of the dataset or the way it was collected and preprocessed/cleaned/labeled that might impact future uses?** *No, there is not. The repository stores raw and processed data, which adds to the flexibility of future uses.*

4. **Are there tasks for which the dataset should not be used?** *No.*

