# OpenReview forum: "Salvador Urban Network Transportation (SUNT): A Landmark Spatiotemporal Dataset for Public Transportation"
_ICLR.cc/2025/Conference — Submitted to ICLR 2025_

### Official Review · Reviewer_DZhi · 2024-10-28

**Soundness:** 3
**Presentation:** 4
**Contribution:** 3
**Rating:** 6
**Confidence:** 4

**Summary:**

The paper introduces SUNT, a comprehensive dataset from public transportation in Salvador, Brazil, with the aim of advancing urban mobility research. It covers detailed data on passengers, vehicles, routes, and geographic features, which makes it a useful resource for various fields like transportation planning, computational modeling, and even managing environmental impacts. By using Graph Neural Networks, the authors show that this dataset can effectively predict transit patterns. It’s great to see that they emphasize reproducibility and potential for future work, which is always crucial for building on research like this.

**Strengths:**

* It provides detailed data on passengers, vehicles, routes, and geographic features, making it valuable for fields like transportation planning, computational modeling, and environmental management.
* The use of Graph Neural Networks showcases the dataset’s potential in predicting transit patterns.
* The authors emphasize reproducibility and potential for future work, which is crucial for extending research.
* The paper is well-written, with an engaging introduction that sets the context and importance of the dataset effectively.
* The dataset is particularly impressive given the challenges of accessing AVL and AFC data from government sources, making it highly valuable for future research.

**Weaknesses:**

* Much of the content related to data collection in Section 3 is familiar to the transportation community, lacking novelty for that audience.
* The computer science community, on the other hand, would benefit more from a stronger focus on benchmarking.
* Figure 4 could be more informative by including additional benchmarking tests, as the current single-stop time series is limited in demonstrating the dataset’s predictive capabilities.
* There is a need for more discussion of comparable SOTA benchmarks across different tasks, which would facilitate broader research and model comparisons.

**Questions:**

* On page 5, line 260, the authors state that “these values were estimated by local specialists based on the passengers’ profiles and the transportation infrastructure in Salvador.” How are passenger profiles integrated into the origin-destination matrix at the regional or stop level?
* Regarding the time range of the dataset, was it specifically chosen for a reason, or is it simply the most recent data available during submission?
* In transportation research, studying mobility changes over longer periods—especially during disruptions like COVID-19—can provide additional insights. Could the dataset be extended to include a longer time range, possibly looking backward? Can the authors justify the current selection?

---

> ### Author Response · Authors · 2024-11-27
> **Rebuttal by Authors (Weaknesses 1, 2, 3, and 4)**
>
> **Weaknesses 1 (transportation community) and 2 (computer science community):** We appreciate the reviewer’s perspective. The design of our manuscript was carefully planned to align with the specific focus of the track to which it was submitted, "Datasets and Benchmarks." This approach was also inspired by previously accepted papers in the same track of prior conference editions. From our perspective, the steps used to create the dataset are complex tasks, rooted in current public transportation literature. As this literature evolves, readers can replicate our methodology to derive increasingly precise estimations. Moreover, based on feedback from other reviewers, we believe that the summary in Section 3 is essential for a comprehensive understanding of our dataset. For instance, comments such as the following support this view: “The paper is well-structured, with a clear introduction, detailed methodology, and comprehensive results section. The dataset construction process is rigorous and well-documented, with clear explanations of data cleaning steps and assumptions.” In recognition of the reviewer’s suggestion, we have clarified that a detailed discussion of the benchmarks is presented in Appendix C, "Learning Models."
>
> **Weaknesses 3 (additional benchmarking tests) and 4 (SOTA benchmarks):**
>
> We appreciate your valuable comment and agree that this research area has rapidly developed. Consequently, including more recently published methods is crucial to enhance our contribution and draw attention to our dataset. In response to your comment, we have invested a significant effort in identifying and executing recently published methods to analyze our dataset. The revised manuscript now incorporates the following recent references:
>
> Experiments with a recent GNN approach published in ICLR 2023, named A-DGN
>
> [1] Alessio Gravina, Davide Bacciu, and Claudio Gallicchio. Anti-symmetric DGN: a stable architecture for deep graph networks. In The Eleventh International Conference on Learning Representations, 2023.
>
> New experiments with Chronos, which is one of the state-of-the-art foundation models for time series analyses published in Transactions on Machine Learning Research 2024
>
> [2] Abdul Fatir Ansari et al. Chronos: Learning the language of time series. Transactions on Machine Learning Research, 2024. ISSN 2835-8856.
>
> A new experiment using Concept Drift method to detect changes in data streams using concepts published in IJCAI 2024
>
> [3] Yun Sing Koh et al. Time-evolving data science and artificial intelligence for advanced open environmental science (taiao) programme. International Joint Conferences on Artificial Intelligence (IJCAI), 2024.
>
> Prediction using Multivariate Time Series Forecasting
>
> [4] Lu HAN et al. SOFTS: Efficient Multivariate Time Series Forecasting with Series-Core Fusion. NeurIPS 2024.

---

> ### Author Response · Authors · 2024-11-27
> **Rebuttal by Authors (Questions 1, 2, and 3)**
>
> **Question 1:** The reviewer’s observation is absolutely valid. We have revised this section to provide a clearer discussion, as detailed below:
>
> As shown in Figure 1(d), a walking distance is deemed acceptable if it is limited to 1.1 km. Concerning the average velocity, Figure 1(e), and the trip time, Figure 1(f), all registers with values greater than 80 km/h and 2 hours are unconsidered. These values were estimated by local specialists based on the passengers’ usage patterns and the transportation infrastructure in Salvador. We emphasize that the reader can modify these values according to their needs once both raw and processed data are shared.
>
> **Question 2:** The construction of this dataset involves complex tasks, employing a boarding-to-boarding strategy to map users’ patterns and identify origin-destination locations. Our initial approach was to analyze the most recent data and work backward to the first day following the summer vacation, a period characterized by higher public transportation usage. However, as highlighted in our Data Sheet (see Appendices), our goal is to continuously expand the dataset with both past and future months.
>
> **Question 3:** We thank the reviewer for highlighting this important aspect of urban mobility. We fully agree that mobility patterns can vary significantly over extended or specific periods due to events such as the COVID-19 pandemic, holidays, or strikes. During the COVID-19 pandemic, for instance, a local law in Salvador prohibited the use of public transportation, reducing passenger loads to zero for a specific time interval. Even after restrictions were lifted, usage remained very limited, significantly lowering the maximum number of passengers. Nowadays, data from these periods are excluded from our daily analyses, as they affect all estimation models and do not provide meaningful insights for understanding current transportation patterns. To demonstrate that the SUNT dataset can be extended to include a longer time range and we can deal with mobility changes over these longer periods, the revised manuscript includes a new experiment focused on trends in data stream analyses. Specifically, we collected data from March to June 2024 and applied a concept drift detection method to identify shifts in passenger behavior over time. To illustrate the benefits of using such detectors, we selected a bus stop located near the Federal University of Bahia, the largest university in Salvador. In the figure below, the red vertical dashed lines indicate the moments when concept drifts were detected. The first drift corresponds to the start of the academic semester, marked by an increase in student use of public transportation. The second drift captures the onset of a strike, which disrupted classes and led to a decrease in passenger numbers. Due to space constraints, the detailed methodology and results of this investigation have been included in the appendix.
>
> [Experiment with Concept Drift to detect changes in passengers' behavior](https://github.com/suntdataset/sunt/blob/main/integration/concept_drift.png
> )

---

> > ### Comment · Reviewer_DZhi · 2024-12-02
> >
> > Thank you for the comments and the additional experiments. I will keep my scores.

---

> > > ### Author Response · Authors · 2024-12-02
> > >
> > > We sincerely thank the reviewer for their valuable time and effort in analyzing our responses to their comments. The reviewer’s suggestions, particularly regarding SOTA methods, allowed us to explore our dataset from new perspectives. Following these recommendations, we incorporated several advanced models into our experiments. Specifically, we:
> > >
> > > - Integrated a foundation model (Chronos) to predict our data as univariate time series;
> > > - Introduced a multivariate model (SOFTS) to investigate relationships between different transportation modes;
> > > - Applied a concept drift detection approach to identify significant changes in real-world data;
> > > - Conducted additional experiments, including predictions using Graph Neural Networks (GNNs) and outlier detection techniques.
> > >
> > > These contributions have significantly enriched our analyses, and we are grateful to the reviewer for inspiring these improvements and their collaboration in advancing our work.

---

### Official Review · Reviewer_nhDu · 2024-11-02

**Soundness:** 3
**Presentation:** 3
**Contribution:** 3
**Rating:** 6
**Confidence:** 3

**Summary:**

The paper introduces the Salvador Urban Network Transportation (SUNT) dataset, emphasizing its role in enhancing public transportation management in urban areas. It highlights the benefits of effective public transport, including improved mobility, economic development, reduced traffic congestion, and lower environmental impacts. The authors detail the methodologies employed in constructing the dataset, which integrates various spatiotemporal data sources to facilitate comprehensive analyses of public transportation systems.

**Strengths:**

Originality: The SUNT dataset presents a novel approach to public transportation analysis by integrating spatiotemporal data, which has not been extensively explored in prior studies. This originality lies in both the dataset's construction and its potential applications across various research domains.
Quality: The methodology employed in creating the dataset is rigorous, ensuring that the data is reliable and applicable for future studies. The thorough documentation of processes adds to its overall quality.
Clarity: The paper is well-organized, making it accessible to readers with varying levels of expertise. Key concepts and methodologies are clearly articulated, which aids in comprehension.
Significance: The implications of this work are significant, as the dataset addresses important challenges in urban transportation management. It has the potential to influence policy decisions and improve transportation systems in urban settings.

**Weaknesses:**

Data Sourcing and Preprocessing: Further details on data sourcing and integration are recommended, specifically covering how each source (e.g., AVL, AFC, GTFS) was merged and standardized, along with measures for data consistency. The handling of missing data and inconsistencies in time or location stamps, particularly during peak traffic times, could be clarified to enhance reproducibility for future researchers.
Experimental Validation and Real-World Applications: The paper could be strengthened by including case studies and examples of real-world scenarios, particularly highlighting the dataset's performance under various conditions, such as peak versus non-peak hours or seasonal variations. This would help substantiate the practical value and adaptability of the dataset in different settings.
Use of the Latest Models: The selection of models, while robust, does not appear to include some recent advancements. The latest models utilized date up to 2022. A few state-of-the-art spatiotemporal and multimodal GNN architectures developed after 2022 were not explored, potentially limiting the demonstration of the dataset's full capabilities. Indicating any constraints or considerations regarding model selection could provide helpful context for future research.
Visualization and Presentation of Results: Additional visual aids could enhance comprehension of complex data trends, such as high-passenger-load areas, traffic flow, or frequent delay points. Developing an interactive dashboard or map to display variations in model performance across geographic regions could also illustrate prediction accuracy or confidence levels in different network zones.

**Questions:**

Dataset Limitations: Clarifying the specific limitations encountered during the creation of the SUNT dataset, such as constraints in data collection or accuracy, would help understand their impact on the dataset’s applicability in various urban contexts.
Comparative Analysis with Existing Datasets: A comparison with other spatiotemporal datasets could strengthen the paper by positioning the SUNT dataset within the broader field of urban transportation research, highlighting where it advances the state of the art or addresses gaps in existing datasets.
Use of State-of-the-Art Models: A more detailed discussion on how the selected models align with recent advancements in spatiotemporal data modeling, including post-2022 innovations, would enhance the paper’s relevance and demonstrate the dataset’s adaptability to current methodologies.

---

> ### Author Response · Authors · 2024-11-27
> **Rebuttal by Authors (Weaknesses 1 and 2)**
>
> **Weakness 1(Data Sourcing and Preprocessing):** This is an excellent and important question raised by the reviewer. Indeed, integrating all data sources into our dataset is a complex task, as detailed in Section 3 (Sunt Dataset Construction) and Appendix B. While we have utilized the most recommended approaches to construct our dataset, we have also shared all raw data. This provides flexibility for creating additional features or applying more precise methodologies in future research. Regarding missing data, this is not an issue commonly encountered in this type of research, as discussed in the literature. Our dataset is based on data collected from passengers who pass through the turnstiles, and it is not possible to make a trip without registering it. In the rare event of a failure in the fare collection system, it is immediately replaced by the local public transportation management company. Local policies mandate that buses cannot operate without a functioning fare collection system. For this reason, our dataset does not contain missing data, as the reviewer can verify through our GitHub repository.
>
> **Weakness 2 (Experimental Validation and Real-World Applications):** The proposed dataset originates from a real-world public transportation network in Salvador, Brazil. It is important to emphasize that the dataset contains no synthetic or simulated data. All experiments were meticulously designed to address actual challenges faced by the local public transportation system. Furthermore, significant effort was dedicated to presenting new case studies and real-world examples to highlight the potential applications of the proposed dataset in practical scenarios.
>
> 1. Outlier Detection: This experiment aims to identify atypical traffic behavior in urban settings, e.g., those influenced by weather conditions.
>
> 2. Behavior Change Detection: We employ concept drift techniques to recognize shifts in passenger behavior over time.
>
> 3. Multi-Modal Analysis: This experiment focuses on prediction tasks that utilize data from different transportation modes included in the dataset (Bus, BRT, and Subway).
>
> 4. Max Load Detection: The objective here is to identify locations with the highest passenger concentrations during specific time intervals.
>
> 5. Node and Edge Classification Tasks: These tasks involve classifying different nodes and edges within the transportation network.
>
> 6. Node Regression Tasks: This involves regression analyses on nodes to derive meaningful insights.
>
> 7. Integration with other datasets: Finally, we explore the potential of integrating the proposed dataset with datasets from other contexts, such as educational institutions.
>
> These experiments collectively highlight the versatility and applicability of the dataset in enhancing urban transportation planning.

---

> ### Author Response · Authors · 2024-11-27
> **Rebuttal by Authors (Weaknesses 3, 4 and 5 - Question 1)**
>
> **Weaknesses 3 (Use of the Latest Models) and 4 (Use of State-of-the-Art Models):** We appreciate your valuable comment and agree that this research area has rapidly developed. Consequently, including more recently published methods is crucial to enhance our contribution and draw attention to our dataset. In response to your comment, we have invested a significant effort in identifying and executing recently published methods to analyze our dataset. The revised manuscript now incorporates the following recent references:
> - Experiments with a recent GNN approach published in ICLR 2023, named A-DGN
>
> [1] Alessio Gravina, Davide Bacciu, and Claudio Gallicchio. Anti-symmetric DGN: a stable architecture
> for deep graph networks. In The Eleventh International Conference on Learning Representations, 2023.
>
> - New experiments with Chronos, which is one of the state-of-the-art foundation models for time series analyses published in Transactions on Machine Learning Research 2024
>
> [2] Abdul Fatir Ansari et al. Chronos: Learning the language of time series. Transactions on Machine Learning Research, 2024. ISSN 2835-8856.
>
>  - A new experiment using Concept Drift method to detect changes in data streams using concepts published in IJCAI 2024
>
> [3] Yun Sing Koh et al. Time-evolving data science and artificial intelligence for advanced open environmental science (taiao) programme. International Joint Conferences on Artificial Intelligence (IJCAI), 2024.
>
> - Prediction using Multivariate Time Series Forecasting
>
> [4] Lu HAN et al. SOFTS: Efficient Multivariate Time Series Forecasting with Series-Core Fusion. NeurIPS 2024.
>
> **Weaknesses 5 (Visualization and Presentation of Results):** We thank the reviewer for calling our attention to this important issue. Coincidently, another reviewer suggested the same analyses. Therefore, by looking for qualitative insights to support transportation planners and policymakers, we have developed an application capable of identifying the bus stops with the highest maximum load for each line over time. According to transportation experts collaborating with us on this project, a particularly valuable feature is the ability to detect stops that experience the highest passenger concentrations during specific time intervals. This information enables more effective planning and distribution of buses, ultimately improving service to meet the needs of the population more efficiently. In the following figure, we have selected one of this lines and analyzed the max-load stops in four different time intervals: i) rush time at 7 a.m., and 4 p.m., (A) and (B), respectively; and ii) off-peak time at 10 a.m., and 3 p.m., (C) and (D), respectively. By analyzing this maps, one might be able to understand how the maximum load stops are varying within different time intervals. This information facilitates more effective bus planning and distribution, enhancing service delivery to better meet the population's needs. More details about how these maps were produced are discussed in our manuscript.
>
> [Plot with Max Load over time](https://github.com/suntdataset/sunt/blob/main/outputs/max_load_times.jpeg)
>
> **Question 1 (Dataset Limitations):**
> In our manuscript, there is a subsection in our conclusions highlighting all limitations and challenges faced during the dataset creation. In the new manuscript version, we’ll include some questions raised by all reviewers. In summary, we discuss that although we have sought the best-known models in the literature, their architectures and parametrization can be individually analyzed to improve the results. Secondly, we explored a limited variation of attributes available in our dataset. Learning from other attributes or broad combinations of them is also possible. Thirdly, we have analyzed three learning tasks based on node and edge classification and regression. Researchers can also use different attributes and their transformations as targets. Fourthly, other graph structures may provide important information, mainly varying the edges' weights. Finally, we have predefined some parameters related to the application, such as 5-minute intervals and 1.1km walking distance, due to the particularities of our local scenario. Such definitions may not attend other research. However, by operating our shared scripts used to create SUNT, readers can use the raw datasets to redefine them according to their needs.

---

> ### Author Response · Authors · 2024-11-27
> **Rebuttal by Authors (Question 2 )**
>
> **Question 2 (Comparative Analysis with Existing Datasets):** We thank the reviewer for raising this important question. Our dataset provides detailed information about a complex network of interconnected stops and stations, covering multiple modes of public transportation (Bus, BRT, and Subway). It also includes data on millions of passengers from a large city, which have been meticulously organized, processed, and documented. To the best of our knowledge, no similar dataset is publicly available.
>
> After conducting an extensive search for similar studies, we observed that related work is often published without fully sharing the underlying data. The papers in the related work section were selected to illustrate this limitation. Despite recent advances in data analysis and intelligent public transportation systems, research in this area remains constrained to groups that have exclusive access to the data. To address the reviewer’s question, we have included the following comparison table, highlighting the significance of our dataset for the scientific community.
>
> | **Dataset**       | **#Nodes** | **#Edges** | **Period**                                                                 | **Shortest Time Interval** |
> |--------------------|------------|------------|-----------------------------------------------------------------------------|----------------------------|
> | METR-LA           | 207        | 2,369      | March 1, 2012 to June 30, 2012                                             | 5 minutes                 |
> | PeMS-BAY          | 325        | 1,515      | January 1, 2017 to May 31, 2017                                            | 5 minutes                 |
> | TaxiBJ         | --         | --         | July 1, 2013 to October 30, 2013                                           | 30 minutes                |
> |                    |            |            | March 1, 2014 to June 30, 2014                                             |                            |
> |                    |            |            | March 1, 2015 to June 30, 2015                                             |                            |
> |                    |            |            | November 1, 2015 to April 10, 2016                                         |                            |
> | BikeNYC           | 50         | --         | April 1, 2014 to September 30, 2014                                        | 1 hour                    |
> | Shanghai Metro    | 288        | 958        | July 1, 2016 to September 30, 2016                                         | 15 minutes                |
> | Hangzhou Metro    | 80         | 248        | January 1, 2019 to January 31, 2019                                        | 15 minutes                |
> | Beijing Metro     | 276        | --         | February 29, 2016 to April 3, 2016                                         | --                        |
> | Chongqing Metro   | 170        | --         | March 1, 2019 to March 31, 2019                                            | 15 minutes                |
> | Stockholm County  | --         | --         | --                                                                          | --                        |
> | UVDS              | 104        | --         | Three months                                                               | 5 minutes                 |
> | **SUNT**          | **2,871**  | **4,526**  | **March 1, 2024 to October 31, 2024**                                      | **< 1 minute**            |
>
> *Table: Characteristics of SUNT Compared to Common Graph Datasets*
>
> The missing information (-) in this table reflects the fact that several datasets commonly used in research articles are partially described in the publications and are not freely shared in public repositories with the same level of detail as ours. For example, we have noticed that information about the number of nodes, edges, or specific temporal intervals is often unavailable. As a result, researchers face challenges in reproducing experiments or fully understanding the scope and limitations of the datasets referenced in these studies. On the other hand, we offer the SUNT dataset, which stands out as an exception, offering 2,871 nodes, 4,526 edges, and a temporal granularity of less than one minute, with an in-depth dataset construction, which are pivotal for addressing key deficiencies identified in recent studies on learning benchmarks. SUNT offers a robust foundation for developing models that can learn complex spatiotemporal patterns and adapt to rapidly changing conditions.
>
> Rather than keeping this valuable dataset confined to our laboratory, we believe that sharing it through a prestigious venue like ICLR will significantly advance research across various fields. By enabling other research groups to derive insights from this dataset, we anticipate an acceleration of valuable findings that can ultimately benefit the local population.

---

### Official Review · Reviewer_yCGo · 2024-11-03

**Soundness:** 3
**Presentation:** 3
**Contribution:** 2
**Rating:** 5
**Confidence:** 3

**Summary:**

The paper presents the Salvador Urban Network Transportation (SUNT) dataset, which is structured into raw data and a graph that connects various transportation data while respecting geospatial and temporal constraints. It includes four raw datasets collected over five months, detailing information about vehicles, passengers, and stops/stations, as well as a preprocessed dataset for complex network analysis. The authors provide benchmark models using Graph Neural Networks for classification and regression tasks.

**Strengths:**

+ The paper provides a dataset that includes extensive information about the public transportation system in Salvador, Brazil.
+ The authors provide detailed information on the datasets in the GitHub repo.
+ Very detailed information has been provided in the appendix.This could be useful to beginners in this domain.

**Weaknesses:**

- Limited to the public transportation system of only Salvador which may limit the overall impact of this work. Providing more detailed case studies or examples of how the proposed dataset can help resolve the current limitations in urban research could illustrate its impact.
- Although a bunch of methods have been included in the benchmark, there are only 2 methods from 2022 and no method from 2023/2024. Considering the fast development in the spatiotemporal AI domain, it would be more convincing to include more recent methods.
- For the first node classification task, all the methods performed pretty well. It feels like a solved problem. Why use this task for the benchmark?
- Including more challenging settings would be interesting, such as anomaly detection, long-term forecasting etc.

**Questions:**

1. What are the other potential applications of using the 4 raw datasets? In the paper, the 4 raw datasets are used to form an OD dataset. What are the other potentials?
2. In the data collection and dataset formulation process, are there any suggestions/feedback/insights from domain experts such as traffic planners and/or transport operators?

**Details Of Ethics Concerns:**

No ethics concerns. The authors have discussed the potential concerns.

---

> ### Author Response · Authors · 2024-11-27
> **Rebuttal by Authors (Weaknesses 1 and 2)**
>
> **Weakness 1:** We would like to thank the reviewer for the valuable comment. This paper aims to introduce a dataset focused on urban traffic in Salvador, which is the fifth largest city in Brazil, with a population exceeding 2.5 million residents. It is important to emphasize that, although we have developed important knowledge about how to organize and share such datasets, all information about passengers is private and is owned by the public transportation companies operating in the respective regions. After considerable effort, we successfully convinced the local public transportation company to share this data, which is a considerable advance to the state of the art. By making this passenger data publicly available (while ensuring privacy protections), we are certainly paving the way to accelerate research efforts to enhance the public transportation sector. To the best of our knowledge, although it is possible to find other datasets related to traffic and public transportation routes, none provide a similar level of detail by associating passenger information as our proposed dataset. Collecting data from Salvador's public transport system presents significant challenges due to data volume and complexity. This proposed dataset is the result of a collaboration between the authors and the public transport consortium company. It is important to note that incorporating urban transport data from other cities would create a new dataset outside the scope of this paper, because each city has its characteristics of urban traffic, public transport consortium company, population size, etc. We hope this work will encourage the development of new datasets for public transportation in other major cities in the future.
>
> To demonstrate the potential applications of the proposed dataset, as suggested by the reviewer, in supporting transportation planning efforts, we have included the following experiments in the revised manuscript:
>
> 1. Outlier Detection: This experiment aims to identify atypical traffic behavior in urban settings, e.g., those influenced by weather conditions.
>
> 2. Behavior Change Detection: We employ concept drift techniques to recognize shifts in passenger behavior over time.
>
> 3. Multi-Modal Analysis: This experiment focuses on prediction tasks that utilize data from different transportation modes included in the dataset (Bus, BRT, and Subway).
>
> 4. Max Load Detection: The objective here is to identify locations with the highest passenger concentrations during specific time intervals.
>
> 5. Node and Edge Classification Tasks: These tasks involve classifying different nodes and edges within the transportation network.
>
> 6. Node Regression Tasks: This involves regression analyses on nodes to derive meaningful insights.
>
> 7. Integration with other datasets: Finally, we explore the potential of integrating the proposed dataset with datasets from other contexts, such as educational institutions.
>
> These experiments collectively highlight the versatility and applicability of the dataset in enhancing urban transportation planning.
>
> **Weakness 2:** We appreciate your valuable comment and agree that this research area has rapidly developed. Consequently, including more recently published methods is crucial to enhance our contribution and draw attention to our dataset. In response to your comment, we have invested a significant effort in identifying and executing recently published methods to analyze our dataset. The revised manuscript now incorporates the following recent references:
> - Experiments with a recent GNN approach published in ICLR 2023, named A-DGN
>
> [1] Alessio Gravina, Davide Bacciu, and Claudio Gallicchio. Anti-symmetric DGN: a stable architecture
> for deep graph networks. In The Eleventh International Conference on Learning Representations, 2023.
>
> - New experiments with Chronos, which is one of the state-of-the-art foundation models for time series analyses published in Transactions on Machine Learning Research 2024
>
> [2] Abdul Fatir Ansari et al. Chronos: Learning the language of time series. Transactions on Machine Learning Research, 2024. ISSN 2835-8856.
>
>  - A new experiment using Concept Drift method to detect changes in data streams using concepts published in IJCAI 2024
>
> [3] Yun Sing Koh et al. Time-evolving data science and artificial intelligence for advanced open environmental science (taiao) programme. International Joint Conferences on Artificial Intelligence (IJCAI), 2024.
>
> - Prediction using Multivariate Time Series Forecasting
>
> [4] Lu HAN et al. SOFTS: Efficient Multivariate Time Series Forecasting with Series-Core Fusion. NeurIPS 2024.

---

> ### Author Response · Authors · 2024-11-27
> **Rebuttal by Authors (Weaknesses 3 and 4)**
>
> **Weakness 3:** This is an excellent question raised by the reviewer. First, before addressing practical issues related to public transportation in Salvador, we designed an experimental setup with a set of applications to demonstrate the relevance of our dataset for various learning tasks. By presenting strong results, we highlight that our dataset contains meaningful patterns and that learning tasks can indeed be successfully executed. Regarding this specific experiment, we chose to discretize passenger loading into "High" and "Low" categories based on the average. While introducing subscales can extract more detailed information, it may also increase imprecision in the results. This trade-off is worth discussing as it underscores the flexibility of our dataset and demonstrates how different learning tasks can be derived from it.
>
> **Weakness 4:** We thank the reviewer for suggesting this valuable new application, which significantly enhances the relevance of our dataset. To address the reviewer’s feedback, we incorporated an outlier detection task using the Isolation Forest algorithm. The figure below presents a time series of travel times for a specific edge in our graph (route connecting two stops), with data collected from April 1st to April 6th, 2024. The figure below depicts a time series of travel times for a specific edge in our graph (a route connecting two bus stops), based on data collected from April 1st to April 6th, 2024. The x-axis represents the monitored days, while the y-axis indicates the total time taken by buses to complete this route. The analysis highlights outliers detected on April 4th, 2024, attributed to a weather event that severely disrupted traffic flow across the city. The revised manuscript provides a detailed description of the Isolation Forest algorithm and a discussion of the results. We believe that the inclusion of outlier detection adds substantial value to our work by tackling a challenging scenario and demonstrating the versatility of our dataset for applying diverse machine-learning techniques.
>
> [Example of outlier detections](https://github.com/suntdataset/sunt/blob/main/outputs/outlier_detection.png)
>
> Still focused on answering the reviewers’ question, we trained a foundation model to predict observations in a time series, employing a long-term prediction strategy as recommended. The results are presented in the following figure. It is important to note that, while the predictions may appear to cover a short time span, they actually encompass 288 observations, given that our data is structured in 5-minute intervals. We sincerely thank the reviewer for suggesting this task, which allowed us to extend the prediction horizon and incorporate a highly relevant model. This approach leverages transformer-based language model architectures, a cutting-edge methodology in the field.
>
> [Prediction using Chronos](https://github.com/suntdataset/sunt/blob/main/integration/forecasting_chronos.png)
>
> Finally, expanding our analyses, we have included another experiment with a time series composed of observations from all available data. Specifically, we collected data from March to June 2024 and applied a concept drift detection method to identify shifts in passenger behavior over time. To illustrate the benefits of using such detectors, we selected a bus stop located near the Federal University of Bahia, the largest university in Salvador. In the figure below, the red vertical dashed lines indicate the moments when concept drifts were detected. The first drift corresponds to the start of the academic semester, marked by an increase in student use of public transportation. The second drift captures the onset of a strike, which disrupted classes and led to a decrease in passenger numbers. Due to space constraints, the detailed methodology and results of this investigation have been included in the appendix, under the section “Supplementary Results”.
>
> [Example using Concept Drift](https://github.com/suntdataset/sunt/blob/main/integration/concept_drift.png)

---

> ### Author Response · Authors · 2024-11-27
> **Rebuttal by Authors (Questions 1 and 2)**
>
> **Question 1:** Thank you for your insightful question. The four raw datasets provide a range of data within the realm of urban public transport. Professionals in urban traffic planning can leverage these datasets to assess various related issues, including the usability of the public transport network, passenger capacity on bus lines and at stations, as well as metrics related to fuel consumption and carbon footprint. This information is essential for enhancing the efficiency of urban transport systems and promoting the responsible use of resources. We are optimistic that the dataset presented in our work will be useful for researchers and professionals interested in urban mobility issues.
>
> **Question 2:** One of the co-authors of this work is a specialist in urban public transportation and played a key role in supervising the dataset's construction, ensuring the accuracy of the analyses, and validating the results.

---

### Official Review · Reviewer_wuhs · 2024-11-08

**Soundness:** 2
**Presentation:** 3
**Contribution:** 2
**Rating:** 5
**Confidence:** 4

**Summary:**

This paper introduces SUNT (Salvador Urban Network Transportation), a new large-scale spatiotemporal dataset for public transportation research. The key contributions are:

1. A comprehensive dataset containing information from about 710,000 passengers and 2,000 vehicles operating on nearly 400 lines and 3,000 stops/stations in Salvador, Brazil, collected over 5 months in 2024.

2. The dataset is provided in two forms:
   - Raw data from multiple sources including automatic vehicle location, fare collection, transit feed specifications, and local trip information.
   - A processed graph-based dataset integrating spatial and temporal information.

3. Benchmarking results using various machine learning models, including Graph Neural Networks, for tasks such as node classification, edge classification, and node regression.

4. Pre-trained models and code to reproduce the benchmarking experiments.

5. Detailed methodology for constructing the dataset, including techniques for inferring passenger boarding and alighting locations.

6. Discussion of potential applications and future research directions enabled by this dataset.

The authors emphasize that SUNT allows for modeling real-world urban mobility data in different ways, reproducing their results, improving on the baseline models, and investigating various open problems in transportation research. They provide the full dataset, code, and models to facilitate further research in this area.

**Strengths:**

Originality:
- The SUNT dataset is a novel and comprehensive public transportation dataset, combining data from multiple sources (AVL, AFC, GTFS, LTI) for a major Brazilian city over a 5-month period.
- The authors develop an innovative methodology to infer passenger alighting locations, which is a common challenge in transportation datasets where only boarding information is directly captured.
- The integration of spatial, temporal, and quantitative data into a complex network representation can enable rich analysis.

Quality:
- The dataset construction process is rigorous and well-documented, with clear explanations of data cleaning steps and assumptions.
- The benchmarking experiments are thorough, testing multiple graph neural network architectures on three distinct tasks (node classification, edge classification, node regression).
- The authors provide pre-trained models and code, enhancing reproducibility and enabling further research.

Clarity:
- The paper is well-structured, with a clear introduction, detailed methodology, and comprehensive results section.
- Figures and tables effectively illustrate the dataset construction process and benchmark results.
- The authors provide extensive supplementary materials, including appendices with additional details on models, metrics, and future research directions.

Significance:
- SUNT addresses a gap in publicly available transportation datasets, providing a large-scale, multi-modal dataset that can drive research in urban mobility and intelligent transportation systems.
- The dataset's comprehensive nature allows for diverse applications, from traffic prediction to route optimization and environmental impact assessment.

**Weaknesses:**

1. Limited comparison to existing datasets:
The authors mention several existing transportation datasets (e.g., TaxiBJ, BikeNYC, METR-LA) but don't provide a detailed comparison of SUNT's features and capabilities relative to these datasets. A table comparing key attributes (e.g., data types, temporal/spatial resolution, coverage) would help readers understand SUNT's unique contributions and advantages.

2. Lack of error analysis in data processing:
The paper describes a complex process for inferring passenger alighting locations, but doesn't provide an analysis of potential errors or biases introduced by this method. An evaluation of the accuracy of these inferences, perhaps through a small-scale validation study, would strengthen confidence in the dataset's quality.

3. Limited exploration of temporal aspects:
While the dataset includes 5 months of data, the benchmarking experiments focus on only a few specific days. A more comprehensive analysis exploring temporal patterns (e.g., weekday vs. weekend, seasonal variations) would better demonstrate the dataset's potential for longitudinal studies.

4. Narrow range of benchmark tasks:
The benchmarking focuses on three tasks (node classification, edge classification, node regression). Expanding this to include tasks more directly relevant to transportation planning (e.g., demand prediction, route optimization) would better showcase the dataset's practical utility.

5. Lack of multi-modal analysis:
Although the dataset includes information from buses, BRT, and subway systems, the benchmarks don't explicitly demonstrate how these different modes can be analyzed together. Showcasing multi-modal analyses would highlight a key strength of the dataset.

6. Limited discussion of privacy considerations:
While the paper mentions that privacy was respected in data collection, there's little discussion of potential re-identification risks or mitigation strategies employed in the public release of the dataset.

7. Absence of qualitative insights:
The paper focuses heavily on quantitative benchmarks but doesn't provide qualitative insights into transportation patterns or urban dynamics that the dataset reveals. Including some case studies or visualizations of interesting patterns would make the paper more compelling to transportation planners and policymakers.

8. Lack of scalability analysis:
Given the large scale of the dataset, an analysis of computational requirements for processing and analyzing the full dataset would be valuable for researchers planning to use it.

9. Limited discussion of data quality and potential biases:
While the paper describes the data collection process, it doesn't deeply address potential biases or quality issues in the raw data sources (e.g., GPS errors, fare evasion).

**Questions:**

Same as the weaknesses

---

> ### Author Response · Authors · 2024-11-27
> **Rebuttal by Authors (Comment 1)**
>
> **Comment 1:** We thank the reviewer for raising this important question. Our dataset provides detailed information about a complex network of interconnected stops and stations, covering multiple modes of public transportation (Bus, BRT, and Subway). It also includes data on millions of passengers from a large city, which have been meticulously organized, processed, and documented. To the best of our knowledge, no similar dataset is publicly available.
>
> After conducting an extensive search for similar studies, we observed that related work is often published without fully sharing the underlying data. Despite recent advances in data analysis and intelligent public transportation systems, research in this area remains constrained to groups that have exclusive access to the data. To address the reviewer’s question, we have included the following comparison table in the appendices, highlighting the significance of our dataset for the scientific community.
> | **Dataset**       | **#Nodes** | **#Edges** | **Period**                                                                 | **Shortest Time Interval** |
> |--------------------|------------|------------|-----------------------------------------------------------------------------|----------------------------|
> | METR-LA           | 207        | 2,369      | March 1, 2012 to June 30, 2012                                             | 5 minutes                 |
> | PeMS-BAY          | 325        | 1,515      | January 1, 2017 to May 31, 2017                                            | 5 minutes                 |
> | TaxiBJ         | --         | --         | July 1, 2013 to October 30, 2013                                           | 30 minutes                |
> |                    |            |            | March 1, 2014 to June 30, 2014                                             |                            |
> |                    |            |            | March 1, 2015 to June 30, 2015                                             |                            |
> |                    |            |            | November 1, 2015 to April 10, 2016                                         |                            |
> | BikeNYC           | 50         | --         | April 1, 2014 to September 30, 2014                                        | 1 hour                    |
> | Shanghai Metro    | 288        | 958        | July 1, 2016 to September 30, 2016                                         | 15 minutes                |
> | Hangzhou Metro    | 80         | 248        | January 1, 2019 to January 31, 2019                                        | 15 minutes                |
> | Beijing Metro     | 276        | --         | February 29, 2016 to April 3, 2016                                         | --                        |
> | Chongqing Metro   | 170        | --         | March 1, 2019 to March 31, 2019                                            | 15 minutes                |
> | Stockholm County  | --         | --         | --                                                                          | --                        |
> | UVDS              | 104        | --         | Three months                                                               | 5 minutes                 |
> | **SUNT**          | **2,871**  | **4,526**  | **March 1, 2024 to October 31, 2024**                                      | **< 1 minute**            |
>
> *Table: Characteristics of SUNT Compared to Common Graph Datasets*
>
> The missing information (-) in this table reflects the fact that several datasets commonly used in research articles are partially described in the publications and are not freely shared in public repositories with the same level of detail as ours. For example, we have noticed that information about the number of nodes, edges, or specific temporal intervals is often unavailable. As a result, researchers face challenges in reproducing experiments or fully understanding the scope and limitations of the datasets referenced in these studies. On the other hand, we offer the SUNT dataset, which stands out as an exception, offering 2,871 nodes, 4,526 edges, and a temporal granularity of less than one minute, with an in-depth dataset construction, which are pivotal for addressing key deficiencies identified in recent studies on learning benchmarks. SUNT offers a robust foundation for developing models that can learn complex spatiotemporal patterns and adapt to rapidly changing conditions.
>
> Rather than keeping this valuable dataset confined to our laboratory, we believe that sharing it through a prestigious venue like ICLR will significantly advance research across various fields, including time series analysis, signal processing, machine learning, search, optimization, and graph theory. By enabling other research groups to derive insights from this dataset, we anticipate an acceleration of valuable findings that can ultimately benefit the local population.

---

> ### Author Response · Authors · 2024-11-27
> **Rebuttal by Authors (Comments 2 and 3)**
>
> **Comment 2:** We understand the reviewer’s concern regarding potential errors in data processing. In public transportation systems, it is common for passengers to register only their boarding, and, due to ethical considerations, it is not possible to personally identify them or determine their exact alighting points. The available data only includes the timestamp of when a passenger’s card is used onboard buses or trains. To address this limitation, we explored a boarding-to-boarding approach to infer passengers’ trips, which we have adopted in our paper. However, we would like to highlight that we are sharing all raw data. This ensures that, if more precise methods are developed in the future to reduce such errors or biases, they can be readily applied to our dataset. Finally, as outlined in our manuscript, we identify a percentage of passengers for whom there is no record of subsequent boardings, making it impossible to estimate their alighting points. In the manuscript, we address this issue and discuss how considering different distributions can help mitigate its impact, as usually considered in the public transportation literature.
>
> **Comment 3:** We thank the reviewer for highlighting the importance of incorporating larger time series in our analysis. To address this valuable feedback, we have conducted a new experiment focused on trends in data stream analyses, which has been included in the manuscript. Specifically, we collected data from March to June 2024 and applied a concept drift detection method to identify shifts in passenger behavior over time. To illustrate the benefits of using such detectors, we selected a bus stop located near the Federal University of Bahia, the largest university in Salvador. In the figure below, the red vertical dashed lines indicate the moments when concept drifts were detected. The first drift corresponds to the start of the academic semester, marked by an increase in student use of public transportation. The second drift captures the onset of a strike, which disrupted classes and led to a decrease in passenger numbers. Due to space constraints, the detailed methodology and results of this investigation have been included in the appendix, under the section “Supplementary Results”.
>
> LINK: [Concept Drift Notebook](https://github.com/suntdataset/sunt/blob/main/integration/concept_drift_fft.ipynb)
>
> In these experiments, we selected two well-known events to evaluate the feasibility of detecting changes. After implementing this solution derived from the reviewer’s suggestion, we incorporated the concept drift detector into our daily workflows. This enables the system to provide real-time alerts when new events occur, thereby supporting decision-makers more effectively.

---

> > ### Author Response · Authors · 2024-11-27
> > **Rebuttal by Authors (Comments 4 and 5)**
> >
> > **Comment 4:** We fully agree with the reviewer. To address this limitation, we not only applied an AI-based approach to detect changes in traffic dynamics using Concept Drift detection but also included a new experiment in the appendices focused on route optimization. The results demonstrate how the proposed dataset can support different applications, such as optimizing vehicle schedules to enhance user experience and mitigate traffic impact. As discussed by Ceder (2016), one of the major challenges in transit service is determining the most suitable frequency (vehicles/hour) for each route based on time-of-day and day type. Achieving this goal starts with calculating the maximum number of onboard passengers along the entire route within a given time period, which is essential for creating precise transit timetables.
> >
> > Details of this computation are provided in our manuscript, and we have included the following table to illustrate the significance of these results. Typically, when passenger data at stops/stations and along routes is unavailable, it becomes difficult to optimize bus services effectively. For example, during a rush hour interval (e.g., 8 AM–9 AM), 9 buses might be scheduled to serve passengers traveling from stop A to stop B, passing through intermediate stops. Without an estimate of maximum passenger load, all buses would need to return from stop B to stop A via the same route, stopping at all intermediate stations. Such a strategy has some problems: it wastes time and fuel, besides delaying the arriving time at A. Considering A is a neighborhood and B downtown, the amount of passengers from B to A is considerably lower during this rush time. Knowing the optimal number of buses required for the return trip and their appropriate schedules can significantly mitigate these issues.
> > Our manuscript (appendices) provides detailed explanations of how the maximum load is computed and describes strategies such as Random, Next, and Nearest Neighbors to address these challenges.
> >
> > [Image with Normal and Express Planning](https://github.com/suntdataset/sunt/blob/main/integration/normal_express.png)
> >
> > **Comment 5:** The reviewer has raised an important question regarding the potential for multimodal applications of our dataset. In all previous analyses, we independently examined buses, subways, and BRT systems. We agree that integrating these data into a single analysis enhances the significance of our dataset. To address this, we have included a new experiment utilizing multivariate time series analysis. The following figure (extracted from Google Maps) illustrates a region of Salvador where passengers from all three modes of transportation are aggregated.
> >
> > [Image with different transportation modes](https://github.com/suntdataset/sunt/blob/main/integration/bus_types.png)
> >
> > The following figure illustrates the time series utilized to analyze all transportation modes. The results of this analysis will be incorporated into our manuscript, specifically in Appendix A (Supplementary Results).
> >
> > [Time series for all transportation modes](https://github.com/suntdataset/sunt/blob/main/outputs/brt_bus_subway_boardings.png)
> >
> > In the following plot, we show bus load predictions produced after modeling all time series [1] . In our manuscript, we discuss how this type of application is still considered an open problem in time series analyses.
> >
> > [Prediction using Multivariate Time Series Forecasting](https://github.com/suntdataset/sunt/blob/main/integration/forecasting_mutimodal.png)
> >
> > [1] HAN, Lu et al. SOFTS: Efficient Multivariate Time Series Forecasting with Series-Core Fusion. NeurIPS 2024.

---

> ### Author Response · Authors · 2024-11-27
> **Rebuttal by Authors (Comments 6 and 7)**
>
> **Comment 6:** To address the reviewers’ concerns regarding passengers’ privacy, we emphasize that the data we access is highly limited and includes only the following: card ID, card type, and the timestamp and location of turnstile crossings. Regarding the card ID, we use an internal reference that differs from the actual user ID printed on their card. As a result, even if this internal ID is shared, users cannot identify themselves in our database.
> Additionally, we have implemented an extra layer of privacy protection to prevent any potential linkage of user information, even in the unlikely event of a data leak involving the internal reference. To achieve this, we transformed the internal ID using a hashing mechanism, and the corresponding hash table is not included in the shared repository. This approach ensures compliance with Brazilian data protection laws and further safeguards user privacy.
> In summary, it is not possible to identify individual users based solely on the information we have shared, which includes only the boarding time and location for anonymized IDs.
>
> **Comment 7:** By looking for qualitative insights to support transportation planners and policymakers, we have developed an application capable of identifying the bus stops with the highest maximum load for each line over time. According to transportation experts collaborating with us on this project, a particularly valuable feature is the ability to detect stops that experience the highest passenger concentrations during specific time intervals. This information enables more effective planning and distribution of buses, ultimately improving service to meet the needs of the population more efficiently. In the following figure, we have selected one of these lines and analyzed the max-load stops in four different time intervals: i) rush time at 7 a.m., and 4 p.m., (A) and (B), respectively; and ii) off-peak time at 10 a.m., and 3 p.m., (C) and (D), respectively. By analyzing this maps, one might be able to understand how the maximum load stops are varying within different time intervals. This information facilitates more effective bus planning and distribution, enhancing service delivery to better meet the population's needs. More details about how these maps were produced are discussed in our manuscript.
>
> [Image with max load over time](https://github.com/suntdataset/sunt/blob/main/outputs/max_load_times.jpeg)

---

> ### Author Response · Authors · 2024-11-27
> **Rebuttal by Authors (Comments 8 and 9)**
>
> Comment 8: We thank the reviewer for their valuable comment. We agree that a computational cost analysis for processing the full dataset adds to the value of our work. To bridge this lack, the revised supplementary material includes Tables 1 and II, which report computational costs for extracting data and executing GNN models, respectively. Table 1 shows Execution time (in seconds) and Peak Memory (in Megabytes) for each step of the dataset construction. Table 2 shows the Current Memory (in Megabytes), Execution Time (in seconds), and Peak Memory (in Megabytes) spent by each tested model. The table shows that PANConv and SuperGATConv spent more processing time than the other models. SuperGATConv achieved the highest Peak Memory followed by GAT. Also, GAT consumed the most memory on average. We hope these tables show the feasibility of the SUNT dataset concerning scalability, construction, and GNN execution.
>
> Table 1. Information about execution time and memory peak estimated for the main functions to create the proposed dataset.
>
> | **Step**                        | **Execution Time (s)** | **Peak Memory (MB)** |
> |---------------------------------|-------------------------|-----------------------|
> | **I) Load Data**                | 3849                   | 2457.1               |
> | Read GTFS files              | 43                     | 318.4                |
> | Read LTI file               | 42                     | 320.0                |
> | Read AVL file                | 141                    | 1754.9               |
> | Read Trip Time Series file   | 2072                   | 2186.0               |
> | Read AFC validator file      | 1548                   | 2457.1               |
> | **III) Boarding**               | 2769                   | 3950.2               |
> | Estimating boarding          | 2713                   | 3950.2               |
> | **IV) Landing**                 | 2672                   | 4497.9               |
> | Estimating landing           | 2640                   | 4497.9               |
> | **V) Count Boarding/Alighting** | 591                    | 6063.0               |
> | Estimated                    | 128                    | 4503.5               |
> | Adjusted                     | 155                    | 6005.0               |
> | Max load FR                  | 155                    | 6063.0               |
> | Ranges                       | 8                      | 6086.0               |
> | **VII) Calculate FR**           | 10                     | 6086.3               |
> | **Total**                       | 9892                   | 6088.6               |
>
>
> Table 2. Information about mean and peak memory usage and execution time for the GNN models.
> | **Model** | **Mean Memory (MB)** | **Execution Time (s)** | **Peak Memory (MB)** |
> |-------------------|----------------------|---------------------|------------------|
> | A-DGN | 40.75 | 0.398202 | 305.57 |
> | EGC | 34.93 | 0.473959 | 299.40 |
> | LEConv | 42.60 | 0.446057 | 304.59 |
> | PAN | 34.09 | 5.704129 | 271.49 |
> | SuperGAT | 38.64 | 1.558936 | 517.16 |
> | CHEB | 39.93 | 0.427713 | 301.31 |
> | GAT | 81.78 | 0.583302 | 335.01 |
> | GCN | 35.55 | 0.445687 | 298.96 |
> | SAGE | 38.54 | 0.408992 | 299.97 |
> | S²GC | 34.38 | 0.341934 | 298.08   |
>
> **Comment 9:** To address this issue, we have included a section in the manuscript to inform readers about potential estimation errors. However, these errors primarily affect local collaborators from the transportation company. Through meetings and collaborative work, we have concluded that achieving perfect control over all equipment across our 2,000 vehicles is impossible, even when using reliable devices and conducting regular maintenance. Monitoring system errors is an inherent aspect of any computational environment.
> Nonetheless, we emphasize that these errors are less problematic for policymakers compared to the lack of monitoring and passenger estimation. For researchers utilizing our dataset, results are based on simulations and will remain fully comparable to our benchmarks. Finally, we stress that we plan to keep the database updated. If all vehicles are eventually replaced with more precise equipment, our dataset will be adjusted to reflect these improvements.

---

### Meta-Review · Area_Chair_HhYq · 2024-12-20

**Metareview:**

This paper presents a new large scale spatiotemporal dataset from Salvador and a benchmark performed on the dataset. The dataset is very comprehensive, covering multiple sources (AVL, AFC, GTFS, LTI). The dataset construction is well documented. The focus on practical public transport management tasks are commendable. The weakness on the dataset part is that the data only comes from one city. The novelty of the contributions is also raised by reviewers.
The authors performed a comprehensive benchmark on the dataset. The focus of the benchmark is only on GNN-based methods.
The authors were also highly engaged during the rebuttal and discussion period. As a result of the discussion and feedback from the reviewers, the authors have added more details about the dataset, comparisons with other datasets, new experiments on new tasks, related to public transport management.
All the efforts are commendable. However, the reviewers are still not convinced of the novelty of the contributions. All the reviewers kept their score.
As such, I can only recommend this borderline paper as a reject case, especially given that most of the discussed changes are still not yet reflected on the paper. I sincerely wish the authors to incorporate all the reviewers' comments, improve the framing of the paper's contributions-- particularly on the new domain-specific tasks.

**Additional Comments On Reviewer Discussion:**

The discussion period with the authors was very active. However there seems to be no champion for this paper from the reviewers' end. All are still convinced that the technical contribution is limited.

---

### Decision · Program_Chairs · 2025-01-22

Reject